# Reconstitution of BNIP3/NIX-mitophagy initiation reveals hierarchical flexibility of the autophagy machinery

Elias Adriaenssens [1,2,3] ✉, Stefan Schaar [1,2,3], Annan S. I. Cook[3,4,5], Jan F. M. Stuke[6], Justyna Sawa-Makarska [1,2,3], Thanh Ngoc Nguyen [3,7,8,9], Xuefeng Ren [3,4,10], Martina Schuschnig[1,2], Julia Romanov[1,2], Grace Khuu [3,7,8,9], Louise Uoselis[3,7,8,9], Michael Lazarou [3,7,8,9], Gerhard Hummer [3,6,11], James H. Hurley [3,4,5,10,12] & Sascha Martens [1,2,3] ✉

Selective autophagy is a lysosomal degradation pathway that is critical for maintaining cellular homeostasis by disposing of harmful cellular material. Although the mechanisms by which soluble cargo receptors recruit the autophagy machinery are becoming increasingly clear, the principles governing how organelle-localized transmembrane cargo receptors initiate selective autophagy remain poorly understood. Here we demonstrate that the human transmembrane cargo receptors can initiate autophagosome biogenesis not only by recruiting the upstream FIP200/ULK1 complex but also via a WIPI–ATG13 complex. This latter pathway is employed by the BNIP3/NIX receptors to trigger mitophagy. Additionally, other transmembrane mitophagy receptors, including FUNDC1 and BCL2L13, exclusively use the FIP200/ULK1 complex, whereas FKBP8 and the ER-phagy receptor TEX264 are capable of utilizing both pathways to initiate autophagy. Our study defines the molecular rules for initiation by transmembrane cargo receptors, revealing remarkable flexibility in the assembly and activation of the autophagy machinery, with important implications for therapeutic interventions.

Selective autophagy maintains cellular homeostasis by ensuring the degradation of damaged or superfluous components such as organelles, protein aggregates and cytosol-invading pathogens within lysosomes. This targeted removal is orchestrated by cargo receptors, which link the cargo material to the autophagy machinery[1].

A crucial distinction exists between soluble and transmembrane cargo receptors. Soluble cargo receptors, such as SQSTM1/p62, NBR1, TAX1BP1, NDP52 and optineurin (OPTN), are dynamically recruited to the cargo material upon its ubiquitination. Once recruited, these receptors attract components of the upstream machinery to induce autophagosome biogenesis in proximity to the cargo[2]. Canonically, the cargo receptors recruit the FIP200 proteins, a subunit of the upstream ULK1 kinase[3–6]. Recently, it was shown that OPTN recruits the TBK1 kinase and ATG9A, which are also upstream factors in selective autophagy[7,8].

In contrast, transmembrane cargo receptors reside on the various organelles and display a great diversity in terms of number and structure. Currently, over 15 different membrane-embedded cargo receptors are known, and the list is expanding rapidly. Notably, for mitochondria these include BNIP3[9–11], NIX[12–15] (also known as BNIP3L), FKBP8[16], PHB2[17], NLRX1[18], MCL-1[19], FUNDC1[20] and BCL2L13[21]; for the endoplasmic reticulum (ER), ATL3[22], CCPG1[23], FAM134A[24], FAM134B[25], FAM134C[24,26], Sec62[27], RTN3[28] and TEX264[29,30]; for the Golgi apparatus, YIPF3 and YIPF4[31]; and for peroxisomes, NIX and BNIP3[32].

Although the mechanisms of autophagy initiation by soluble cargo receptors have been elucidated, the process by which transmembrane cargo receptors recruit the autophagy machinery remains less clear. Ubiquitin-driven clustering was recently shown to be a contributing factor for FAM134B activation[33,34]. It remains unknown how universal this mechanism is among the different transmembrane receptors and how the autophagy machinery is recruited and activated. Given the large number of transmembrane cargo receptors spread across the different organelles, understanding their mode of action is crucial for a comprehensive understanding of selective autophagy.

In this Article we investigate the mechanism of autophagosome biogenesis by transmembrane cargo receptors. We find that, in contrast to soluble cargo receptors, transmembrane cargo receptors can initiate autophagosome biogenesis through two distinct pathways: one by recruiting the upstream FIP200/ULK1 complex and another by recruiting a WIPI–ATG13 complex. Our results reveal unexpected flexibility among selective autophagy pathways and show that the general principles of soluble cargo receptors do not universally apply to all transmembrane cargo receptors.

## Results

### NIX and BNIP3 are unable to bind FIP200

Human cells express numerous transmembrane cargo receptors, typically several for each organelle[35]. To understand how these receptors recruit the autophagy machinery, we focused on mitochondria, where several transmembrane cargo receptors have been identified (Fig. 1a)[36]. Unlike other organelles such as the ER, mitochondria can be targeted for selective autophagy using chemical agents like deferiprone (DFP) that induce mitophagy via individual receptors[10].

To investigate the recruitment process of the autophagy machinery by transmembrane mitophagy receptors, we reconstituted the initiation of autophagosome biogenesis using purified components. We purified the soluble, cytosol-exposed domains of BNIP3, NIX, FUNDC1 and BCL2L13 (Fig. 1b), substituting the transmembrane domains with green fluorescent protein (GFP)- or glutathione S-transferase (GST)-moieties to study the mitophagy receptors in either a monomeric or dimeric state. For instance, for NIX and BNIP3, the activated state is thought to be a dimer[37], but for FUNDC1 and BCL2L13 this is yet to be elucidated.

To confirm that our purified mitophagy receptors are active, we tested their ability to bind LC3 and GABARAP proteins using a microscopy-based bead assay (Extended Data Fig. 1a). Similar to soluble cargo receptors, GABARAP proteins were bound more readily, whereas LC3 proteins showed varying degrees of binding, depending on the receptor. Specificity was verified by mutating the LC3-interacting (LIR) motifs, resulting in the loss of binding for NIX, BNIP3 and FUNDC1 (Extended Data Fig. 1b). For BCL2L13, multiple functional LIR motifs were observed (Extended Data Fig. 1c,d), similar to how the yeast Atg19 interacts with Atg8[38].

Next we sought to determine how they recruit the remaining autophagy machinery. Soluble cargo receptors, such as SQSTM1/p62, initiate autophagosome biogenesis by binding to FIP200 through a FIP200-interacting (FIR) motif that docks into a conserved groove of the C-terminal FIP200 Claw domain[4]. We therefore tested whether the transmembrane mitophagy receptors could also bind the C-terminal region of FIP200, which encompasses the Claw domain and a portion of the coiled-coil domain (residues 1429–1591). Using microscopy-based bead assays, we observed that FUNDC1 and BCL2L13, but not BNIP3 or NIX, directly bind to the C-terminal FIP200 domain (Fig. 1c). Moreover, mutating the LIR/FIR motifs of FUNDC1 or BCL2L13 abrogated this interaction (Extended Data Fig. 1e).

Not all soluble cargo receptors bind to FIP200 in the Claw domain. For instance, NDP52 binds the coiled-coil region just upstream of the C-terminal region[5,6,39]. We thus tested whether BNIP3 and NIX could bind to full-length FIP200 (Fig. 1d). However, we were unable to detect a direct interaction between BNIP3/NIX and FIP200.

Next we asked if BNIP3/NIX require activation by a kinase, such as TBK1, which is known to phosphorylate soluble cargo receptors and cargo co-receptors to enhance their LC3-binding capacities[40,41]. In particular, we tested four candidate kinases: TBK1, ULK1, Src and casein kinase 2 (CK2). TBK1 and ULK1 have previously been shown to play essential roles in selective autophagy pathways involving soluble cargo receptors[7,41,42], and Src and CK2 have been associated with hypoxia-induced mitophagy[20,43]. We therefore purified TBK1, MBP-ULK1, Src and CK2 (Extended Data Fig. 2a) and confirmed their activity (Extended Data Fig. 2b–e). In microscopy-based protein–protein interaction assays between BNIP3/NIX and either full-length or the C-terminal region of FIP200, we observed that, although the positive controls FUNDC1 and BCL2L13 were able to bind FIP200, the addition of the kinases and ATP/MgCl$_2$ did not facilitate interaction between BNIP3/NIX and FIP200 (Fig. 1e,f).

We hypothesized that purified BNIP3/NIX might already be pre-phosphorylated, which could inhibit their FIP200 interaction. To test this, we performed a microscopy-based bead assay in the presence of lambda protein phosphatase after validating its activity (Extended Data Fig. 2f). BCL2L13 and FUNDC1 would readily bind to FIP200 under these conditions, but we could not observe a direct binding of NIX to FIP200 (Fig. 1g,h).

Some soluble cargo receptors, such as optineurin, have been shown to recruit other components of the upstream autophagy machinery[7,8,44]. Therefore, we tested whether BNIP3/NIX could initiate autophagy not by recruiting FIP200 but through the recruitment of TBK1, the PI3KC3–C1 complex, or ATG9A-vesicles. However, no interaction of these factors with BNIP3/NIX was observed (Extended Data Fig. 3a).

In summary, although our findings confirm that the mitophagy receptors FUNDC1 and BCL2L13 directly bind to FIP200, we could not detect any direct binding between the mitophagy receptors BNIP3/NIX and FIP200 or other upstream autophagy machinery components.

### NIX and BNIP3 initiate mitophagy by recruiting WIPI proteins

Because we were unable to establish a direct interaction between BNIP3/NIX and any of the tested upstream autophagy machinery, we explored whether BNIP3/NIX utilize an alternative mechanism for recruiting the autophagy machinery upon mitophagy induction. Recent studies have shown that NIX interacts with WIPI2[45], a downstream factor in the autophagy cascade, and PPTC7[46,47], a mitochondrial phosphatase that accumulates on the mitochondrial surface upon iron depletion by DFP treatment[46,48,49].

To identify other potential interactors of NIX and BNIP3 that could link these receptors to the upstream autophagy machinery, we performed a pulldown with GST-tagged BNIP3/NIX and HeLa cell lysates. Mass spectrometry analysis revealed that PPTC7 was the strongest binder for NIX and one of the strongest binders for BNIP3 (Fig. 2a). Additionally, we detected WIPI2 among the top binders for NIX, and WIPI3 as a top binder for BNIP3. The interaction between BNIP3 and WIPI3 has not been reported before, but, given the concomitant interaction between NIX and WIPI2 and the absence of other upstream autophagy components in our dataset, it suggests a potentially important role for WIPI2 and WIPI3 in BNIP3/NIX-mediated mitophagy.

The processes triggered by the direct recruitment of WIPI proteins by the cargo receptors BNIP3/NIX in mitophagy, typically recruited only after the upstream ULK1– and PI3KC3–C1 complexes have been loaded onto ATG9-vesicle seeds, are unclear. However, given our failure to identify any upstream regulatory factors of the autophagy machinery, we decided to investigate the interaction with WIPI proteins in more detail.

First, to confirm the mass spectrometry results, we incubated GST, NIX-GST and BNIP3-GST with HeLa cell lysate and immunoblotted for different WIPI proteins. Indeed, NIX and BNIP3 bound WIPI2, and BNIP3 also pulled down WIPI3 (Fig. 2b and Extended Data Fig. 3b). To test whether NIX and BNIP3 bind WIPI2 and WIPI3 directly, we

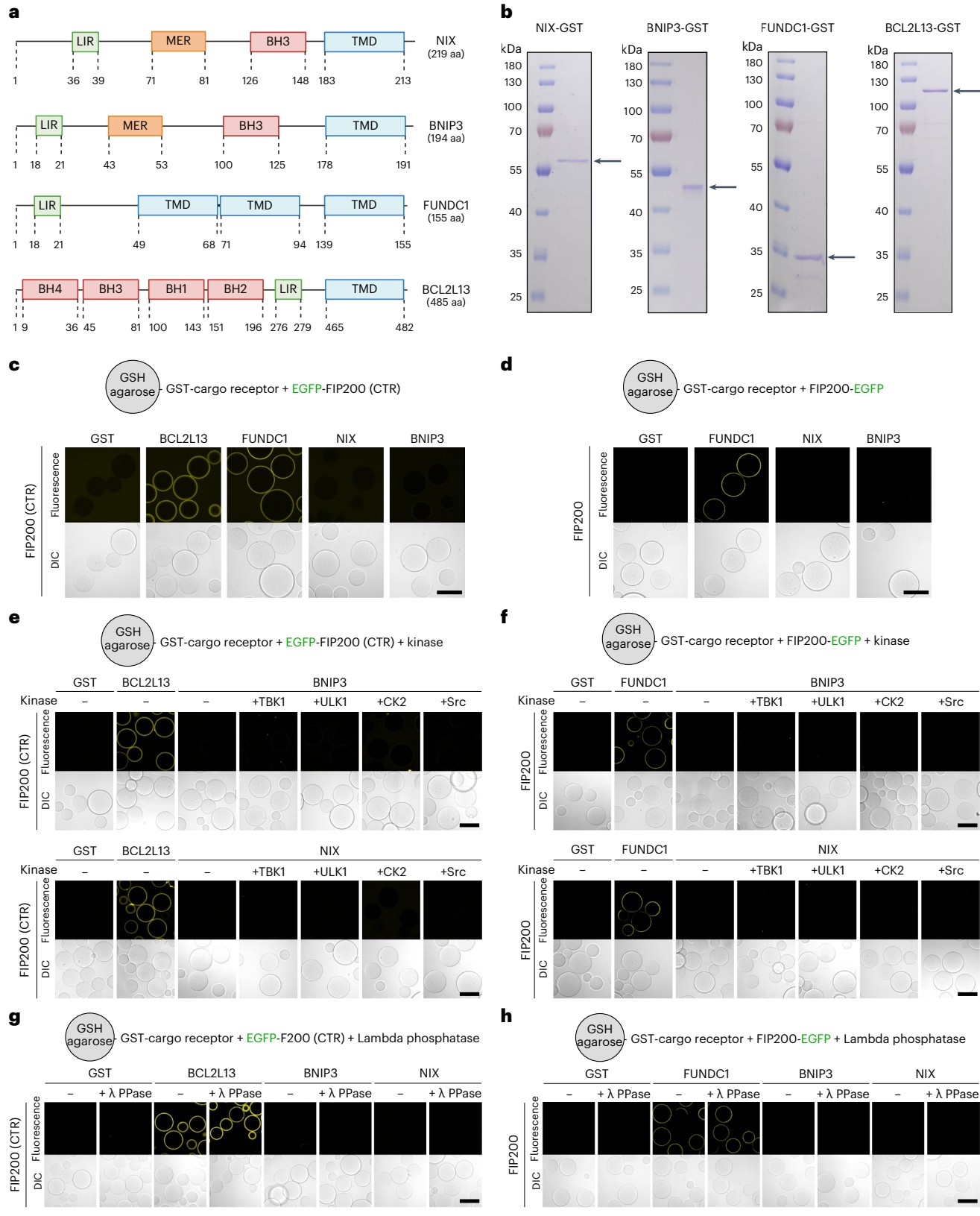

**Fig. 1 | NIX and BNIP3 are unable to bind FIP200 in vitro. a**, Schematic of the domain structures of NIX, BNIP3, FUNDC1 and BCL2L13. MER, minimal essential region; BH, Bcl-2 homology domain; TMD, transmembrane domain. **b**, Representative SDS–PAGE gels of NIX-GST, BNIP3-GST, FUNDC1-GST and BCL2L13-GST. Arrows indicate the predicted molecular weight. **c–h**, Microscopy-based bead assay of agarose beads coated with the indicated GST-tagged cargo receptors and incubated with GFP-tagged FIP200-CTR (residues 1429–1591) (**c**),

GFP-tagged full-length FIP200 (**d**), FIP200-CTR and kinases TBK1, MBP-ULK1, CK2 or Src (Y530F; constitutively active mutant) (**e**), full-length FIP200 and kinases TBK1, MBP-ULK1, CK2 or Src (Y530F; constitutively active mutant) (**f**), FIP200-CTR and lambda protein phosphatase (**g**), full-length FIP200 and lambda protein phosphatase (**h**). Samples were analysed by confocal imaging. Scale bars, 100 μm. Results are representative of three replicates (**b–h**). Unprocessed blots are available in the source data. Schematic generated with BioRender.

incubated purified WIPI1-4 with NIX- or BNIP3-coated beads. This revealed that NIX binds to WIPI2, but not WIPI3, under these conditions (Fig. 2c), consistent with our mass spectrometry dataset. We also observed that NIX can bind to WIPI1. For BNIP3, we detected an interaction with WIPI2 and a much stronger binding to WIPI3 (Fig. 2d).

Using AlphaFold2 (AF2) Multimer, we modelled the NIX–WIPI2 and BNIP3–WIPI2 complexes. These predictions suggested that a short amino-acid stretch, conserved between NIX and BNIP3, interacts with WIPI2 (Fig. 2e and Extended Data Fig. 4a–h). To test this model, we introduced point mutations in the predicted binding interfaces and observed a complete loss of binding between NIX and WIPI2 (Fig. 2f). Interestingly, we also observed a role for the LIR motif of NIX, as mutating the LIR motif abrogated the interaction (Fig. 2g). Consistently, mutating the LIR motif of BNIP3 abrogated the BNIP3–WIPI2 and BNIP3–WIPI3 interactions (Fig. 2h).

We then employed further AF2 Multimer modelling and molecular dynamics (MD) simulations to model where the LIR motif of NIX may engage with WIPI2d. This revealed an interaction of the LIR with the surface of WIPI2d (Fig. 2i and Extended Data Fig. 4i) that was, with some minor structural rearrangements, stable for several hundred nanoseconds in our MD simulations (Fig. 2j). Interestingly, we observed the opening of a cryptic pocket in WIPI2d, which accommodated the Trp residue of the LIR of NIX (Fig. 2k), suggesting a possible mechanism for the LIR–WIPI2d interaction. When we mutated the LIR motif, it was no longer predicted to bind the cryptic pocket in WIPI2d (Extended Data Fig. 4j), consistent with our biochemical data. Combined, our biochemical and MD data reveal that BNIP3/NIX bind WIPI2 using two motifs.

To assess the importance of the BNIP3/NIX–WIPI interactions in cells, we generated BNIP3/NIX double-knockout HeLa cells and confirmed their defect in DFP-induced mitophagy (Extended Data Fig. 5a,b). We then rescued these cells with wild-type BNIP3, wild-type NIX, WIPI2-binding-deficient or LIR-deficient NIX mutants. This revealed that the BNIP3/NIX–WIPI interactions are essential for DFP-induced mitophagy, as both WIPI2-binding-deficient and LIR-deficient NIX were unable to rescue the knockouts (Fig. 2l).

Our data thus reveal that NIX and BNIP3 use two binding motifs to interact with WIPI2 and/or WIPI3, respectively. Furthermore, we demonstrate that these interactions are essential for BNIP3/NIX-mediated mitophagy.

### Recruitment of WIPI1/2/3 suffices to initiate mitophagy
To investigate whether the BNIP3/NIX-mediated recruitment of WIPI proteins—typically considered downstream factors—is sufficient for mitophagy initiation, we artificially tethered WIPI proteins to the mitochondrial surface. Using the FK506 binding protein (FKBP) and FKBP-rapamycin binding (FRB) system, which facilitates chemical-induced dimerization, we generated FKBP–GFP–WIPI fusion

proteins for WIPI1, WIPI2, WIPI3 and WIPI4, and expressed those constructs via stable lentiviral transduction in HeLa cells expressing Fis1-FRB (Fig. 3a). By co-expressing the mitochondrially targeted monomeric Keima (mt-mKeima) probe, we assessed mitochondrial turnover to determine whether the recruitment of WIPI proteins to the mitochondrial surface could initiate mitophagy. The addition of rapalog resulted in a strong induction of mitophagy for WIPI1, WIPI2 and WIPI3, but not WIPI4 (Fig. 3b).

To confirm that this mitochondrial turnover was mediated by autophagy, we repeated the experiment for WIPI1, WIPI2 and WIPI3 in the presence of a VPS34 kinase inhibitor, which blocks autophagosome formation (Fig. 3c). VPS34-inhibitor treatment completely inhibited mitochondrial turnover, confirming that tethering WIPI1, WIPI2 and WIPI3 to the mitochondrial surface is sufficient to induce mitophagy.

We repeated the tethering of WIPI2 in BNIP3/NIX double-knockout cells to test if it can act downstream and independent of BNIP3/NIX. Indeed, we still observed robust mitophagy induction in BNIP3/NIX double-knockout cells (Fig. 3d). We further validated our tethering results with western blotting, confirming that tethering of WIPI1, WIPI2 or WIPI3 leads to the turnover of the mitochondrial protein COXII and is accompanied by lipidation of LC3B (Fig. 3e). Furthermore, quantitative proteomics confirmed that depletion of mitochondrial proteins upon BNIP3/NIX-mitophagy induction with DFP is phenocopied by tethering FKBP–GFP–WIPI2 to the mitochondrial surface in both wild-type and BNIP3/NIX double-knockout cells (Fig. 3f,h).

In summary, tethering WIPI1, WIPI2 or WIPI3 to the mitochondrial surface induces mitophagy. This finding is unexpected, as WIPI proteins are generally considered downstream factors in autophagosome biogenesis, recruited to the expanding phagophore only after PI3KC3–C1 phosphorylation of phosphatidylinositol. However, our data demonstrate that the recruitment of these downstream factors to mitochondria is sufficient to initiate autophagosome formation.

### Mitophagy initiation through WIPIs requires the ULK1 complex
To investigate how WIPI proteins initiate autophagosome biogenesis during mitophagy, we first examined whether upstream autophagy complexes, including the ULK1 complex (composed of FIP200, ATG13, ATG101 and the ULK1 kinase), are recruited. Using the rapalog system, we tethered WIPI2 to the mitochondrial surface and immunostained for ATG13, demonstrating that ATG13 is recruited to mitochondria upon WIPI2 tethering (Fig. 4a). Immunoblotting for phosphorylated ATG13 confirmed that this recruitment coincides with ULK1 complex activation (Fig. 4b).

We next assessed whether upstream autophagy complexes are required for WIPI-mediated mitophagy initiation. Depletion of ATG13 or FIP200 using siRNAs and inhibition of ULK1/2 kinase activity with

---

**Fig. 2 | NIX and BNIP3 initiate mitophagy through WIPI2 and WIPI3.**
**a**, Identification of interactors of NIX(1–182)-GST and BNIP3(1–158)-GST by pulldown from HeLa cell lysates and mass spectrometry. Tables represent the top hits for NIX (upper) and BNIP3 (lower). **b**, Validation of mass spectrometry data by pulldowns with SDS–PAGE and western blot. **c,d**, Microscopy-based bead assays of NIX-GST or BNIP3-GST incubated with mCherry-tagged WIPI1, WIPI2d, WIPI3 or WIPI4. **e**, AF2 predicted structure of NIX or BNIP3 and WIPI2d. Note that the indicated residue numbers for WIPI2 correspond to their residue number in the WIPI2d sequence. Conservation of the interaction interface between NIX and BNIP3 is displayed. **f,g**, As in **c**, but with NIX wild-type (WT), E72A/L75A/D77A/E81A mutant (4A) or W36A/L39A (ΔLIR) and mCherry-tagged WIPI2d WT or K87A/K88A mutant. **h**, As in **c**, but with BNIP3 WT or W18A/L21A mutant (ΔLIR) and mCherry-tagged WIPI3 or WIPI2d. **i**, AF2 Multimer predicted complex structure of WIPI2d and NIX (residues 30–82). The zoom highlights the interaction between the LIR of NIX and WIPI2d. The C-terminal intrinsically disordered region of WIPI2d is omitted for visual clarity. **j**, Number of backbone hydrogen bonds, $n_{\text{H-bonds}}$, between the LIR of NIX and WIPI2d, insertion depth

$d_{\text{TRP}}$ of NIX W36, and minimum heavy atom distance $d_{\text{pocket}}$ between WIPI2d F169 and I133 from three 1-μs MD simulations. **k**, Representative snapshots of W36 interacting with WIPI2d (top) and inserted into an initially closed pocket (bottom). The symbols in the lower left corner indicate the point in the trajectory in **j** from which the respective snapshots were extracted. **l**, Mitophagy flux measured by flow cytometry of WT or NIX/BNIP3 double-knockout (2KO) HeLa cells, rescued with V5-BNIP3, V5-NIX, V5-NIX ΔLIR (W36A/L39A mutant) or V5-NIX ΔWIPI2 (4A mutant; E72A/L75A/D77A/E81A), untreated or treated with DFP for 24 h. The percentage of non-induced cells (lower right) versus mitophagy-induced cells (upper left) is indicated. Representative fluorescence-activated cell sorting (FACS) plots are shown from one of three biological replicates, and data are presented as mean ± s.d. ($n = 3$ biologically independent experiments). Two-way analysis of variance (ANOVA) with Tukey's multiple comparisons test. ****$P < 0.0001$. NS, not significant. Results are representative of three replicates (**b**–**h**). Scale bars, 100 μm. Source numerical data, including exact $P$ values, and unprocessed blots are available in the source data.

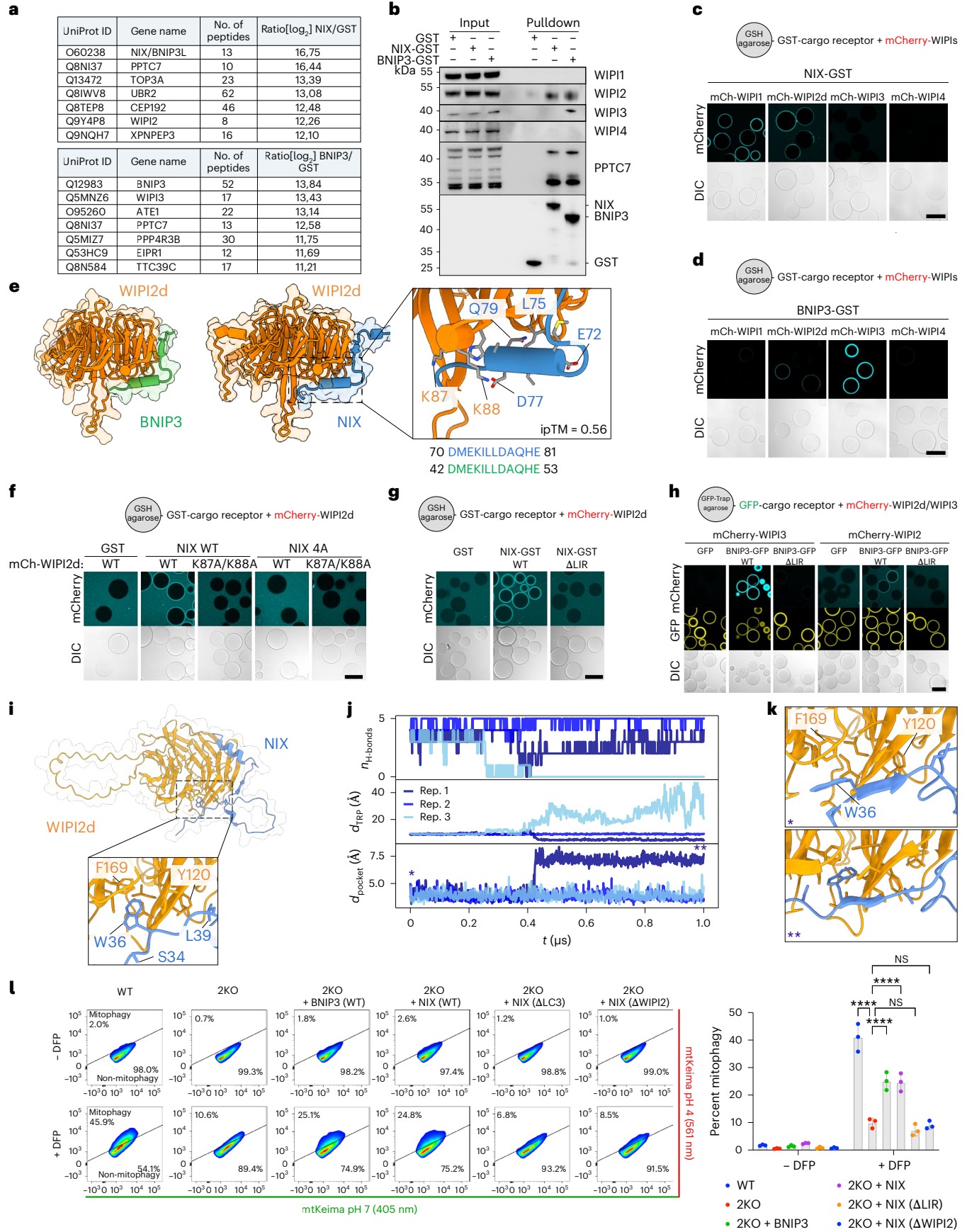

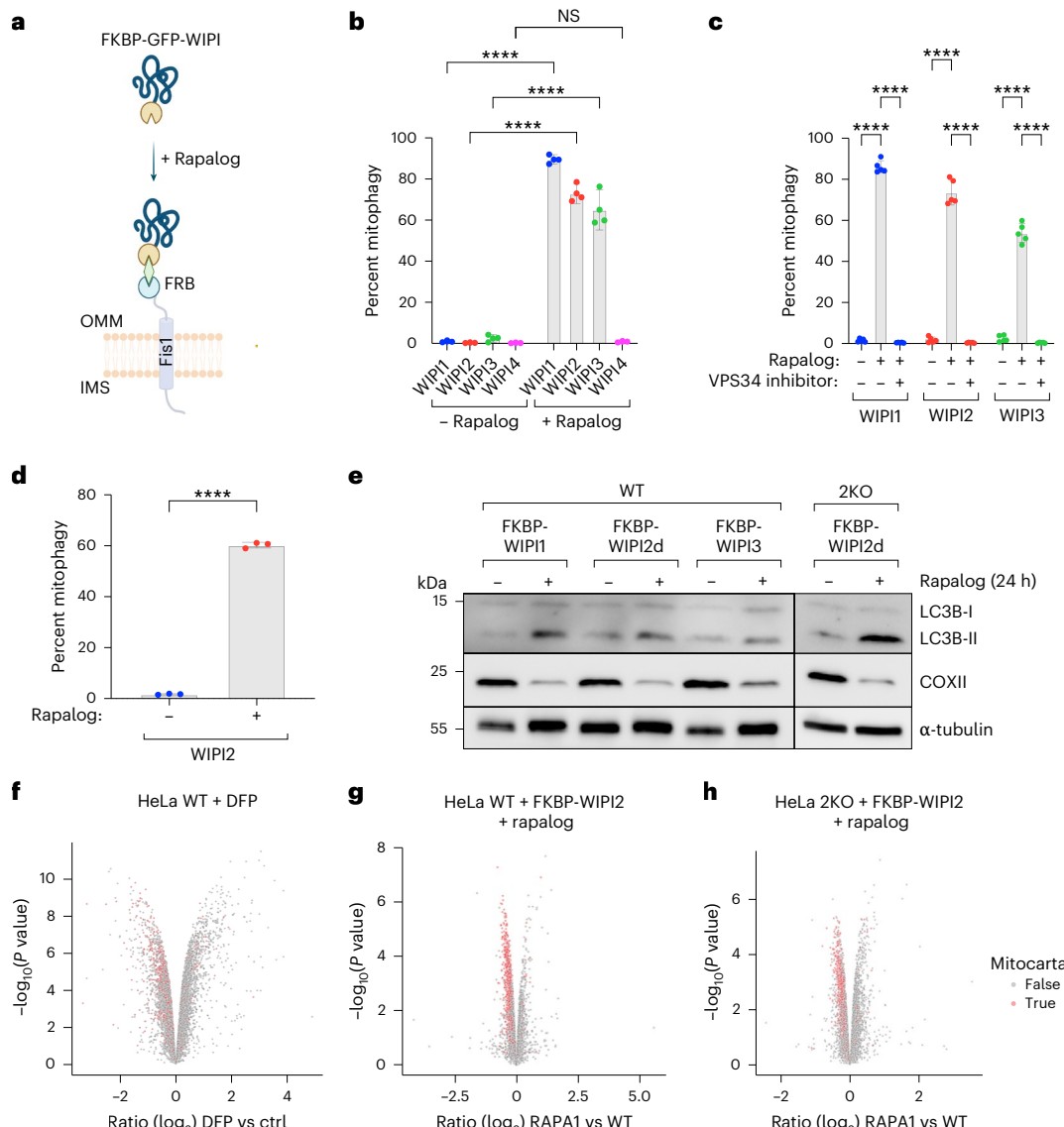

**Fig. 3 | Mitochondrial localization of WIPI1, WIPI2 and WIPI3 can initiate autophagosome biogenesis. a**, Schematic of the experimental approach and effect of rapalog treatment, leading to the tethering of WIPI proteins to the outer mitochondrial membrane. IMS, intermembrane space; OMM, outer mitochondrial membrane. **b**, Mitophagy flux measured by flow cytometry in WT HeLa cells expressing Fis1-FRB, FKBP–GFP–WIPI1/2/3/4 and mt-mKeima, not induced or induced for 24 h by rapalog treatment. Data are presented as mean ± s.d. (*n* = 4 biologically independent experiments). Two-way ANOVA with Šídák's multiple comparisons test. **c**, As in **b** but with or without the addition of the VPS34-inhibitor VPS34-IN1. Data are presented as mean ± s.d. (*n* = 5 biologically independent experiments). Dunnett's multiple comparisons test. **d**, As in **b** but in NIX/BNIP3 2KO HeLa cells expressing Fis1-FRB, FKBP–GFP– WIPI2 and mt-mKeima. Data are presented as mean ± s.d. (*n* = 3 biologically

independent experiments). Unpaired *t*-test. **e**, Immunoblotting for LC3B lipidation (LC3B-II) and turnover of the mitochondrial protein COXII in WT or NIX/BNIP3 2KO HeLa cells before and after tethering of FKBP–WIPI1/2/3 with rapalog. Results are representative of three biological replicates. **f**–**h**, Quantitative proteome analysis of WT HeLa cells treated for 24 h with DFP (**f**) or quantitative proteome analysis of WT (**g**) and NIX/BNIP3 2KO (**h**) HeLa cells treated for 24 h with rapalog to artificially tether FKBP–WIPI2 to the outer mitochondrial membrane. Mitochondrial proteins as annotated in MitoCarta 3.0 are marked in red (*n* = 3 biologically independent experiments). Moderated *t*-statistics were calculated and multiple testing correction was applied using the Benjamini–Hochberg method. ****P < 0.0001. NS, not significant. Source numerical data, including exact *P* values, and unprocessed blots are available in the source data. Schematic generated with BioRender.

MRT68921 significantly reduced mitophagy (Fig. 4c,d). Similarly, inhibiting the kinase activity of VPS34 abrogated mitophagy (Fig. 4e). Together, these findings indicate that the WIPI proteins are recruited downstream of BNIP3/NIX, but upstream of the ULK1 and PI3KC3– C1 complexes. Thus, despite being recruited in an unprecedented sequence, these upstream complexes remain essential for BNIP3/ NIX-mediated mitophagy.

To confirm these results, we examined DFP-induced mitophagy. Depletion of FIP200, ATG13 and ULK1, or pharmacological inhibition of ULK1/2 or VPS34, blocked DFP-induced mitophagy (Fig. 4f–i and

Extended Data Fig. 5c,d). Notably, ULK1 inhibition completely abolished mitophagy, consistent with previous reports[32], whereas inhibition of the kinase TBK1 had no effect (Fig. 4h).

Our findings propose a model in which WIPI1, WIPI2 and WIPI3 initiate autophagosome biogenesis by requiring the ULK1 and PI3KC3–C1 complexes but not TBK1. Unlike PINK1/Parkin-mediated mitophagy, where soluble cargo receptors depend on TBK1[7,40,42,50], transmembrane receptors such as BNIP3/NIX drive selective mitophagy independently of TBK1. This highlights a critical distinction between mitochondrial turnover mediated by transmembrane versus soluble cargo receptors.

Furthermore, our data suggest that BNIP3/NIX recruit WIPI proteins upstream of the FIP200/ULK1 complex, in contrast to PINK1/Parkin-mitophagy, where the FIP200/ULK1 complex precedes WIPI proteins[7]. To directly test this BNIP3/NIX model, we hypothesized that WIPI2 should still localize to mitochondria in FIP200 knockout cells. Indeed, following DFP treatment in wild-type and FIP200 knockout cells (with PPTC7 depletion to enhance BNIP3/NIX-mitophagy), we observed that WIPI2 was recruited to mitochondria even in the absence of FIP200 (Fig. 4j). Interestingly, WIPI2 accumulation was particularly evident in FIP200 knockout cells, further supporting our model that WIPI proteins are recruited directly by BNIP3/NIX. Consistently, WIPI2 was also still recruited when ULK1 kinase activity was inhibited (Extended Data Fig. 6). However, although wild-type cells displayed autophagosome-like structures engulfing mitochondrial fragments upon DFP treatment, these structures were absent in both FIP200 knockout cells and ULK1-inhibited cells (Fig. 4k), reinforcing that the FIP200/ULK1 complex is required downstream of the WIPIs to complete autophagosome biogenesis.

### WIPI2 and WIPI3 bind the ULK1 complex via ATG13/101

Given that BNIP3/NIX cannot directly recruit FIP200 but still require activation of the ULK1 complex downstream of the WIPIs, we aimed to elucidate how the FIP200/ULK1 complex is recruited and define the sequence in which the autophagy machinery components assemble in this pathway. We hypothesized that ATG16L1 might act as a bridging factor, given its known interactions with both WIPI2 and FIP200[51,52]. To test this, we generated a WIPI2 mutant (R108E/R125E) that is deficient in ATG16L1-binding[51]. Upon rapalog treatment, mitophagy was induced by wild-type FKBP–GFP–WIPI2 and the ATG16L1-binding deficient WIPI2 mutant (Fig. 5a), suggesting that WIPI2 can recruit the ULK1 complex independently of its ATG16L1-binding ability. This finding aligns with our observation that BNIP3/NIX occupy the ATG16L1-binding site on WIPI2, indicating that these interactions are probably mutually exclusive and that the R108E/R125E mutant co-immunoprecipitates more ULK1[53].

We then investigated whether WIPI proteins might directly bind the ULK1 complex. Indeed, WIPI2d and WIPI3 were recruited to beads coated with GFP-tagged ULK1 complex (Fig. 5b), with WIPI2d showing stronger binding than WIPI3. To identify which ULK1 complex subunits interact with WIPI proteins, we incubated mCherry-tagged WIPIs with individual ULK1 complex subunits. WIPI2d bound to the heterodimeric ATG13/101 subcomplex and weakly to FIP200, but not the ULK1 kinase subunit (Fig. 5c–e). WIPI3 bound only to the ATG13/101 subcomplex.

Structurally, the four WIPI proteins share a similar seven-blade β-propeller domain, with each blade composed of four antiparallel β-strands[54–56]. WIPI2 contains a binding site for ATG16L1 between blades 2 and 3[57,58]. Both WIPI1 and WIPI2 have a C-terminal intrinsically

disordered region (IDR), whereas WIPI3 lacks this IDR but still binds ATG13/101. We thus hypothesized that the interaction is mediated by the β-propeller domains.

To test this, we attempted to purify WIPI2d without its C-terminal IDR, but its low solubility prevented successful purification of the β-propeller domain alone. Instead, we purified the C-terminal IDR and, consistent with our hypothesis, found that it was unable to recruit ATG13/101 to mCherry–WIPI2d-IDR-coated beads (Fig. 5f). Additionally, when we artificially tethered the IDR of WIPI2d to the mitochondrial surface in HeLa cells, robust mitophagy induction was no longer observed, unlike when the full-length WIPI2d was tethered (Fig. 5g).

These findings suggest the existence of a novel autophagy initiation complex involving the β-propeller domains of WIPI proteins and the ATG13/101 subcomplex.

### Characterization of the WIPI–ULK1 initiation complex

To structurally characterize the WIPI–ATG13/101 mitophagy initiation complex in more detail, we set out to identify the minimal binding region between WIPI proteins and ATG13/101. ATG13 contains a HORMA domain and a C-terminal IDR region[59–61], whereas ATG101 only contains a HORMA domain necessary for dimerization with ATG13[60,61]. We investigated whether WIPI proteins bind to the ATG13/101 HORMA dimer or the ATG13 IDR by incubating WIPI2d and WIPI3 with either the ATG13 IDR or the ATG13/101 HORMA dimer lacking the IDR. Our results showed that WIPI2d and WIPI3 bind to the ATG13 IDR but not the HORMA domain dimer (Fig. 6a).

Next, we mapped the minimal binding region using truncated versions of ATG13. We found that the initial stretch of the ATG13 IDR (191–230 aa) is both required and sufficient to bind both WIPI2d and WIPI3 (Fig. 6b and Extended Data Fig. 7). Our biochemical mapping suggests that WIPI2d and WIPI3 bind neighbouring sequences on the ATG13 IDR (residues 191–202 for WIPI2d; residues 206–230 for WIPI3). We confirmed this by expressing the ATG13 IDR alone, without the HORMA domain, and deleting the entire binding region (residues 191–230) or only the minimal binding regions for WIPI2d (residues 191–205) or WIPI3 (residues 206–230). The results confirmed that the ATG13 IDR could still recruit WIPI2d if residues 191–205 were present and WIPI3 if residues 206–230 were present (Fig. 6c).

To identify the interacting residues within these minimal binding regions, we predicted the structure of the complex using AF2 Multimer. After removing the ten most carboxyl-terminal residues from WIPI2d, which were incorrectly predicted to bind the HORMA dimer, AF2 Multimer correctly predicted that WIPI2d binds the initial segment of the ATG13 IDR (Fig. 6d and Extended Data Fig. 8). The prediction suggested that ~20 residues interact directly with the WIPI2d β-propeller domain. To validate this, we created two ATG13 IDR variants: one with three

**Fig. 4 | ULK1 complex and PI3KC3–C1 complex are required downstream of WIPI-driven autophagosome biogenesis. a**, Representative maximum intensity projection images of WT HeLa cells stably expressing Fis1-FRB and FKBP–GFP–WIPI2. Cells were left untreated or treated with rapalog for 16 h and immunostained for ATG13. Scale bars, 20 μm and 10 μm (zooms). Results are representative of two biologically independent replicates. **b**, Immunoblotting for phosphorylated ATG13 in HeLa cells overexpressing Fis1-FRB and FKBP–EGFP–WIPI2d, treated with rapalog for the indicated time. Results are representative of three biologically independent replicates. **c**, Mitophagy flux measured by flow cytometry in WT HeLa cells transfected with siRNAs targeting FIP200 or ATG13, and expressing Fis1-FRB, FKBP–GFP–WIPI1/2/3 and mt-mKeima, not induced or induced for 24 h by rapalog treatment. Data are presented as mean ± s.d. (n = 4 biologically independent experiments). Two-way ANOVA with Dunnett's multiple comparisons test. **d,e**, As in **c**, but with or without the addition of the ULK1/2 inhibitor MRT68921 (**d**) or the VPS34-inhibitor VPS34-IN1 (**e**). Data are presented as mean ± s.d. (n = 6 biologically independent experiments in **d** and n = 3 in **e**). Two-way ANOVA with Dunnett's multiple comparisons test. **f,g**, As in **c**, but with WT HeLa cells expressing mt-mKeima and transfected with

siRNAs targeting ATG13, FIP200 or ULK1, and treated with DFP for 24 h. Data are presented as mean ± s.d. (n = 3 biologically independent experiments). One-way ANOVA with Dunnett's multiple comparisons test. **h,i**, As in **f**, but with the kinase inhibitors GSK8612 for TBK1, MRT68921 for ULK1/2, VPS34-IN1 for VPS34, or bafilomycin A1 (BafA1). Data are presented as mean ± s.d. (n = 3 biologically independent experiments in **h** and n = 4 in **i**). Two-way ANOVA with Dunnett's multiple comparisons test (**i**) or one-way ANOVA with Dunnett's multiple comparisons test (**h**). **j**, Immunofluorescence images of WIPI2 and mitochondrial HSP60 in WT or FIP200 KO HeLa cells, untreated or treated for 24 h with DFP. All cells were depleted for PPTC7, aiding the visualization of mitophagy events. Scale bars, 20 μm and 5 μm (zoom). Data are representative of two biologically independent experiments. **k**, Quantification of the percentage of cells in each field of view that contained autophagosome-like cup structures that colocalized with the mitochondrial marker HSP60. Data are presented as mean ± s.d. (n = 4 biologically independent biological samples). One-way ANOVA with Dunnett's multiple comparisons test. **P < 0.005, ***P < 0.001, ****P < 0.0001. NS, not significant. Source numerical data, including exact P values, and unprocessed blots are available in the source data.

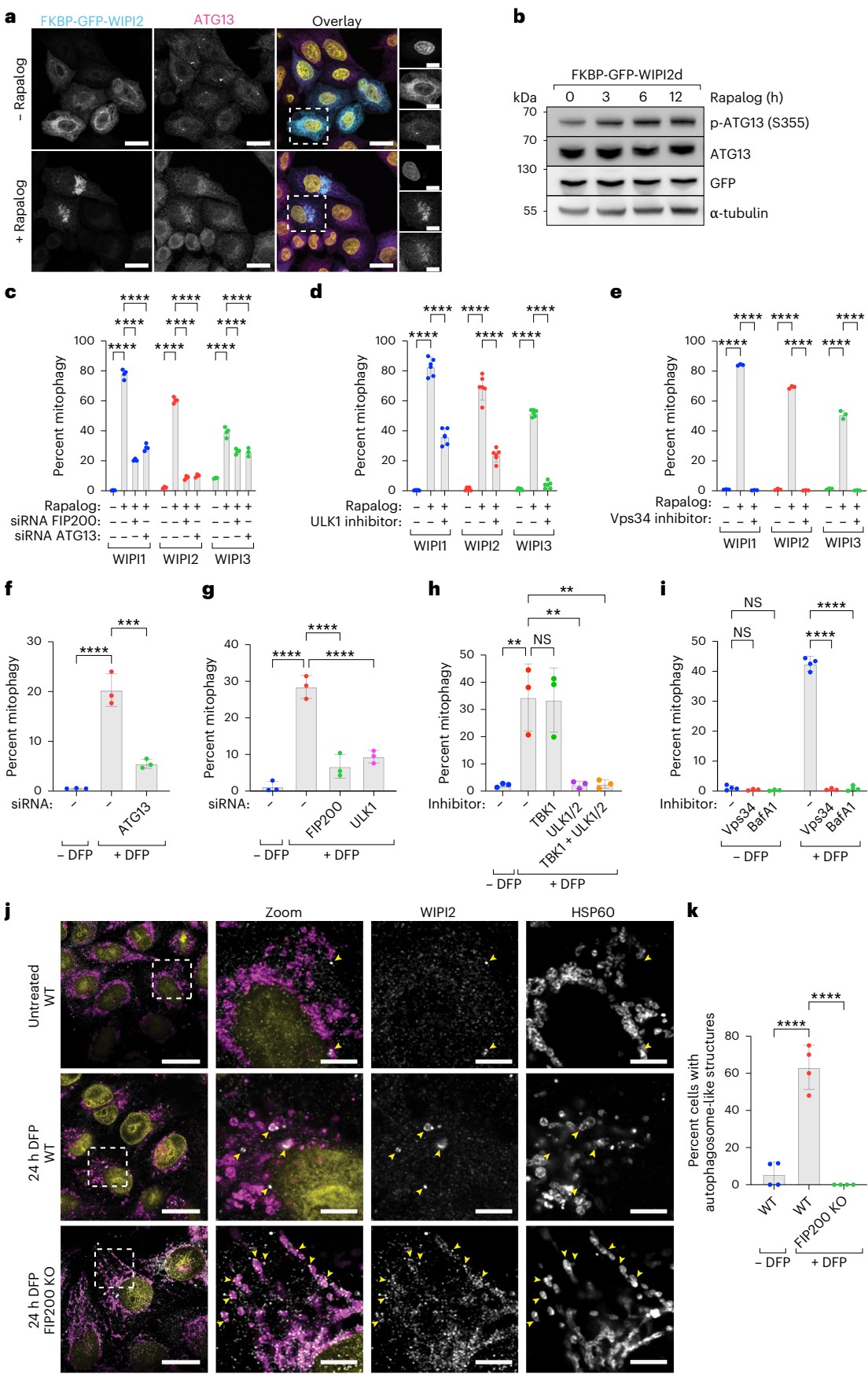

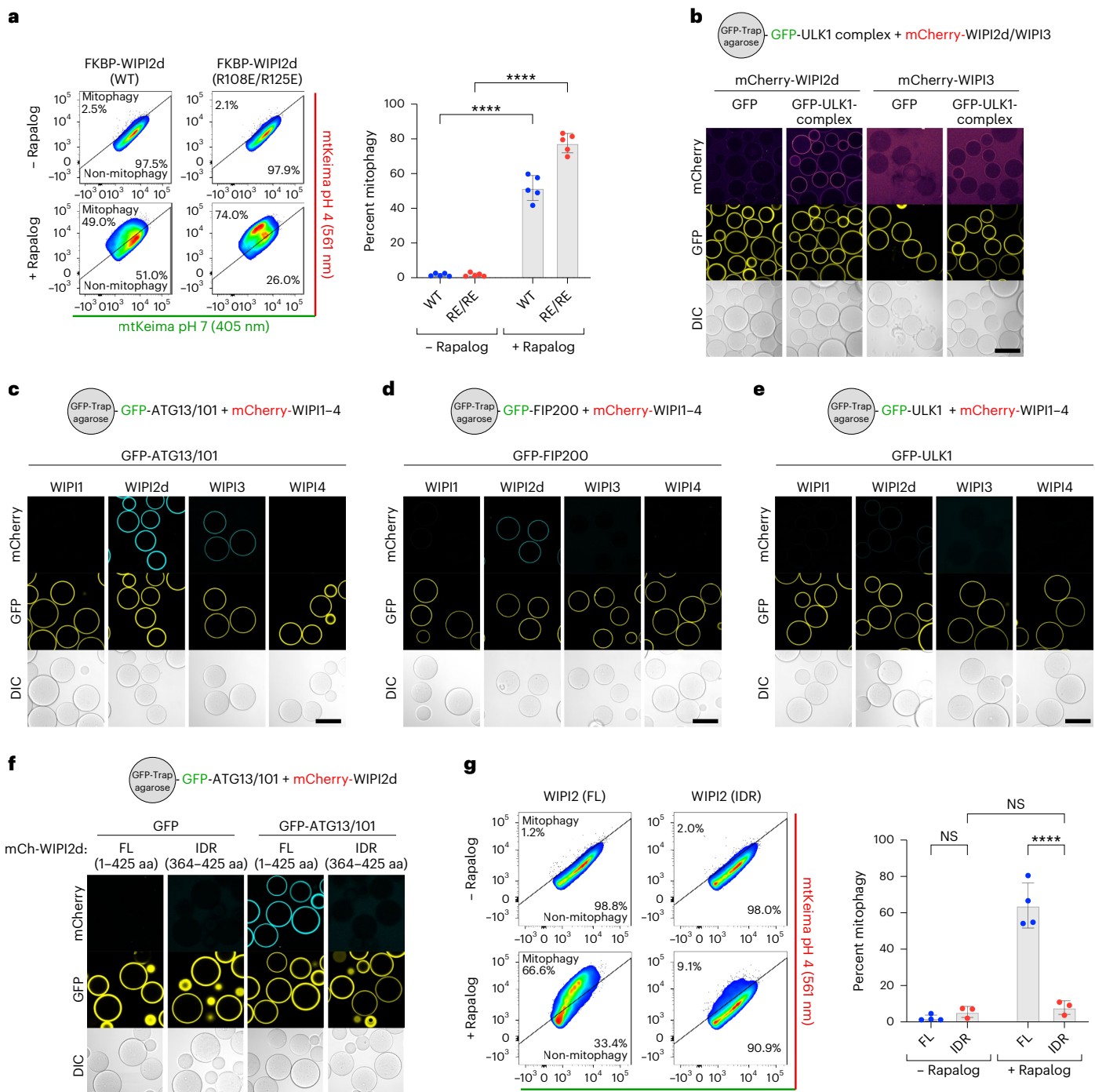

**Fig. 5 | WIPI2 and WIPI3 bind directly to the ULK1 complex. a**, Mitophagy flux measured by flow cytometry in WT HeLa cells expressing Fis1-FRB, FKBP–GFP–WIPI2 WT or ATG16L1-binding mutant R108E/R125E, and mt-mKeima, not induced or induced for 24 h by rapalog treatment. Data are presented as mean ± s.d. (*n* = 5 biologically independent experiments). Two-way ANOVA with Šídák's multiple comparisons test. **b**, Microscopy-based bead assay of agarose beads coated with GFP-tagged ULK1 complex (consisting of FIP200-GFP, ULK1, ATG13, ATG101) and incubated with mCherry-tagged WIPI proteins. **c**, As in **b**, but with GFP-tagged ATG13/101 subcomplex and incubated with mCherry-tagged WIPI proteins. **d**, As in **b**, but with GFP-tagged FIP200-coated beads and incubated with mCherry-tagged WIPI proteins. **e**, As in **b**, but with GFP-tagged

ULK1-coated beads and incubated with mCherry-tagged WIPI proteins. **f**, As in **b**, but with GFP-tagged ATG13/101-coated agarose beads incubated with mCherry-tagged full-length (FL) or IDR only (residues 364–425) WIPI2d. **g**, Mitophagy flux measured by flow cytometry in WT HeLa cells expressing Fis1-FRB, FL or IDR only (364–425 aa) FKBP–GFP–WIPI2, and mt-mKeima, not induced or induced for 24 h by rapalog treatment. Data are presented as mean ± s.d. (*n* = 4 biologically independent experiments). Two-way ANOVA with Šídák's multiple comparisons test. ****$P$ < 0.0001. NS, not significant. Results are representative of three biologically independent replicates (**b**–**f**). Scale bars, 100 µm. Source numerical data, including exact $P$ values, are available in the source data.

residues and another with 11 residues replaced by alanine. Only the 11x Ala mutant abrogated the interaction, demonstrating that an extended stretch of the ATG13 IDR interacts with WIPI2d (Fig. 6e).

We then assessed the functional relevance of the identified binding interface during BNIP3/NIX-mitophagy by measuring mitophagy flux in wild-type HeLa cells, ATG13 knockout cells and ATG13 knockout cells rescued with wild-type or mutant ATG13 (Δ190–230, Δ190–205 and Δ206–230) (Fig. 6f). DFP treatment induced mitophagy in ~20% of wild-type HeLa cells, which was completely abrogated in ATG13 knockout cells but rescued to nearly 60% with wild-type ATG13 overexpression. The ATG13 Δ190–230 mutant exhibited a significant defect, reducing mitophagy to 13%. The ATG13 Δ190–205 mutant displayed an intermediate phenotype with ~30% mitophagy, and the ATG13 Δ206–230 mutant showed a near wild-type phenotype with 57% mitophagy.

Our results demonstrate that BNIP3/NIX initiate autophagosome biogenesis by recruiting WIPI proteins, which in turn recruit the upstream ULK1 complex. WIPI2d and WIPI3 binding to the initial segment of the ATG13 IDR is critical for the formation of the WIPI–ULK1 complex during BNIP3/NIX-mitophagy.

### Flexibility in autophagy machinery assembly

Our findings reveal distinct assembly sequences during autophagosome biogenesis in the BNIP3/NIX versus PINK1/Parkin-mitophagy pathways. Specifically, in BNIP3/NIX-mitophagy, WIPI protein recruitment to mitochondria occurs upstream of the ULK1 and PI3KC3–C1 complexes, underscoring the crucial role of the WIPI–ATG13 interaction. This observation raises the question of whether this interaction is also important in other forms of selective or non-selective autophagy.

To investigate this, we first examined the role of ATG13 in basal autophagy. In ATG13 knockout cells, we observed substantial accumulation of activated SQSTM1/p62 (Fig. 7a), a pattern also seen in FIP200 knockout cells[4]. The elevated levels of heavily phosphorylated SQSTM1/p62 suggest a blockage in its basal turnover. Notably, reintroducing wild-type ATG13 or the Δ190–230 variant (which is deficient in BNIP3/NIX-mitophagy) restored SQSTM1/p62 levels, indicating that the WIPI–ATG13 interaction is not essential for basal autophagy.

We then assessed the impact of the WIPI–ATG13 interaction on starvation-induced non-selective autophagy. In ATG13 knockout cells, lipidated LC3-II levels remained unchanged following starvation plus bafilomycin A1 treatment, demonstrating a complete blockage of autophagy flux (Fig. 7b). However, this blockade was rescued by reintroducing either wild-type ATG13 or the Δ190–230 variant, suggesting that the WIPI–ATG13 complex is not critical for non-selective autophagy induction.

Next, we explored the role of the WIPI–ATG13 interaction in PINK1/Parkin-mitophagy. Unlike BNIP3/NIX-mitophagy, where ATG13 is absolutely essential, PINK1/Parkin-mitophagy was only mildly affected by ATG13 deletion. Both ATG13 knockout and ATG13 siRNA-depleted cells showed a modest reduction in mitophagy flux but did not impair PINK1/Parkin-mitophagy (Fig. 7c–e).

Our data thus show that transmembrane cargo receptors such as BNIP3/NIX can recruit the autophagy machinery in a distinct order compared to soluble cargo receptors, and use a WIPI-driven pathway instead of a FIP200-driven pathway.

### WIPI recruitment is common among transmembrane receptors

Inspired by our findings that BNIP3/NIX initiate autophagosome biogenesis by first recruiting WIPI proteins, we investigated whether other transmembrane cargo receptors could also bind and recruit WIPIs. To explore this possibility, we performed an AF3 screen to identify additional candidate autophagy receptors that might interact with WIPI2. The predictions were ranked using the ipTM score, identifying potential interactions between WIPI2 and several transmembrane autophagy receptors, including the ER-phagy receptors TEX264 and FAM134C, as well as the mitophagy receptor FKBP8 (Fig. 8a). Notably, TEX264 (ipTM 0.54) and FKBP8 (ipTM 0.58) scored above the 0.5 threshold, similar to BNIP3 (ipTM 0.66) and NIX (ipTM 0.65). However, FAM134C (ipTM 0.46) scored slightly below this cutoff. We repeated the predictions with AF2 Multimer, which also predicted interactions for TEX264 and FKBP8 but not for FAM134C. Interestingly, TEX264 and FKBP8 were predicted to bind the same pocket on WIPI2 as BNIP3/NIX (Extended Data Fig. 9a,b), suggesting a potentially conserved feature among different autophagy receptors.

We next tested these predicted interactions using recombinant proteins, focusing on TEX264, FKBP8 and FAM134C, with CCPG1 (ipTM 0.2) serving as a negative control. To achieve this, we expressed and purified the soluble domains of each receptor, substituting their transmembrane regions with GST (Extended Data Fig. 9c). Using a microscopy-based bead assay, we assessed the ability of these receptors to bind mCherry-tagged WIPI2d. TEX264 and FKBP8 demonstrated binding to WIPI2d, but FAM134C and CCPG1 did not (Fig. 8b). These results align with the previous identification of WIPI2 as the strongest hit for TEX264 in proximity labelling experiments conducted in cells undergoing ER-phagy[29]. Together, these findings suggest that WIPI-mediated autophagy initiation might represent a conserved mechanism across multiple organelles.

Next, we investigated whether TEX264 and FKBP8 can also bind FIP200 in addition to WIPI2d. We found that both TEX264 and FKBP8 could bind FIP200 (Fig. 8c,d), similar to FAM134C and CCPG1. This indicates that TEX264 and FKBP8 can recruit both FIP200 and WIPI2, whereas BNIP3/NIX exclusively recruit WIPI2/3. Notably, the binding strength for FIP200 was comparable between FAM134C, TEX264 and FKBP8, but significantly stronger for CCPG1, probably due to CCPG1's dual FIR motifs[23].

Because TEX264 and FKBP8 can bind both FIP200 and WIPI2d, we examined whether these receptors could recruit both autophagy initiation arms simultaneously. We coated agarose beads with GST-tagged TEX264 or FKBP8 and incubated the cargo receptors with the GFP-tagged FIP200 C-terminal region and mCherry-tagged WIPI2d. This revealed that both TEX264 and FKBP8 can recruit FIP200 and

---

**Fig. 6 | Biochemical characterization of the WIPI–ULK1 mitophagy initiation complex. a**, Microscopy-based bead assay of GST-tagged WIPI2d or WIPI3 incubated with mCherry-tagged ATG13/101 complex, composed of FL ATG13 (mCh-ATG13/101), HORMA domain only (mCh-HORMA; ATG13 (residues 1–191)/101) or IDR only (mCh-IDR; ATG13 residues 191–517). **b**, As in **a**, but with GFP-tagged ATG13 IDR-coated beads, either as full IDR (residues 191–517) or fragments (residues 191–230; residues 191–205; residues 206–230), incubated with mCherry-tagged WIPI2d or WIPI3. **c**, As in **a**, but with GFP-tagged ATG13 IDR-coated beads, either as full-length IDR (191–517 aa) or with variants containing deletion fragments (Δ191–230 aa), (Δ191–205 aa) or (Δ206–230 aa), and incubated with mCherry-tagged WIPI2d or WIPI3. **d**, AlphaFold predicted structure of WIPI2d (orange) and ATG13 (green) plus ATG101 (blue) with zoom-in on the interaction interface. Note that the indicated residue numbers for WIPI2 correspond to their residue number in the WIPI2d sequence (which match Y113

and R143 in WIPI2b). Structures were trimmed for visual clarity. Displayed are ATG13 (residues 1–223), ATG101 (residues 1–218) and WIPI2d (residues 1–383). **e**, As in **a**, but with GFP-tagged ATG13 IDR (residues 191–517) coated beads and incubated with mCherry-tagged WIPI2d or WIPI3. The IDR is composed of either WT, 3x Ala mutant (3A) or 11x Ala mutant (11A) (as indicated). **f**, Mitophagy flux measured by flow cytometry of WT or ATG13 KO HeLa cells, rescued (as indicated) with ATG13 WT, ATG13 lacking residues 191–230 (Δ191–230), ATG13 lacking residues 191–205 (Δ191–205) or ATG13 lacking residues 206–230 (Δ206–230), left untreated or treated with DFP for 24 h. Data are presented as mean ± s.d. (n = 3 biologically independent experiments). Two-way ANOVA with Dunnett's multiple comparisons test. ****P < 0.0001. NS, not significant. Results are representative of three independent replicates (**a**–**c**,**e**). Scale bars, 100 μm. Source numerical data, including exact P values, are available in the source data.

WIPI2d at the same time (Fig. 8e), with no indication of competitive binding or overlapping binding sites under the tested conditions. This suggests the formation of a mega-initiation complex. However, we cannot rule out the possibility of competitive or sequential binding to these receptors in cells, depending on the physiological context.

To test whether the TEX264–WIPI2 interaction is important during ER-phagy in cells, we first validated the interaction in cells by co-immunoprecipitation, using FAM134C as a negative control (Fig. 8f). We then tested if WIPI2 is essential for ER-phagy of the WIPI2-binding receptor TEX264. By inducing ER-phagy through starvation, we

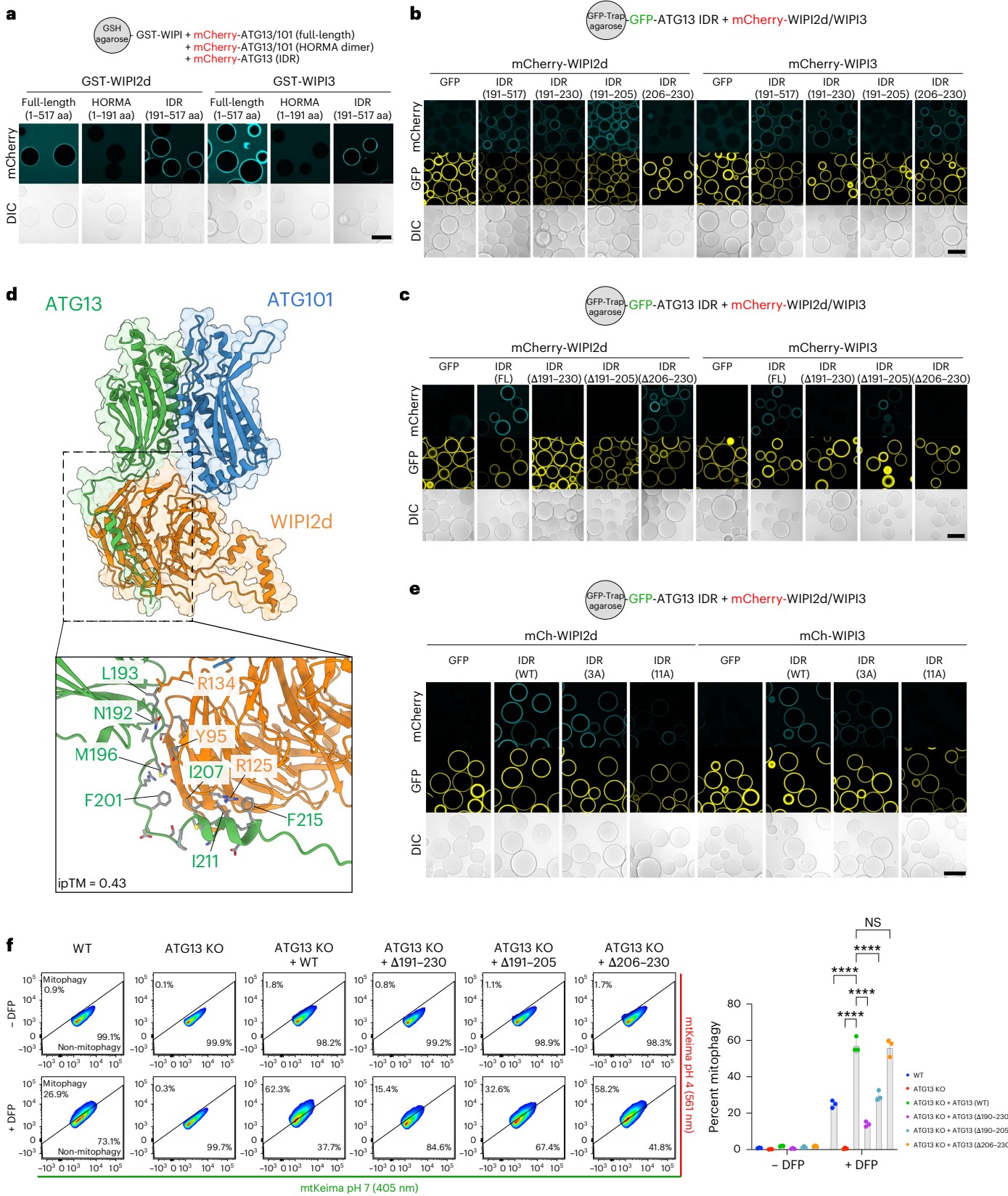

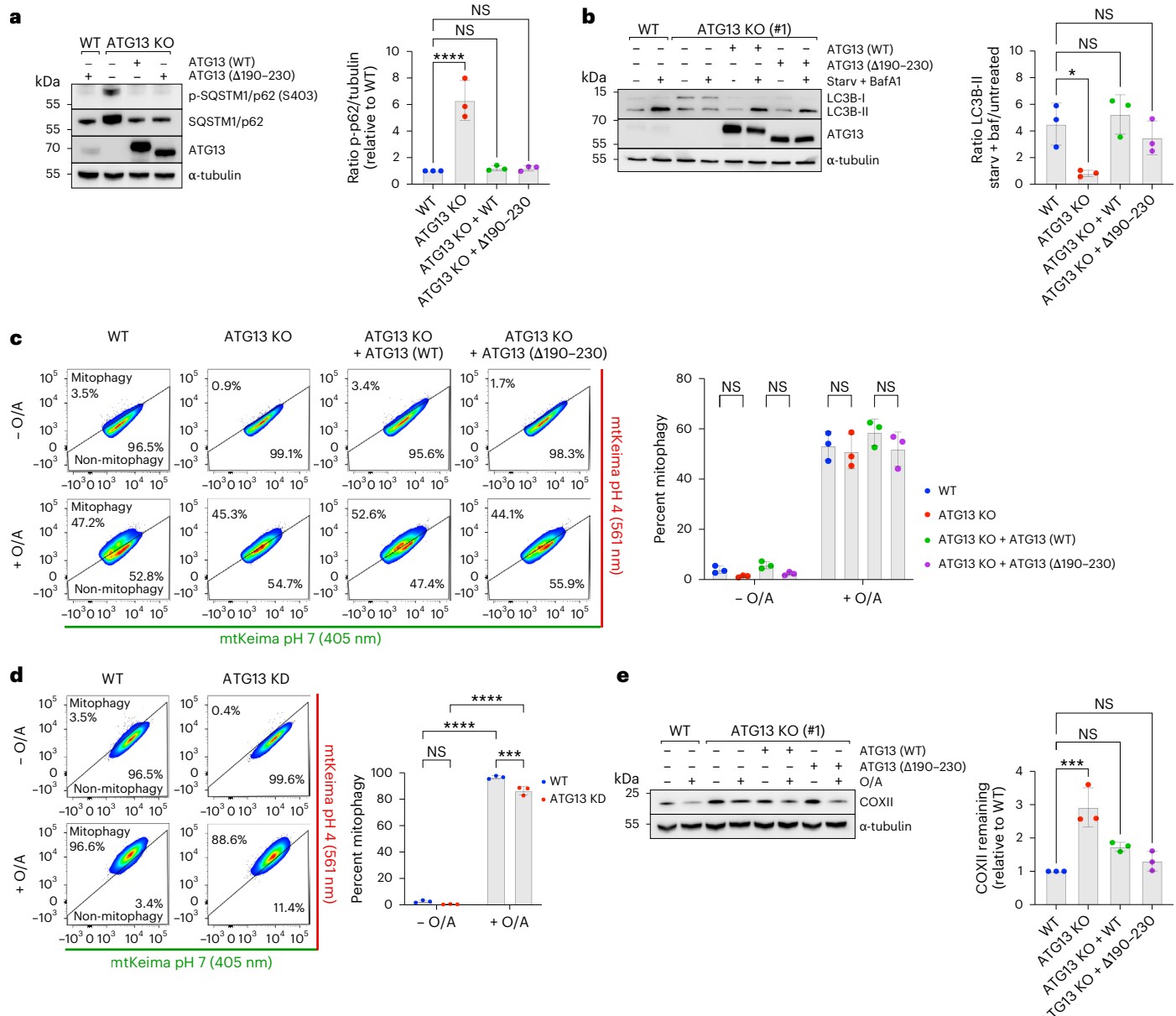

**Fig. 7 | Distinct hierarchy of assembly between WIPI-driven mitophagy and FIP200-driven mitophagy or starvation-induced autophagy.**
**a**, Immunoblotting for phosphorylated SQSTM1/p62 in WT or ATG13 KO cells (clone #1), where indicated rescued with ATG13 WT or ATG13 lacking residues 190–230 (Δ190–230). Data are presented as mean ± s.d. (n = 3 biologically independent experiments). One-way ANOVA with Dunnett's multiple comparisons test. **b**, Immunoblotting for LC3B in the same cell lines as used in **a**, but treated with 2-h Earle's balanced salt solution (EBSS) starvation medium and bafilomycin A1 (BafA1) where indicated. Data are presented as mean ± s.d. (n = 3 biologically independent experiments). Tukey's multiple comparisons test. **c**, Mitophagy flux measured by flow cytometry of WT or ATG13 KO HeLa cells, overexpressing BFP-Parkin, where indicated rescued with ATG13 WT or ATG13 lacking residues 190–230 (Δ190–230), and left untreated or treated with

O/A for 5 h. Data are presented as mean ± s.d. (n = 3 biologically independent experiments). Two-way ANOVA with Tukey's multiple comparisons test. **d**, As in **c**, but with WT HeLa cells, overexpressing BFP-Parkin, transfected with siRNAs targeting ATG13, left untreated or treated with O/A for 5 h. Data are presented as mean ± s.d. (n = 3 biologically independent experiments). Two-way ANOVA with Tukey's multiple comparisons test. **e**, Immunoblotting of COXII levels in WT or ATG13 KO HeLa cells, overexpressing BFP-Parkin, and where indicated rescued with ATG13 WT or ATG13 lacking residues 190–230 (Δ190–230), left untreated or treated with O/A for 24 h. Data are presented as mean ± s.d. (n = 3 biologically independent experiments). One-way ANOVA with Dunnett's multiple comparisons test. *P < 0.05, ***P < 0.001, ****P < 0.0001. NS, not significant. Source numerical data, including exact P values, are available in the source data.

assessed the turnover of TEX264 (WIPI2 and FIP200 binder) and CCPG1 (FIP200 binder) and observed that TEX264 was no longer turned over in WIPI2 knockout cells, but the WIPI2-independent receptor CCPG1 still was (Fig. 8g). As expected, both receptors failed to undergo ER-phagy in FIP200 knockout cells, as FIP200 appears essential in both the FIP200 and WIPI–ATG13 pathways of autophagy initiation. These cellular results confirm our in vitro interaction studies but also

reveal that TEX264 critically depends on the WIPI2 interaction during starvation-induced ER-phagy in HeLa cells.

In summary, our study reveals that selective autophagy mediated by transmembrane cargo receptors can be initiated through two distinct modes: either by first recruiting FIP200 or by recruiting WIPI proteins (Fig. 8h). Although WIPI proteins were previously considered downstream factors, our work shows that several transmembrane

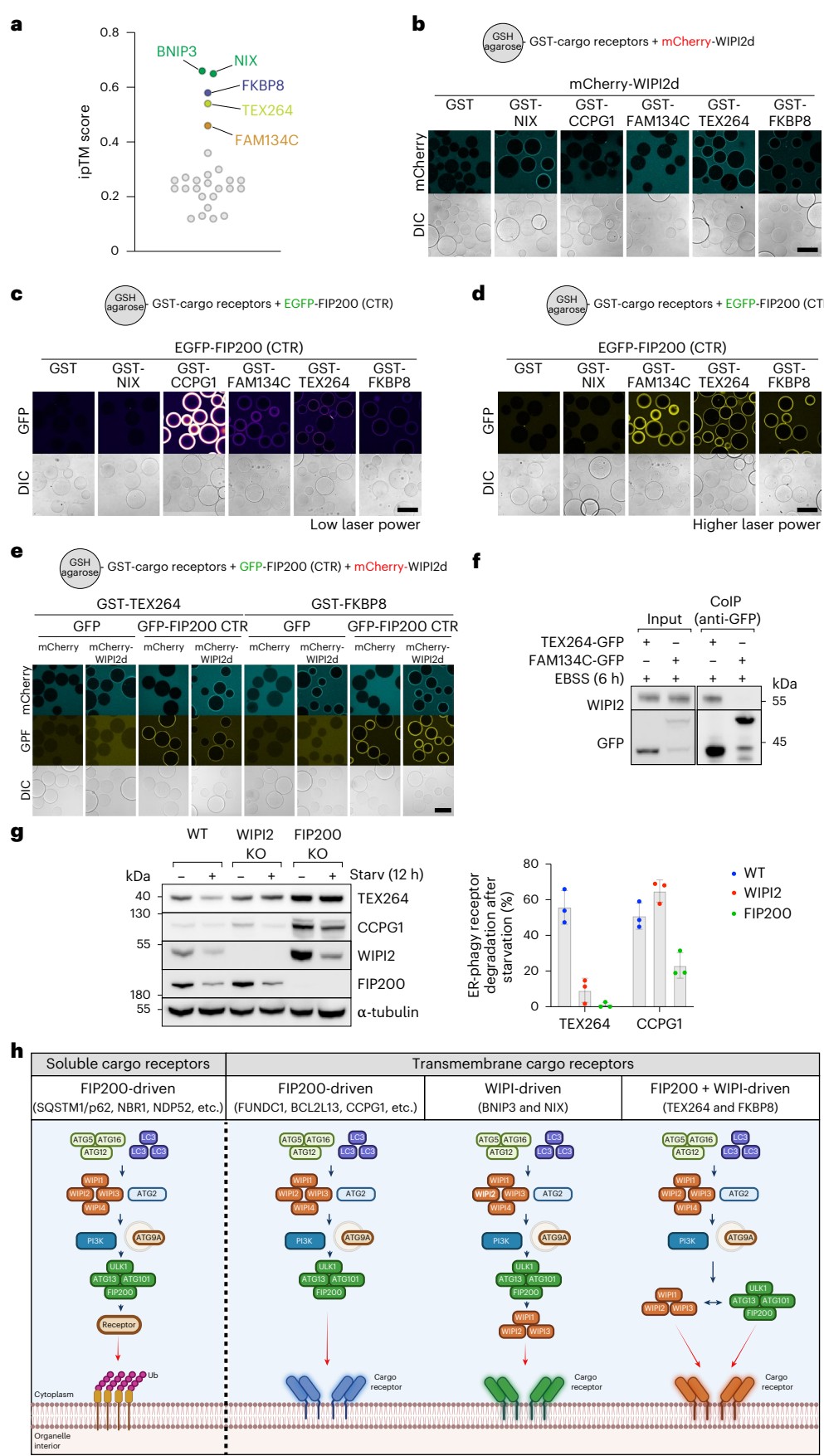

**Fig. 8 | Several transmembrane cargo receptors can bind WIPI proteins.**
**a**, AF3 screen for interaction between all known cargo receptors (soluble and transmembrane) and WIPI2. Predicted interactions are plotted for their ipTM score. **b**, Microscopy-based bead assay of GST-tagged NIX, CCPG1, FAM134C, TEX264 and FKBP8 or GST alone as a negative control and incubated with mCherry-tagged WIPI2d. **c,d**, As in **b**, but with a GFP-tagged C-terminal region of FIP200 (CTR). The laser power was either very low to visualize CCPG1-FIP200 interaction (**c**) or with higher laser power to visualize FAM134C, TEX264, FKBP8 and FIP200 interaction (**d**). In **c**, we used the Fire LUT to better visualize the difference in binding strength between the different receptors. **e**, As in **b**, but with mCherry-tagged WIPI2d and/or GFP-tagged C-terminal region of FIP200 (CTR). **f**, Co-immunoprecipitation of TEX264-GFP or FAM134C-GFP after 6-h starvation treatment with EBSS and immunoblotting for the interaction with WIPI2. **g**, Turnover analysis of different

ER-phagy receptors upon starvation treatment for 12 h with EBSS in WT, WIPI2 KO or FIP200 KO HeLa cells. Data are presented as mean ± s.d. (*n* = 3 biologically independent experiments). **h**, Schematic overview of the different selective autophagy pathways. Soluble cargo receptors are recruited to ubiquitinylated organelles and recruit the ULK1 complex through FIP200 to initiate autophagosome biogenesis. Transmembrane cargo receptors can initiate autophagosome biogenesis either by recruiting FIP200 or by recruiting WIPI proteins. The latter then recruits the ULK1 complex through interactions with ATG13. Depending on the cargo receptor, autophagosome biogenesis can be initiated through FIP200- and/or WIPI-driven mechanisms. Results are representative of three independent replicates (**b–g**). Scale bars, 100 µm. Unprocessed blots are available in the source data. Schematic generated with BioRender.

cargo receptors contain motifs enabling them to bind and recruit WIPI proteins to initiate autophagosome biogenesis. This finding highlights an unexpected flexibility in the hierarchical assembly of the autophagy machinery during autophagosome formation.

## Discussion

In this Article we uncover the mechanisms by which selective autophagy receptors can initiate selective autophagy, expanding our understanding beyond the well-characterized pathways involving soluble cargo receptors. We have delineated distinct pathways utilized by different transmembrane receptors to initiate selective autophagy.

Our findings demonstrate that various transmembrane cargo receptors, including FUNDC1, BCL2L13, CCPG1 and FAM134C, recruit the autophagy machinery through interaction with FIP200. This mechanism mirrors the way soluble cargo receptors initiate autophagosome biogenesis, underscoring the conservation of autophagy initiation processes across different receptor types. The depletion of ULK1-complex components was shown to impair mitophagy driven by FUNDC1 and BCL2L13[62,63], and co-immunoprecipitation experiments confirmed that ULK1 interacts with both receptors[62,63], highlighting the crucial role of the ULK1 complex in these processes. Moreover, the binding of these transmembrane receptors to the C-terminal domain of FIP200 further emphasizes the critical role of FIP200 in autophagosome biogenesis[1,2,35,64–69], supporting the notion that transmembrane receptors engage the autophagy machinery through conserved motifs.

By contrast, NIX and BNIP3 utilize a fundamentally different strategy to initiate mitophagy, which does not involve direct interaction with FIP200 or other upstream components of the canonical autophagy pathway. These results do not rule out the possibility that BNIP3/NIX can bind to FIP200 under different conditions than those tested here, but we were unable to establish a direct interaction between the two mitophagy receptors and FIP200. Instead, our data demonstrate that NIX and BNIP3 recruit downstream WIPI proteins to the mitochondrial surface, which in turn engage the upstream ULK1 complex via ATG13/101 subunits. This order of recruitment represents a previously unrecognized mode of autophagy initiation, highlighting an extraordinary flexibility in the assembly and activation of autophagy machinery.

The interaction of BNIP3/NIX with ATG13 via WIPI2 and WIPI3 suggests a mechanism where downstream autophagy factors can facilitate the recruitment of upstream components, thereby reversing the classical sequence of autophagy initiation events. This reverse recruitment mechanism was validated by our experiments showing that tethering WIPI proteins to the mitochondrial surface is sufficient to initiate autophagosome biogenesis, contingent upon the presence of functional ULK1 and PI3KC3–C1 complexes.

Further biochemical characterization and AF modelling provided structural insights into the interactions between WIPI proteins and the ULK1 complex. We identified specific binding interfaces within the β-propeller domains of WIPI2 and WIPI3 that interact with the ATG13/101 subcomplex. These interactions were essential for BNIP3/NIX-mitophagy, as mutations disrupting the WIPI–ULK1 complex

formation abrogated autophagic flux. That WIPI2 and WIPI3 bind neighbouring sequences on the ATG13 IDR, and that BNIP3/NIX form dimers in their active state[37], suggests that the same ATG13 molecule might interact with two WIPI molecules. This interaction could thus result in the formation of one large mitophagy initiation complex composed of BNIP3/NIX–WIPI2–WIPI3–ATG13/101–FIP200–ULK1.

The recruitment of WIPI proteins by NIX and BNIP3 and their ability to initiate mitophagy independently of TBK1, a kinase often essential in soluble cargo receptor-mediated autophagy[1,2,35,66,67], together with the critical role of ATG13 during BNIP3/NIX-mitophagy but not PINK1/Parkin-mitophagy, represents a critical distinction between the autophagy pathways initiated by soluble versus transmembrane cargo receptors. This distinction not only underscores the diversity of autophagy initiation mechanisms but also suggests that cells might employ different strategies to ensure the turnover of specific organelles under varying physiological conditions.

Importantly, the WIPI–ATG13 axis we uncover here may be widely used by transmembrane cargo receptors, as we found that another mitophagy transmembrane receptor, FKBP8, as well as the ER-phagy receptor TEX264, bind to WIPI2. Notably, these receptors also bind to FIP200, suggesting that they can activate selective autophagy through both the WIPI and FIP200 pathways.

Overall, our study advances our understanding of the molecular mechanisms underlying transmembrane receptor-mediated selective autophagy. The discovery of distinct pathways for different receptors enriches the conceptual framework of autophagy and opens new avenues for targeted therapeutic interventions in diseases characterized by dysfunctional autophagy. Future studies will be necessary to dissect further the regulatory mechanisms governing these pathways and to explore their implications in various cellular contexts and disease states.

## Online content

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

[1]Max Perutz Labs, Vienna Biocenter Campus (VBC), Vienna, Austria. [2]Department of Biochemistry and Cell Biology, Max Perutz Labs, University of Vienna, Vienna, Austria. [3]Aligning Science Across Parkinson's (ASAP) Collaborative Research Network, Chevy Chase, MD, USA. [4]California Institute for Quantitative Biosciences, University of California, Berkeley, CA, USA. [5]Graduate Group in Biophysics, University of California, Berkeley, CA, USA. [6]Department of Theoretical Biophysics, Max Planck Institute of Biophysics, Frankfurt am Main, Germany. [7]Walter and Eliza Hall Institute of Medical Research, Parkville, Victoria, Australia. [8]Department of Biochemistry and Molecular Biology, Biomedicine Discovery Institute, Monash University, Melbourne, Australia. [9]Department of Medical Biology, University of Melbourne, Melbourne, Victoria, Australia. [10]Department of Molecular and Cell Biology, University of California, Berkeley, CA, USA. [11]Institute of Biophysics, Goethe University Frankfurt, Frankfurt am Main, Germany. [12]Helen Wills Neuroscience Institute, University of California, Berkeley, CA, USA. ✉e-mail: elias.adriaenssens@univie.ac.at; sascha.martens@univie.ac.at

## Methods

### Key resources table

A key resources table is provided in Supplementary Table 1 and provides a detailed overview of catalogue and research resource identifier (RRID) numbers. Further details can be found in the Nature Portfolio Reporting Summary linked to this Article.

### Reagents

The following chemicals were used in this study: rapalog A/C hetero-dimerizer (635057, Takara), bafilomycin A1 (sc-201550, Santa Cruz Biotech), TBK1 inhibitor GSK8612 (S8872, Selleck Chemicals), ULK1/2 inhibitor (MRT68921, BLDpharm), Vps34-IN1 inhibitor (APE-B6179, ApexBio), CK2 kinase inhibitor (CX4945, Selleck Chemicals), deferiprone (DFP; 379409, Sigma-Aldrich), oligomycin A (A5588, ApexBio), antimycin A1 (A8674, Sigma-Aldrich), Q-VD-OPh (A1901, ApexBio) and dimethylsulfoxide (DMSO; D2438, Sigma).

### siRNAs

The following siRNAs were used in this study: FIP200 (SMARTPOOL; LQ-021117-00-0002), ATG13 (SMARTPOOL; L-020765-01-0005), ULK1 (SMARTPOOL; L-005049-00-0005), PPTC7 (SMARTPOOL; L-017008-00-0005) and non-targeting control pool (D-001810-10).

### Plasmid construction

The sequences of all complementary DNAs (cDNAs) were obtained by amplifying from existing plasmids, HAP1 or HeLa cDNA, or gene synthesis (GenScript). For insect cell expressions, the sequences were codon-optimized and gene-synthesized (GenScript). Plasmids were generated by Gibson cloning. Plasmids were verified by Sanger or whole plasmid sequencing (Plasmidsaurus). A detailed cloning protocol is available in ref. 70.

### Cell lines

HeLa (CVCL_0058) and HEK293T (CVCL_0063) cells were acquired from the American Type Culture Collection (ATCC). Cells were grown in Dulbecco's modified Eagle medium (DMEM, Thermo Fisher) supplemented with 10% (vol/vol) fetal bovine serum (FBS; Sigma-Aldrich), 25 mM HEPES (Thermo Fisher), 1% (vol/vol) non-essential amino acids (Thermo Fisher) and 1% (vol/vol) penicillin–streptomycin (Thermo Fisher). HAP1 cells were cultured in Iscove's modified Dulbecco's medium (Thermo Fisher) supplemented with 10% (vol/vol) FBS (Thermo Fisher) and 1% (vol/vol) penicillin–streptomycin (Thermo Fisher). Cell lines were cultured at 37 °C in a humidified 5% $CO_2$ atmosphere. All cell lines were tested regularly for mycoplasma contaminations. A detailed protocol is available in ref. 71.

### Generation of CRISPR/Cas9 knockout cells

Knockout cell lines were generated using CRISPR/Cas9. CRISPick (SCR_025148) was used to identify single-guide RNAs (sgRNAs) targeting all common splicing variants and cloned into pSpCas9(BB)-2A-GFP vector (Addgene_48138). Single clones were expanded and verified for knockout of the target gene. For NIX and BNIP3 double-knockout cells, WIPI2, FIP200 or ATG13 single-knockout cells, we transfected sgRNAs for the respective target genes into naïve HeLa cells (CVCL_0058) to obtain BNIP3/NIX double-knockout cells #6 (CVCL_E1HA) and #10 (CVCL_E1HB), WIPI2 knockout cells #7 (CVCL_E6FD), FIP200 knockout cells #39 (CVCL_D1LA) or ATG13 knockout cells #1 (CVCL_E1HE). Detailed protocols are available in refs. 72,73.

### Generation of stable cell lines

Stable cell lines were generated using lentiviral or retroviral expression systems. The following retroviral vectors were used: pCHAC-mito-mKeima (Addgene_72342), VSV-G (gift from R. Youle) and Gag-Pol (gift from R. Youle). The following lentiviral vectors were used: pHAGE-FKBP-GFP-WIPI1 (Addgene_223767), pHAGE-FKBP-GFP-WIPI2 (Addgene_223757), pHAGE-FKBP-GFP-WIPI3 (Addgene_223768), pHAGE-FKBP-GFP-WIPI4

(Addgene_223769), pHAGE-FKBP-GFP-WIPI2 R108E/R125E (Addgene_223770), pHAGE-FKBP-GFP-WIPI2 IDR (364–425 aa) (Addgene_223758), pHAGE-mt-mKeima-P2A-FRB-Fis1 (Addgene_135295), pHAGE-TEX264-GFP (Addgene_201925), pHAGE-FAM134C-GFP (Addgene_201927), pGenLenti V5-BNIP3 (Addgene_223732), pGen-Lenti V5-NIX (Addgene_223731), pGenLenti V5-NIX W36A/L39A (ΔLIR) (Addgene_223788), pGenLenti V5-NIX E72A/L75A/D77A/E81A (4A mutant; ΔWIPI2) (Addgene_223789), pGenLenti ATG13 (WT) (Addgene_223771), pGenLenti ATG13 (Δ191–230) (Addgene_223772), pGenLenti ATG13 (Δ191–205) (Addgene_223773), pGenLenti ATG13 (Δ206–230) (Addgene_223774), VSV-G and Gag-Pol (gift from the Versteeg laboratory). Detailed protocols are available in refs. 74,75.

### Mitophagy experiments

To induce BNIP3/NIX-mitophagy, cells were treated for 24 h with 1 mM DFP (Sigma-Aldrich). A detailed protocol is available in ref. 76. To induce PINK1/Parkin-mitophagy, cells were treated with 10 μM oligomycin (ApexBio) and 4 μM antimycin A (Sigma-Aldrich). Where cells were treated for more than 8 h, we also added 10 μM Q-VD-OPh (ApexBio) to suppress apoptosis. A detailed protocol is available in ref. 77.

### Non-selective autophagy experiments

To induce non-selective bulk autophagy, cells were starved by culturing them in Earle's balanced salt medium (Sigma-Aldrich). A detailed protocol is available in ref. 78.

### ER-phagy experiments

To induce ER-phagy, cells were starved for 12 h by culturing them in Earle's balanced salt medium (Sigma-Aldrich). Cells were collected and analysed by sodium dodecyl sulfate polyacrylamide gel electrophoresis (SDS–PAGE) and western blot analysis. A detailed protocol is available in ref. 79.

### Rapalog-induced chemical dimerization experiments

The chemical-induced dimerization experiments were performed using the FRB-Fis1 and FKBP fused to our gene of interest system. Cells were treated with 500 nM rapalog A/C hetero-dimerizer rapalog (Takara) for 24 h and analysed by flow cytometry, western blot or mass spectrometry. A detailed protocol is available in ref. 80.

### Flow cytometry

For mitophagy experiments, 700,000 cells were seeded in six-well plates and induced by treating the cells with DFP or oligomycin A/antimycin A1 (O/A), as described above. Cells were collected by trypsinization and resuspended in medium, filtered through 35-μm cell-strainer caps (Falcon), and analysed by an LSR Fortessa Cell Analyser (BD Biosciences). Lysosomal mt-mKeima was measured using dual-excitation ratiometric pH measurements with 405-nm (pH 7) and 561-nm (pH 4) lasers with 710/50-nm and 610/20-nm detection filters, respectively. Additional channels used for fluorescence compensation used GFP. Single fluorescence vector-expressing cells were prepared to adjust the photomultiplier tube voltages to ensure the signal was within detection limits, and to calculate the compensation matrix in BD FACSDiva Software. Depending on the experiment, we gated for GFP-positive and/or mtKeima-positive cells with the appropriate compensation. For each sample, 10,000 mtKeima-positive events were collected, and the data were analysed in FlowJo (SCR_008520). Our protocol was based on a previously described protocol[81].

For rapalog-induced mitophagy experiments, cells were induced for 24 h with 500 nM rapalog A/C hetero-dimerizer (Takara). A detailed protocol is available in ref. 80.

### SDS–PAGE and western blot analysis

For SDS–PAGE and western blot analysis, we used our common laboratory protocol (available in ref. 82). Samples were analysed

with 4–12% SDS–PAGE gels (Thermo Fisher), with the exception of the LC3B blots, which were analysed with 16% Tris-glycine gels (Thermo Fisher). The primary antibodies used in this study were as follows: anti-α-tubulin (1:5,000, Abcam cat. no. ab7291, AB_2241126), anti-ATG13 (Cell Signaling Technology cat. no. 13468, AB_2797419), anti-beclin1 (1:1,000, Cell Signaling Technology cat. no. 3738, AB_490837), anti-phospho-beclin1 Ser30 (1:1,000, Cell Signaling Technology cat. no. 54101, AB_3102019), anti-BNIP3 (1:1,000, Cell Signaling Technology cat. no. 44060, AB_2799259), anti-CCPG1 (1:1,000, Cell Signaling Technology cat. no. 80158, AB_2935809), anti-COXII (1:1,000, Abcam cat. no. ab110258, AB_10887758) or (1:1,000, Cell Signaling Technology cat. no. 31219, AB_2936222), anti-4EBP1 (1:1,000, Proteintech cat. no. 60246-1-Ig, AB_2881368), anti-FIP200 (1:1,000, Cell Signaling Technology cat. no. 12436, AB_2797913), anti-GFP (1:1,000, Millipore cat. no. MABC1689, AB_3675504), anti-penta-His (1:1,000, Qiagen cat. no. 34660, AB_2619735), anti-LC3B (1:500, Nanotools cat. no. 0260-100/LC3-2G6, AB_2943418), anti-NIX/BNIP3L (1:1,000, Cell Signaling Technology cat. no. 12396, AB_2688036), anti-OPTN (1:500, Sigma-Aldrich cat. no. HPA003279, AB_1079527), anti-phospho-OPTN Ser177 (1:1,000, Cell Signaling Technology cat. no. 57548, AB_2799529), anti-p62/SQSTM1 (1:1,000, Abnova cat. no. H00008878-M01, AB_437085), anti-phospho-p62/SQSTM1 Ser403 (1:1,000, Cell Signaling Technology cat. no. 39786, AB_2799162), anti-TEX264 (1:1,000, Sigma-Aldrich cat. no. HPA017739, AB_1857910), anti-ULK1 (1:1,000, Cell Signaling Technology cat. no. 8054, AB_11178668), anti-V5 (1:1,000, Thermo Fisher Scientific cat. no. R960-25, AB_2556564) and anti-WIPI2 (1:1,000, Bio-Rad cat. no. MCA5780GA, AB_10845951). The secondary antibodies used in this study were horseradish peroxidase (HRP)-conjugated polyclonal goat anti-mouse (Jackson ImmunoResearch Labs cat. no. 115-035-003, AB_10015289) and HRP-conjugated polyclonal goat anti-rabbit (Jackson ImmunoResearch Labs cat. no. 111-035-003, AB_2313567).

### Immunofluorescence and confocal microscopy

Cells were seeded on gelatin-coated glass coverslips (12 mm #1.5), treated with rapalog for the indicated time, and immunostained according to the protocol in ref. 83. For mitophagy experiments using DFP, cells were transiently transfected with PPTC7 siRNA to boost NIX/BNIP3 mitophagy. To this end, 400,000 cells per well were seeded in a six-well plate with 10 nM PPTC7 siRNA using Lipofectamine RNAiMAX. After 24 h, cells were transferred to 24-well plates and seeded on gelatin-coated coverslips. From the six-well plate to the 24-well plate, cells were diluted 1:3 to achieve optimal cell density on the coverslips. After inducing NIX/BNIP3 mitophagy for 24 h with 1 mM DFP, cells were prepared for immunofluorescence as described above, with exception of the fixation, which was done with 100% methanol for 20 min at −20 °C followed by three washes in phosphate buffered saline (PBS) and blocking with 5% (vol/vol) bovine serum albumin (no permeabilization step). The rest of the protocol is described in ref. 84. Figure 4a and Extended Data Fig. 6 were obtained with a Zeiss LSM700 laser scanning confocal microscope with Plan-Apochromat ×40/1.30 oil differential interference contrast (DIC), working distance (WD) 0.21-mm objective. Figure 4j was acquired with a Zeiss LSM900 microscope equipped with an Airyscan 2 module and a Plan-Apochromat ×63/1.4 oil DIC, WD 0.19-mm objective. Z-stacks were taken with 0.14-μm step sizes. Images of Fig. 4j were acquired and processed with a two-dimensional (2D) Airyscan processing plug-in in Zen Blue software (Zeiss). For Extended Data Fig. 6, images were deconvolved with Huygens Professional 24.04 (Scientific Volume Imaging). Deconvolution was performed with confocal settings, a total of 40 iterations, and the signal-to-noise ratio parameter was set to 20. Output was generated as 32-bit ICS2 files for further processing with Fiji ImageJ. The primary antibodies used in this study were anti-ATG13 (1:200, Cell Signaling Technology, cat. no. 13468; AB_2797419), anti-WIPI2 (1:100, Abcam cat. no. ab105459, AB_10860881) and anti-HSP60 (1:800, Abcam cat. no. ab46798, AB_881444). The secondary antibodies used in this

study were AlexaFluor-488 goat anti-mouse immunoglobulin G (IgG) (H + L) (1:500, Thermo Fisher Scientific cat. no. A-11001, AB_2534069) and AlexaFluor-546 goat anti-rabbit IgG (H + L) (1:500, Thermo Fisher, cat. no. A-11035; AB_2534093).

### Purification ATG9A-vesicles

HAP1 cells were CRISPR-edited to introduce a C-terminal GFP-TEV-Flag tag into the endogenous locus of ATG9A (CVCL_E2TR). For the isolation of native ATG9A-vesicles, cells were resuspended in vesicle isolation buffer (20 mM HEPES pH 7.5, 150 mM NaCl, 250 mM sucrose, 1× cOmplete ethylenediaminetetraacetic acid (EDTA)-free protease inhibitors (Roche), 20 mM β-glycerophosphate, 1 mM sodium orthovanadate, 1 mM NaF and 1 mM EDTA pH 8.0) and lysed by passing the suspension through a 26-G needle 30 times, chilling on ice for 10 min, followed by another 30 passes through the needle. Cell debris and nuclei were separated by centrifugation and the supernatant was incubated with pre-equilibrated FLAG beads. After overnight incubation at 4 °C on a roller, beads were pelleted by centrifugation, and the unbound supernatant was removed. After washing the beads, ATG9A-vesicles were eluted from the beads using FLAG peptide for 3 h at 4 °C while rolling. The supernatant was collected and used for experiments. A detailed protocol is available in ref. 85.

### Protein expression and purification from *E. coli*

Plasmids were transformed in *E. coli* Rosetta pLysS cells (Novagen) and grown in 2× tryptone yeast extract (TY) medium at 37 °C until reaching an optical density at 600 nm (OD$_{600}$) of 0.4 and then continued at 18 °C. Once the cells reached an OD$_{600}$ of 0.8, protein expression was induced with 100 μM isopropyl β-D-1-thiogalactopyranoside for 16 h at 18 °C. Cells were collected by centrifugation and resuspended in lysis buffer.

To purify NIX-GST, the cytosol-exposed domain of NIX (1–182 aa) was fused to a C-terminal GST-tag through cloning into a pET-DUET1 vector (Addgene_223733). Point mutants were introduced by in vitro mutagenesis to generate NIX W36A/L39A (ΔLIR) (Addgene_223738) and NIX E72A/L75A/D77A/E81A (4A; ΔWIPI2) (Addgene_223753). Proteins were expressed and purified according to the protocol in ref. 86.

To purify FUNDC1-GST, the cytosol-exposed domain of FUNDC1 (1–50 aa) was fused to a C-terminal GST-tag through cloning into a pET-DUET1 vector (Addgene_223734). Point mutants were introduced by in vitro mutagenesis to generate FUNDC1 Y18A/L21A (ΔLIR) (Addgene_223739). Proteins were expressed and purified according to the protocol in ref. 87.

To purify BCL2L13-GST, the cytosol-exposed domain of BCL2L13 (1–465 aa) was fused to a C-terminal GST-tag through cloning into a pET-DUET1 vector (Addgene_223744). Point mutants were introduced by in vitro mutagenesis to generate BCL2L13 W276A/I279A (ΔLIR1) (Addgene_223749), BCL2L13 Y213A/I216A/W276A/I279A (ΔLIR1 + 2) (Addgene_223752), BCL2L13 I224A/L227A/W276A/I279A (ΔLIR1 + 3) (Addgene_223754), BCL2L13 W276A/I279A/I307A/V310A (ΔLIR1 + 4) (Addgene_223755) and BCL2L13 I224A/L227A/W276A/I279A/I307A/V310A (ΔLIR1 + 3 + 4) (Addgene_223756). Proteins were expressed and purified according to the protocol in ref. 88.

To purify GFP-tagged NIX-GFP (Addgene_223736), NIX(W36A/L39A)-GFP (ΔLIR) (Addgene_223748), BCL2L13-GFP (Addgene_223745), BCL2L13(W276A/I279A)-GFP (ΔLIR1) (Addgene_223746), BCL2L13(Y213A/I216A)-GFP (ΔLIR2) (Addgene_223783), BCL2L13(I224A/L227A)-GFP (ΔLIR3) (Addgene_223775), BCL2L13(I307A/V310A)-GFP (ΔLIR4) (Addgene_223776), BCL2L13(Y213A/I216A/W276A/I279A)-GFP (ΔLIR1 + 2) (Addgene_223782), BCL2L13(I224A/L227A/W276A/I279A)-GFP (ΔLIR1 + 3) (Addgene_223780), BCL2L13(W276A/I279A/I307A/V310A)-GFP (ΔLIR1 + 4) (Addgene_223781), BCL2L13(I224A/L227A/W276A/I279A/I307A/V310A)-GFP (ΔLIR1 + 3 + 4) (Addgene_223784), FUNDC1-GFP (Addgene_223737) and FUNDC1(Y18A/L21A)-GFP (ΔLIR) (Addgene_223750), proteins were expressed and purified according to the protocols in refs. 89–91.

To purify GST-TEX264, the cytosol-exposed domain of TEX264 (28–313 aa) fused to an N-terminal GST-tag was gene-synthesized by GenScript and cloned into a pGEX-4T1 vector (Addgene_227714). Proteins were expressed and purified according to the protocol in ref. 92.

To purify GST-FAM134C, the cytosol-exposed domain of FAM134C (250–466 aa) fused to an N-terminal GST-tag was gene-synthesized by GenScript and cloned into a pGEX-4T1 vector (Addgene_227715). Proteins were expressed and purified according to the protocol in ref. 93.

To purify CCPG1-GST, the cytosol-exposed domain of CCPG1 (1–212 aa) fused to a C-terminal GST-tag was gene-synthesized by Gen-Script and cloned into a pET-DUET1 vector (Addgene_227713). Proteins were expressed and purified according to the protocol at https://doi.org/10.17504/protocols.io.e6nvw14dzlmk/v1 (ref. 94).

To purify FKBP8-GST, the cytosol-exposed domain of FKBP8 (1–391 aa) fused to a C-terminal GST-tag was gene-synthesized by Gen-Script and cloned into a pET-DUET1 vector (Addgene_227712). Proteins were expressed and purified according to the protocol in ref. 95.

To purify the GFP-FIP200 C-terminal region (CTR), as described previously[4], the C-terminal domain of FIP200 (1,429–1,591 aa) was fused to an N-terminal 6xHis-TEV-GFP-tag by cloning into a pET-DUET1 vector (Addgene_223724). Proteins were expressed and purified according to the protocol in ref. 96.

To purify Lambda protein phosphatase (λ PPase), the protein phosphatase was fused to an N-terminal 6xHis-tag by cloning into a pET-DUET1 vector (Addgene_223747). Proteins were expressed and purified according to the protocol in ref. 97.

To purify mCherry–WIPI2d and mCherry-WIPI3, as described previously for WIPI2d[98], the coding sequence of WIPI2d or WIPI3 was fused to an N-terminal 6xHis-TEV-mCherry-tag by cloning into a pET-DUET1 vector (Addgene_223725 and Addgene_223763). Proteins were expressed and purified according to the protocol in ref. 99.

To purify mCherry–WIPI2d K87A/K88A (Addgene_223751) or mCherry–WIPI2d IDR (364–425 aa) (Addgene_223790), the same expression and purification methods were used as described above for full-length mCherry–WIPI2d, with the exception that for the mCherry–WIPI2d IDR we used the S75 Increase 10/300 column. An adapted protocol is available in ref. 100.

To purify GFP-tagged or mCherry-tagged ATG13 IDR, the coding sequence for ATG13 (191–517 aa) or ATG13 (230–517 aa) was fused to an N-terminal 6xHis-TEV-mCherry-tag by cloning into a pET-DUET1 vector (Addgene_223762) or by inserting the coding sequence for ATG13 (191–517 aa), (205–517 aa), (231–517 aa), (191–205,231–517 aa), (191–230 aa), (191–205 aa) or (206–230 aa) into a GST-TEV-EGFP-insert by cloning into a pGEX-4T1 vector (Addgene_223760, Addgene_223786, Addgene_223785, Addgene_223787, Addgene_223792, Addgene_223791, Addgene_223793). Mutants 3A (M196A/S197A/R199A; Addgene_223761) and 11A (M196A/S197A/R199A/G202A/T204A/P205A/I207A/M208A/I210A/D213A/H214A; Addgene_223779) were also expressed according to the protocol below. Proteins were expressed and purified according to the protocols in refs. 101,102.

To purify GST-LC3A, GST-LC3B, GST-LC3C, GST-GBRP, GST-GBRPL1 and GST-GBRPL2, as previously described[103], we inserted human LC3/GBRP cDNA in a pGEX-4T1 vector (Addgene_223726, Addgene_216836, Addgene_223727, Addgene_223728, Addgene_223729, Addgene_223730). The last five amino acids of LC3/GBRP were deleted to mimic the cleavage by ATG4. Proteins were expressed and purified according to the protocol in ref. 104.

To purify mCherry-tagged OPTN, we cloned human OPTN cDNA in a pETDuet-1 vector with an N-terminal 6xHis-tag followed by a TEV cleavage site (Addgene_190191). Proteins were expressed and purified according to the protocol in ref. 105.

The negative controls EGFP, mCherry and GST were purified as previously described[4,106]. Plasmids are available from Addgene (Addgene_227710, Addgene_223723).

## Protein expression and purification from insect cells

To generate bacmid DNA, we used the Bac-to-Bac system by transfecting our plasmids into DH10EMBacY cells for amplification. After verifying the bacmid DNA for insertion of the transgene by polymerase chain reaction (PCR), bacmid DNA was purified and transfected into Sf9 insect cells (Thermo Fisher; CVCL_0549). To this end, 2,500 ng of plasmid DNA was mixed with FuGene transfection reagent (Promega) and used to transfect one million Sf9 cells seeded in a six-well plate. About seven days after transfection, the V0 virus was harvested and used to infect 40 ml of one million cells per millilitre of Sf9 cells. Upon decreased viability and confirmation of yellow fluorescence, supernatants were collected after centrifugation and stored as V1 virus. For protein expressions, 1 l of Sf9 cells at one million cells per millilitre, were infected with 1 ml of V1 virus. When the viability of the cells decreased to 90–95%, the cells were collected by centrifugation. Cell pellets were washed with PBS and flash-frozen in liquid nitrogen.

To purify BNIP3-GST, we purchased the gene-synthesized codon-optimized cytosol-exposed domain of BNIP3 (1–158 aa) fused to a C-terminal GST-tag in a pFastBac-Dual vector from GenScript (Addgene_223764). Point mutants were introduced by in vitro mutagenesis to generate BNIP3 W18A/L21A (ΔLIR) (Addgene_223778) and BNIP3 E44A/L47A/D49A/A50K/Q51A (5A; ΔWIPI2) (Addgene_223777). The constructs were used to generate bacmid DNA and subsequent expression in Sf9 cells. Proteins were expressed and purified according to the protocol in ref. 107.

To purify BNIP3-GFP (Addgene_223765) and BNIP3(W18A/L21A)-GFP (ΔLIR) (Addgene_223766), we purchased gene-synthesized codon-optimized vectors from GenScript. Proteins were expressed and purified according to the protocol in ref. 108.

To purify FIP200-GFP from insect cells, we purchased gene-synthesized codon-optimized GST-3C-FIP200-EGFP in a pGB-02-03 vector from GenScript (Addgene_187832). Proteins were expressed and purified according to the protocol in ref. 109.

To purify TBK1, we purchased gene-synthesized codon-optimized GST-TEV-TBK1 in a pFastBac-Dual vector from GenScript (Addgene_208875, Addgene_187830, Addgene_198033) for expression in insect cells. Proteins were expressed and purified according to the protocol in ref. 110.

To purify Src (WT and Y530F), we purchased gene-synthesized codon-optimized GST-TEV-Src in a pFastBac-Dual vector from GenScript (Addgene_223742, Addgene_223743) for expression in insect cells. Proteins were expressed and purified according to the protocol in ref. 111.

To purify the CK2 kinase complex, we subcloned GST-TEV-CK2α together with CK2β in a pFastBac-Dual vector (Addgene_223740) and GST-TEV-CK2α′ together with CK2β in a pFastBac-Dual vector (Addgene_223741) for co-expression of the CK2α/CK2α′/CK2β complex in insect cells. Proteins were expressed and purified according to the protocol in ref. 112.

To purify mCherry-tagged PI3KC3–C1 complex, as published before[98], the codon-optimized genes were purchased from GenScript and cloned by the Vienna BioCenter Core Facilities (VBCF) Protech Facility as GST-3C-mCherry-ATG14/VPS34/VPS15/BECN1 in a pGB-dest vector (Addgene_187936). Proteins were expressed and purified according to the protocol in ref. 113.

## Protein expression and purification from HEK293F cells

Proteins were expressed in FreeStyle HEK293F cells, grown at 37 °C in FreeStyle 293 expression medium (Thermo Fisher). The day before transfection, cells were seeded at a density of $0.7 \times 10^6$ cells ml$^{-1}$. On the day of transfection, a 400-ml culture was transfected with 400 μg of the MAXI-prep DNA, diluted in 13 ml of Opti-MEM I reduced serum medium (Thermo Fisher) and 800 μg of polyethylenimine (PEI 25K, Polysciences), also diluted in 13 ml of Opti-MEM media. One day post transfection, the culture was supplemented with 100 ml of EXCELL 293 serum-free medium (Sigma-Aldrich). Another 24 h later, cells were

collected by centrifugation at 270*g* for 20 min. The pellet was washed with PBS to remove medium, then flash-frozen in liquid nitrogen. Pellets were stored at −80 °C.

To purify GST-WIPI1/GST-WIPI2/GST-WIPI3/GST-WIPI4, we expressed the GST-tagged WIPI1/2d/3/4 from a pCAG backbone encoding GST-TEV-WIPI1/2/3/4 (Addgene_223798, Addgene_223799, Addgene_223800, Addgene_223800). Proteins were expressed and purified according to the protocol in ref. 114.

To purify the mCherry-tagged or GFP-tagged ATG13/101 subcomplex, we expressed mCherry-tagged ATG13 from a pCAG backbone (Addgene_223735) together with GST-TEV-ATG101 (Addgene_171414) or GST-TEV-GFP-tagged ATG13 (Addgene_223797) together with ATG101 (Addgene_223796). Proteins were expressed and purified according to the protocol in ref. 115.

To purify mCherry-ATG13/101 HORMA dimer, we expressed mCherry-tagged ATG13 (1–191 aa) from a pCAG backbone (Addgene_223759) together with GST-TEV-ATG101 (Addgene_171414). The same expression and purification methods were used as described above for full-length mCherry-ATG13/101. A detailed protocol is available in ref. 116.

To purify GFP-tagged ULK1-complex, as described previously[39], we co-expressed GST-TEV-FIP200-MBP/EGFP-ATG13/ATG101 from a pCAG backbones (Addgene_171410, Addgene_171413, Addgene_189590) in parallel to MBP-Strep-Strep-Flag-TEV-ULK1 (Addgene_171416). The subcomplex FIP200/EGFP-ATG13/ATG101 was transfected and expressed separately from the ULK1 subunit. Proteins were expressed and purified according to the protocol in ref. 117.

To purify the MBP-ULK1 from HEK293F cells, we expressed the ULK1 kinase from a pCAG backbone encoding MBP-TSF-TEV-ULK1 (Addgene_171416). Proteins were expressed and purified according to the protocol in ref. 117.

To purify the GFP-ULK1 from HEK293F cells, we expressed the ULK1 kinase from a pCAG backbone encoding MBP-OSF-TEV-ULK1 (Addgene_239015). Proteins were expressed and purified according to the following protocol with the exception of overnight TEV cleavage instead of MBP elution[117].

### Microscopy-based bead assay

Glutathione Sepharose 4B beads (GE Healthcare) were used to bind GST-tagged bait proteins, GFP-trap agarose beads (ProteinTech) were used to bind GFP-tagged bait proteins, and RFP-trap agarose beads (ProteinTech) were used to bind mCherry-tagged bait proteins. For preparation of beads and baits in 384-well plates, a detailed protocol is available in ref. 118.

### In vitro kinase assays

To verify the activity of the kinases TBK1 and MBP-ULK1, we mixed the kinases with mCherry-tagged OPTN or PI3K-complex (composed of VPS15, VPS34, ATG14 and Beclin1) for the indicated time. The kinases were used at 50 nM and mixed with 200 nM OPTN and 130 nM PI3K complex. A detailed protocol is available in ref. 119.

To verify the activity of kinases Src and CK2, 45 µl of mixes containing either only kinase assay buffer (25 mM Tris-HCl pH 7.4, 150 mM NaCl, 1 mM dithiothreitol (DTT) and 2 mM $MgCl_2$), kinase buffer and substrate (0.5 mg ml⁻¹) or kinase buffer, substrate (0.5 mg ml⁻¹) and kinase (100 nM) were added to individual wells of a Pierce white opaque 96-well plate (Thermo Scientific). The substrate peptides used were RRRDDDSDDD 10-mer (Biaffin) and Poly-(Glu,Tyr 4:1) (BPS) for CK2 and Src kinases, respectively. For CK2, a specific inhibitor, Silmitasertib CX-4945 (Selleckchem), was added, where indicated, at a concentration of 1 µM. A detailed protocol is available in ref. 120.

### In vitro phosphatase assay

To verify the activity of our recombinantly purified λ phosphatase, we incubated cleared protein lysate from HeLa cells with λ phosphatase in phosphatase buffer (50 mM Tris-HCl pH 7.4, 100 mM NaCl, 2 mM DTT, 1 mM $MnCl_2$) for 30 min at 30 °C. Samples were analysed by SDS–PAGE and western blotting. A detailed protocol is available in ref. 121.

### Co-immunoprecipitation

Wild-type HeLa cells were transiently transfected with pHAGE_TEX264-GFP (Addgene_201925) or pHAGE_FAM134C-GFP (Addgene_201927) using Lipofectamine 3000 (Thermo Fisher) and treated for 6 h with Earle's balanced salt medium (Sigma-Aldrich) starvation medium to induce ER-phagy. BNIP3/NIX double-knockout HeLa cells were stably transduced with lentivirus to express V5-NIX (Addgene_223731) and, where indicated, treated with DFP to induce mitophagy. After the treatments, cells were lysed in lysis buffer (20 mM Tris-HCl pH 7.4, 150 mM KCl, 2.5 mM $MgCl_2$, 0.5% NP-40), cleared by centrifugation, and 6 mg of cleared lysate was incubated overnight with GFP-Trap agarose beads (Chromotek) for TEX264-GFP and FAM134C-GFP, or 7 mg of cleared lysate was incubated overnight with anti-V5 agarose beads (Sigma-Aldrich) for V5-NIX. In the morning, the samples were washed three times in washing buffer (20 mM Tris-HCl pH 7.4, 150 mM KCl, 2.5 mM $MgCl_2$) before the beads were resuspended in protein loading dye and analysed by western blotting. Detailed protocols are available in refs. 122,123.

### Pulldown with recombinant NIX-GST or BNIP3-GST

HeLa cells were lysed in lysis buffer (20 mM Tris-HCl pH 7.4, 150 mM KCl, 2.5 mM $MgCl_2$, 0.5% NP-40) for 20 min on ice before cell lysates were cleared by centrifugation. Beads were precoated with GST (negative control), NIX-GST or BNIP3-GST as described for the microscopy-based bead assay. Cell lysates were incubated overnight with precoated beads. In the morning, samples were washed four times in washing buffer (20 mM Tris-HCl pH 7.4, 100 mM KCl, 2.5 mM $MgCl_2$) before the beads were either submitted for analysis by mass spectrometry or western blotting. A detailed protocol is available in ref. 124. The primary antibodies used in this study are anti-GST (1:5,000, Sigma-Aldrich cat. no. SAB4200237, AB_2858197), anti-WIPI1 (1:200, Santa Cruz Biotechnology cat. no. sc-376205, AB_10989262), anti-WIPI2 (1:500, Bio-Rad cat. no. MCA5780GA, AB_10845951), anti-WIPI3 (Santa Cruz Biotechnology cat. no. sc-514194, AB_3101990), anti-WIPI4 (Abcam cat. no. ab168532, AB_3101989) and anti-PPTC7 (1:500, Abcam cat. no. ab122548, AB_11127117).

### Sample preparation for mass spectrometry analysis

For the identification of proteins bound to NIX-GST, beads were resuspended in 2 M urea in 50 mM ammonium bicarbonate, digested with LysC (FUJIFILM Wako Chemicals) and trypsin (Promega), washed with 1 M urea and 50 mM ammonium bicarbonate, and the supernatant reduced with 10 mM DTT before alkylation of free thiols with 20 mM iodoacetamide in the dark. The remaining iodoacetamide was quenched with 5 mM DTT. The urea concentration was diluted to 1 M with 50 mM ammonium bicarbonate, and fresh LysC and trypsin were added. The digestion was continued at 37 °C overnight and stopped by the addition of trifluoroacetic acid (TFA) to a final concentration of 0.5%, and the peptides were desalted using C18 StageTips[125,126]. A detailed protocol is available in ref. 127.

For quantitative analysis of whole cell proteomes, cell pellets were lysed in heated 2% sodium deoxycholate 0.1 M Tris/HCl pH 8.8 at 95 °C. When it was cooled down, 1 µl of benzonase was added and incubated on ice for 30 min, followed by sonication in a Bioruptor instrument and centrifugation to pellet cell debris. About 50 µg of the supernatant was reduced with 250 mM DTT, incubated with 500 mM iodoacetamide, and quenched with 250 mM DTT. The solution was diluted to 1% sodium deoxycholate using 50 mM ammonium bicarbonate. Proteins were digested with LysC (FUJIFILM Wako Chemicals) for 3 h, then digested further with trypsin (Promega) at 37 °C overnight. The digest was stopped by the addition of 10% TFA to a final concentration of 1%. The

samples were centrifuged and 10% TFA was added to the supernatant to a final concentration of 2%, followed by centrifugation, and the supernatant was desalted using C18 Stagetips[126]. A detailed protocol is available in ref. 128.

### Liquid chromatography mass spectrometry analysis

For identification of proteins bound to NIX-GST, peptides were separated on a Vanquish Neo nano-flow chromatography system (Thermo Fisher), using a trap-elute method for sample loading (Acclaim Pep-Map C18, 2 cm × 0.1 mm, 5 μm, Thermo Fisher) and a C18 analytical column (Acclaim PepMap C18, 50 cm × 0.75 mm, 2 μm, Thermo Fisher), applying a segmented linear gradient from 2% to 35% and finally 80% solvent B (80% acetonitrile, 0.1% formic acid; solvent A 0.1% formic acid) at a flow rate of 230 nl min$^{-1}$ over 120 min. An Exploris 480 Orbitrap mass spectrometer (Thermo Fisher) coupled to the LC column with a field asymmetric ion mobility spectrometry (FAIMS) pro ion-source (Thermo Fisher) using coated emitter tips (PepSep, MSWil) was used with the following settings. The mass spectrometer was operated in data-dependent acquisition (DDA) mode with two FAIMS compensation voltages (CVs) set to −45 and −60 V and 1.5-s cycle time per CV. The survey scans were obtained in a mass range of 350–1,500 $m/z$, at a resolution of 60k at 200 $m/z$ and a normalized automatic gain control (AGC) target at 100%. The most intense ions were selected with an isolation width of 1.2 $m/z$, fragmented in the higher-energy collisional dissociation cell at 28% collision energy and the spectra recorded for a max. of 50 ms at a normalized AGC target of 100% and a resolution of 15k. Peptides with a charge of +2 to +6 were included for fragmentation, the exclude isotope feature was enabled, and selected precursors were dynamically excluded from repeated sampling for 45 s.

For quantitative analysis of whole cell proteomes, 500-ng peptides were separated on an Ultimate 3000 RSLC nano-flow chromatography system (Thermo Fisher), using a pre-column for sample loading (Acclaim PepMap C18, 2 cm × 0.1 mm, 5 μm, Thermo Fisher) and a C18 analytical column (Aurora ultimate, 25 cm × 0.075 mm, 1.7 μm, Ionopticks), applying a segmented linear gradient from 2% to 35% in 60 min and finally 95% solvent B (80% acetonitrile, 0.1% formic acid; solvent A 0.1% formic acid) at a flow rate of 300 nl min$^{-1}$. Eluting peptides were analysed on a TimsTOF HT mass spectrometer (Bruker Daltonics) via a Captivespray electrospray source (Bruker Daltonics) using the included emitter tip of the Aurora analytical column. The mass spectrometer was operated in data independent acquisition (DIA) mode (diaPASEF) mode[129]. The ion mobility resolution was set to 0.64–1.42 V s cm$^{-1}$. The ramp time was set to 100 ms. The windows scheme included a variable $m/z$ width from 300 $m/z$ to 1,200 $m/z$, three windows per ion mobility scan, and eight ion mobility scans per duty cycle. Collisional induced dissociation energies ranged from 20 to 80 eV and scaled on ion mobilities, $1/k_0$.

### Mass spectrometry data analysis

For the identification of proteins bound to NIX-GST, Exploris raw files were first split according to CVs (−45 V, −60 V) using FreeStyle 1.7 software (Thermo Scientific). The resulting split MS data were analysed with FragPipe (19.1 or 20.0), using MSFragger[130], IonQuant[131] and Philosopher[132]. The default FragPipe workflow for label-free quantification (LFQ-MBR) was used, except 'Normalize intensity across runs' was turned off. Cleavage specificity was set to Trypsin/P, with two missed cleavages allowed. The protein false discovery rate (FDR) was set to 1%. A mass of 57.02146 (carbamidomethyl) was used as fixed cysteine modification, and methionine oxidation and protein N-terminal acetylation were specified as variable modifications. MS2 spectra were searched against the *Homo sapiens* 1 protein per gene reference proteome from UniProt (ID: UP000005640, release 2023_03), *Spodoptera* spp. sequences (UniProt taxonomy ID 7108, release 2023_03) and concatenated with a database of 382 common laboratory contaminants (release 2023.03; https://github.com/maxperutzlabs-ms/perutz-ms-contaminants) and two additional protein sequences

corresponding to the expressed transgenic constructs. Computational analysis was performed using Python and the in-house developed library MsReport (versions 0.0.11 and 0.0.19)[133]. Only non-contaminant proteins identified with a minimum of two peptides were considered for quantitative analysis. Label-free quantitation (LFQ) protein intensities reported by FragPipe were log$_2$-transformed and normalized across samples using the ModeNormalizer from MsReport. This method involves calculating log$_2$ protein ratios for all pairs of samples and determining normalization factors based on the modes of all ratio distributions. Missing values were imputed by drawing random values from a normal distribution. The sigma and mu of this distribution were calculated per sample from the standard deviation and median of the observed log$_2$ protein intensities ($\mu$ = median sample LFQ intensity − 1.8 standard deviations of the sample LFQ intensities, $\sigma$ = 0.3 × standard deviation of the sample LFQ intensities).

For quantitative analysis of whole cell proteomes, MS raw data were converted to htrms format using HTRMS converter (Biognosys), then searched with Spectronaut (18.3 or 19.5, Biognosys) in direct-DIA+ mode against the UniProt human reference proteome (version 2023_03 or 2024_01, www.uniprot.org), as well as a database of most common contaminants. The search was performed with full trypsin specificity and a maximum of two missed cleavages at a protein and peptide spectrum match FDR of 1%. Carbamidomethylation of cysteine residues was set as fixed, and oxidation of methionine and N-terminal acetylation as variable modifications. The cross-run normalization was turned off and all other settings were left as default. Computational analysis was performed using Python and the in-house developed Python library MsReport (version 0.0.19 or 0.0.27)[133]. LFQ protein intensities reported by Spectronaut were log$_2$-transformed and normalized across samples using the ModeNormalizer from MsReport. The missing normalized LFQ intensity values were imputed by drawing random values from a normal distribution after filtering out contaminants, proteins with fewer than two peptides and fewer than two quantified values in at least one group. Differences between groups were statistically evaluated using LIMMA 3.52.1[134] at 5% FDR (Benjamini–Hochberg).

### Protein structure prediction with AlphaFold2 Multimer

Structures of biochemically identified protein complexes were predicted with AlphaFold2 Multimer[135,136]. A locally installed version of AlphaFold2 Multimer was used for structure prediction with five models per prediction followed by Amber relaxation. Interaction scores (ipDT) and diagnostic plots (PAE plot and pLDDT plot) as well as the generated structures were inspected manually. Predicted structures were visualized with ChimeraX-1.8[137,138]. A detailed protocol is available in ref. 139.

### AlphaFold3 screen

AlphaFold3[136,140] was used to run pairwise predictions between WIPI2d and known selective autophagy receptors. Predictions with an ipTM score of >0.5 were considered putative hits, and diagnostic plots (PAE plot and pLDDT plot) as well as the generated structures were inspected manually. We also included FAM134C in our selection for experimental validation due to its ipTM score close to the 0.5 cutoff. The receptors included in the screen were ATL3 (P82987), BCL2L13 (Q9BXK5), BNIP3 (Q12983), C53 (O94874), CALCOCO1 (Q9P1Z2), CCPG1 (Q9ULG6), FAM134A (Q8NC44), FAM134B (Q9H6L5), FAM134C (Q86VR2), FKBP8 (Q14318), FUNDC1 (Q8IVP5), MCL-1 (Q07820), NBR1 (Q14596), NDP52 (Q13137), NIX (O60238), NLRX1 (Q86UT6), NUFIP1 (Q9UHK0), OPTN (Q96CV9), PHB2 (Q99623), RTN3 (O95197), SEC62 (Q99442), SQSTM1/p62 (Q13501), TAX1BP1 (Q86VP1), TEX264 (Q9Y6I9), YIPF3 (Q9GZM5) and YIPF4 (Q9BSR8). Soluble cargo receptors SQSTM1/p62, OPTN, NDP52, NBR1 and TAX1BP1 were predicted as dimers. Predicted structures were visualized with ChimeraX-1.8[137,138]. AlphaFold3 predictions for FKBP8, TEX264 and FAM134C were validated with AlphaFold2

Multimer accessed on the COSMIC2 server[141], resulting in similar predicted structures with the exception of FAM134C. Settings for AlphaFold2 Multimer were one prediction per model, full database and relaxation of best model. A detailed protocol is available in ref. [142].

## Molecular dynamics simulations

The initial complex structure for the simulations was obtained from an AlphaFold2.3 Multimer[135,136] prediction using the full-length WIPI2d sequence and residues 30–82 from NIX. The C-terminal IDR of WIPI2d was truncated by only using residues 1–362 for the simulations. The N terminus of the NIX fragment was capped with an acetyl group, the C termini of both proteins with an aminomethyl group, and standard protonation states were used for a pH of 7.

Simulations of the wild-type and LIR mutant were modelled by manually introducing the W36A and L39A mutations into the wild-type model. GROMACS (versions 2023.3 and 2023.4)[143] (SCR_014565) and the amber-disp force field[144] were used for all simulations. Proteins were solvated in water with 150 mM NaCl and neutralizing ions. We energy-minimized the system using the steepest descent algorithm with position restraints of $1,000 \text{ kJ mol}^{-1} \text{ nm}^{-2}$ on all heavy atoms and a maximum force of convergence of $1,000 \text{ kJ mol}^{-1} \text{ nm}^{-1}$. For equilibration, one NVT and four NPT steps running for 1, 2, 1, 5 and 10 ns were performed, respectively, with a timestep of 1 fs for the first three steps and 2 fs for the last two steps. The position restraints were gradually loosened on heavy atoms during equilibration, using $1,000 \text{ kJ mol}^{-1} \text{ nm}^{-2}$ in step 1 and $2,500 \text{ kJ mol}^{-1} \text{ nm}^{-2}$ in step 3, $100 \text{ kJ mol}^{-1} \text{ nm}^{-2}$ in step 4, and no restraints in step 5. All equilibration steps and the production run used a v-rescale thermostat[145] with a target temperature $T$ of 310 K and a characteristic time $\tau_T$ of 0.1 ps. The first NPT equilibration used a Berendsen barostat[146] with a target pressure $p$ of 1 bar, a characteristic time $\tau_p$ of 5.0 ps, and a compressibility of $4.5 \times 10^{-5} \text{ bar}^{-1}$. All other NPT steps and the production run used a Parrinello–Rahman barostat[147] with $p = 1$ bar, $\tau_p = 5.0$ ps and a compressibility of $4.5 \times 10^{-5} \text{ bar}^{-1}$. Production runs used a timestep of 2 fs and were run for 1 μs. Simulations of both systems were done in triplicate by initiating with different starting velocities.

In all simulations, we used a leapfrog integrator, a Verlet cutoff scheme[148], a cutoff of 1.0 nm modified with a potential shift for van der Waals interactions, a cutoff of 1.0 nm for Coulomb interactions and particle mesh Ewald for long-range electrostatics[149]. Energy and pressure corrections were applied for long-range van der Waals interactions. The LINCS algorithm[150] was used to describe bonds with hydrogens.

The behaviour of the NIX LIR and its interaction with WIPI2d were analysed by calculating three different quantities: the number of backbone hydrogen bonds $n_{\text{H-bonds}}$ between NIX residues 35–39 and WIPI2d residues 129–134, the minimum distance $d_{\text{TRP}}$ ($d_{\text{Ala}}$ in the ΔLIR mutant) between any heavy atom of NIX W36 (A36 in the ΔLIR mutant) and the $C_\alpha$ atom of WIPI2d L119 (as a measure of W36/A36 insertion depth), and the minimum distance $d_{\text{pocket}}$ between the side-chain heavy atoms of WIPI2d I133 and F169 (as a measure of pocket opening). Trajectory frames every 1 ns were used for the analysis. For implementation of the described analysis we used Python3 (SCR_008394) with Anaconda3 (SCR_025572), iPython[151] (SCR_001658), Numpy[152] (SCR_008633), Matplotlib[153] (SCR_008624) and MDAnalysis[154] (SCR_025610). We used VMD[155] (SCR_001820) and ChimeraX[156] (SCR_015872) for visual analysis and renders.

## Statistics and reproducibility

Experiments were repeated at least three times unless otherwise stated in the figure legends. The $N$ number for all cell experiments represents independent experimental cultures. Statistical analyses were performed using the statistical tests stated in each figure legend. Data are presented as mean ± standard deviation if not indicated otherwise. The data distribution was assumed to be normal, but this was not formally tested. For the quantification of immunoblots, images were processed with ImageJ (National Institutes of Health). For flow cytometry quantification, we analysed >10,000 cells with triplicate experiments, which showed consistent results throughout the replicates. FlowJo was used for data analysis and quantification. All graphs were plotted using GraphPad Prism version 9.5.1. Depending on the number of samples, and as specified in the figure legends, we employed a $t$-test, one- or two-way ANOVA test with appropriate multiple comparison tests. All $P$ values below 0.05 were considered significant, with *$P \le 0.05$, **$P \le 0.01$, ***$P \le 0.001$ and ****$P \le 0.0001$. Exact $P$ values are provided in the source numerical data. Error bars are reported as mean ± standard deviation. Statistical analyses were performed using PRISM software (GraphPad Software). We have not excluded any samples. However, we reused the data from Fig. 3c in Fig. 4f as they contain the same experimental set-up. No statistical methods were used to predetermine sample sizes, but our sample sizes are similar to those reported in previous publications[7,41,157]. Data collection and analysis were not performed blind to the conditions of the experiments and no randomization was applied. To ensure the reproducibility of experiments not quantified or subjected to statistical analysis, we show one representative replicate of two replicates or more in the Article, as indicated in the figure legends.

### Reporting Summary

Further information on research design is available in the Nature Portfolio Reporting Summary linked to this Article.

## Data availability

Raw files associated with this work have been made available on Zenodo (https://doi.org/10.5281/zenodo.14867723)[158]. The mass spectrometry proteomics data have been deposited to the ProteomeXchange Consortium via the PRIDE partner repository with the dataset identifiers PXD060351, PXD060356 and PXD060363. Plasmids constructed for and used in this manuscript are available at Addgene. The data, protocols and key laboratory materials used in this study are listed in a Key Resource Table alongside their persistent identifiers as a Supplementary file. All other data supporting the findings of this study are available from the corresponding author on reasonable request. Further information on the research design is available in the Nature Research Reporting Summary linked to this Article. Source data are provided with this paper.

## Code availability

All data cleaning, preprocessing, analysis and visualization were performed using the software packages described in the Reporting Summary. Python scripts are available at https://github.com/bio-phys/nix-lir-binding-to-wipi2d (ref. [159]) and https://doi.org/10.5281/zenodo.15719310 (ref. [160]).

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

## Acknowledgements

We thank members of the Martens laboratory, M. Chen, L. Uoselis, D. Fracchiolla and other members of the Aligning Science Across Parkinson's (ASAP) Mito911 Team for their help and advice. We thank D. Bernklau for the optimization of the ATG9A-vesicle purification protocol, and E. Holzer and S. Tulli for microscopy support. We thank the Max Perutz Labs BioOptics, Flow Cytometry, Monoclonal Antibody, and Mass Spectrometry facilities for their technical support. Proteomics analyses were performed by the Mass Spectrometry Facility at Max Perutz Labs using the VBCF instrument pool. We thank I. Bilusic Vilagos and the rest of the Vienna BioCenter Core Facilities (VBCF) Protech Facility for help with HEK cell expressions. The schematics were generated with BioRender. Molecular graphics and analyses were performed with UCSF ChimeraX, developed by the Resource for Biocomputing, Visualization and Informatics at the University of California, San Francisco, with support from National Institutes of Health R01-GM129325 and the Office of Cyber Infrastructure and Computational Biology, National Institute of Allergy and Infectious Diseases. This work was supported by a Marie Skłodowska-Curie MSCA Postdoctoral fellowship (101062916 to E.A.), a travel grant from the Flanders Fund for Scientific Research (FWO-Flanders to E.A.) and a Rebecca Cooper Foundation Fellowship (RC20241396 to M.L.). J.F.M.S. and G.H. thank the Max Planck Society and the Clusterproject ENABLE funded by the Hessian Ministry for Science and the Arts for financial support, and the Max Planck Computing and Data Facility for computational resources. This research was funded in whole or in part by Aligning Science Across Parkinson's (ASAP-000350 to S.M., J.H.H., M.L. and G.H.) through the Michael J. Fox Foundation for Parkinson's Research (MJFF). For the purpose of open access, we have applied a CC-BY 4.0 public copyright license to all author accepted manuscripts (AAM) arising from this submission.

## Author contributions

E.A. and S.M. conceived the project. E.A., S.S., A.S.I.C., J.S.-M., M.L., J.H.H. and S.M. designed the experiments. E.A., S.S., A.S.I.C., J.S.-M., T.N.N., X.R., M.S., J.R., G.K. and L.U. performed the experiments. J.F.M.S. carried out the MD simulations and part of the AlphaFold predictions, supervised by G.H. E.A. and S.M. wrote the original draft, to which all authors contributed by editing and reviewing.

## Funding

## Competing interests

S.M. is a member of the scientific advisory board of Casma Therapeutics, J.H.H. is a co-founder and shareholder of Casma Therapeutics, has consulted for Corsalex, and receives research funding from Genentech and Hoffmann-La Roche. M.L. is a co-founder and member of the scientific advisory board of Automera. The remaining authors declare no competing interests.

## Additional information

**Extended data** is available for this paper at https://doi.org/10.1038/s41556-025-01712-y.

**Correspondence and requests for materials** should be addressed to Elias Adriaenssens or Sascha Martens.

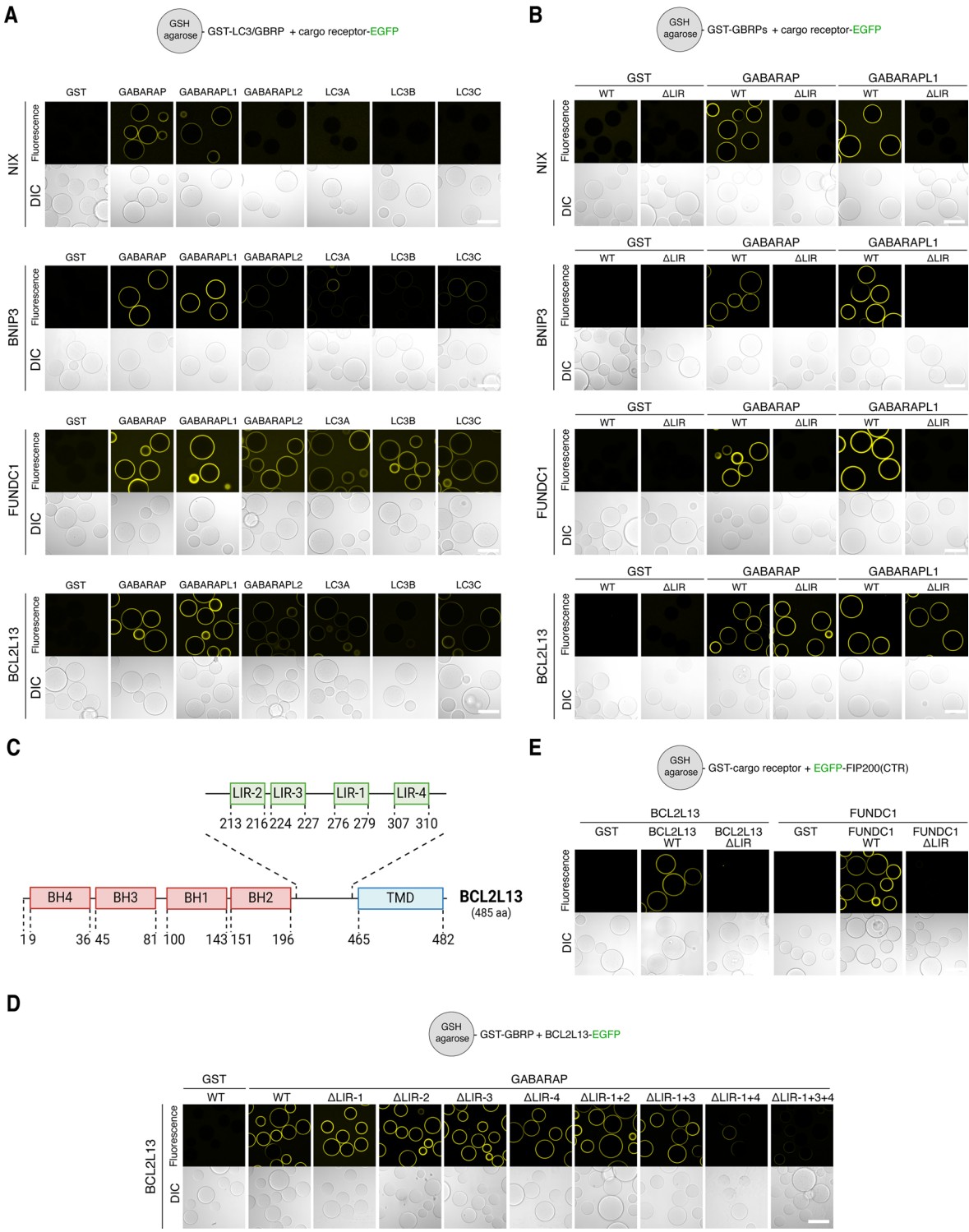

**Extended Data Fig. 1 | In vitro validation of mitophagy cargo receptors and their LIR/FIR motifs.** (**a**) Microscopy-based bead assay of agarose beads coated with GST-tagged LC3A/B/C or GBRP/GBRPL1/GBRPL2 and incubated with GFP-tagged cargo receptors FUNDC1, BCL2L13, NIX, and BNIP3. (**b**) As in (**a**) but with wild-type (WT) or alanine-mutated LIR-motifs (ΔLIR) of the GFP-tagged cargo receptors. (**c**) Schematic of domain structure of BCL2L13 with the candidate LIR/FIR motifs indicated with residue numbers. LIR1 was previously annotated in literature as the active LIR motif. (**d**) As in (**a**), but with different alanine-mutated variants of the different LIR-motifs (ΔLIR) of GFP-tagged BCL2L13. (**e**) As in (**a**) but with GST-tagged cargo receptors and GFP-tagged C-terminal region (CTR; residues 1429–1591) of FIP200. Results are representative of three biologically independent replicates (**a-b, d-e**). Scale bars, 100 μm. Schematic generated with BioRender.

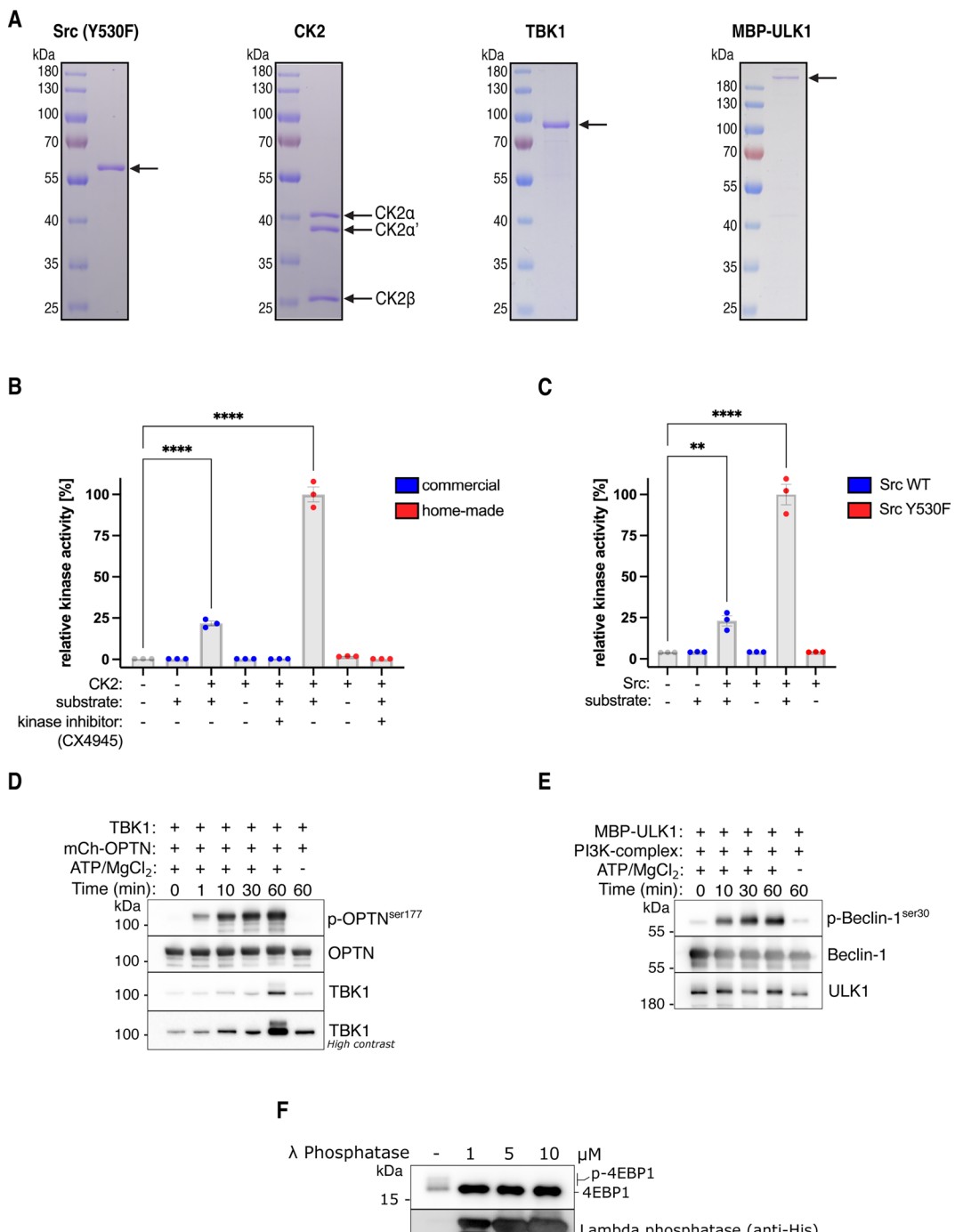

**Extended Data Fig. 2 | Validation of the activity of the purified kinases and λ phosphatase.** (**a**) Representative SDS-PAGE gels of purified Src (Y530F), CK2 complex, TBK1, and MBP-ULK1. Arrows indicate the predicted molecular weight. (**b, c**) Measurement of kinase activity using a plate-reader-based read-out. Kinases were incubated with or without a substrate peptide or kinase inhibitor. Kinase activity was compared between our purified CK2 complex (home-made) and commercially available CK2, or between wild-type (WT) and Y530F mutant Src. Data are presented as mean ± s.d. (n = 3 biologically independent experiments). One-way ANOVA with Dunnett's multiple comparisons test. **P < 0.005, ****P < 0.0001. (**d**) Measurement of kinase activity by mixing

recombinantly purified mCherry-OPTN and TBK1 for the indicated time and western blot analysis using antibodies for phosphorylated OPTN (S177) as a readout for TBK1 activity. (**e**) As in (**d**), but after mixing recombinantly purified MBP-ULK1 and the PI3KC3−C1 complex (composed of ATG14, Beclin-1, Vps15, Vps34) for the indicated time and using antibodies for phosphorylated Beclin-1 (Ser30) as a readout for ULK1 activity. (**f**) Measurement of λ phosphatase activity by mixing recombinantly purified λ phosphatase with HeLa cell lysate for 30 min at 30 degrees Celsius. Immunoblotted for dephosphorylation of 4EBP1. Source numerical data, including exact P values, and unprocessed blots are available in source data.

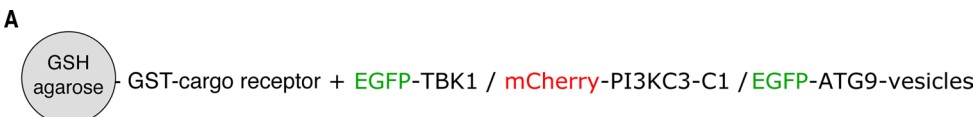

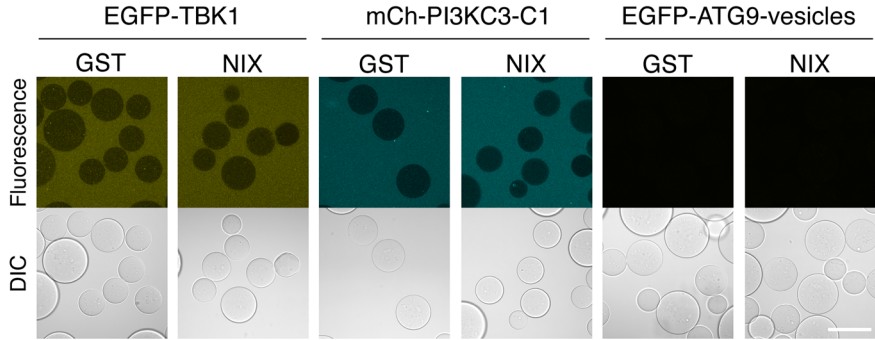

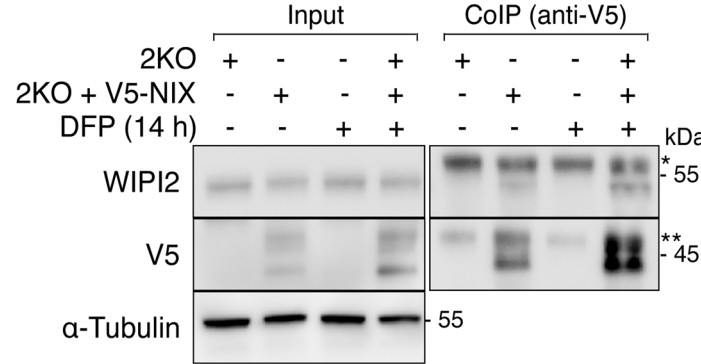

**Extended Data Fig. 3 | NIX interacts with WIPI2 but not with TBK1, PI3KC3–C1 complex, or purified ATG9A-vesicles.** (**a**) Microscopy-based bead assay of agarose beads coated with GST-tagged NIX and incubated with GFP-tagged TBK1, mCherry-tagged PI3KC3–C1, or GFP-tagged ATG9A-vesicles purified from HAP1 cells. GST served as a negative control. (**b**) Co-immunoprecipitation of V5-tagged NIX expressed in NIX/BNIP3 double-knockout (2KO) cells and immunoblotted for the interaction with WIPI2. As a negative control, 2KO cells without V5-NIX were used. * Heavy chains, ** non-specific band. Results are representative of three biologically independent replicates. Scale bar, 100 μm. Unprocessed blots are available in source data.

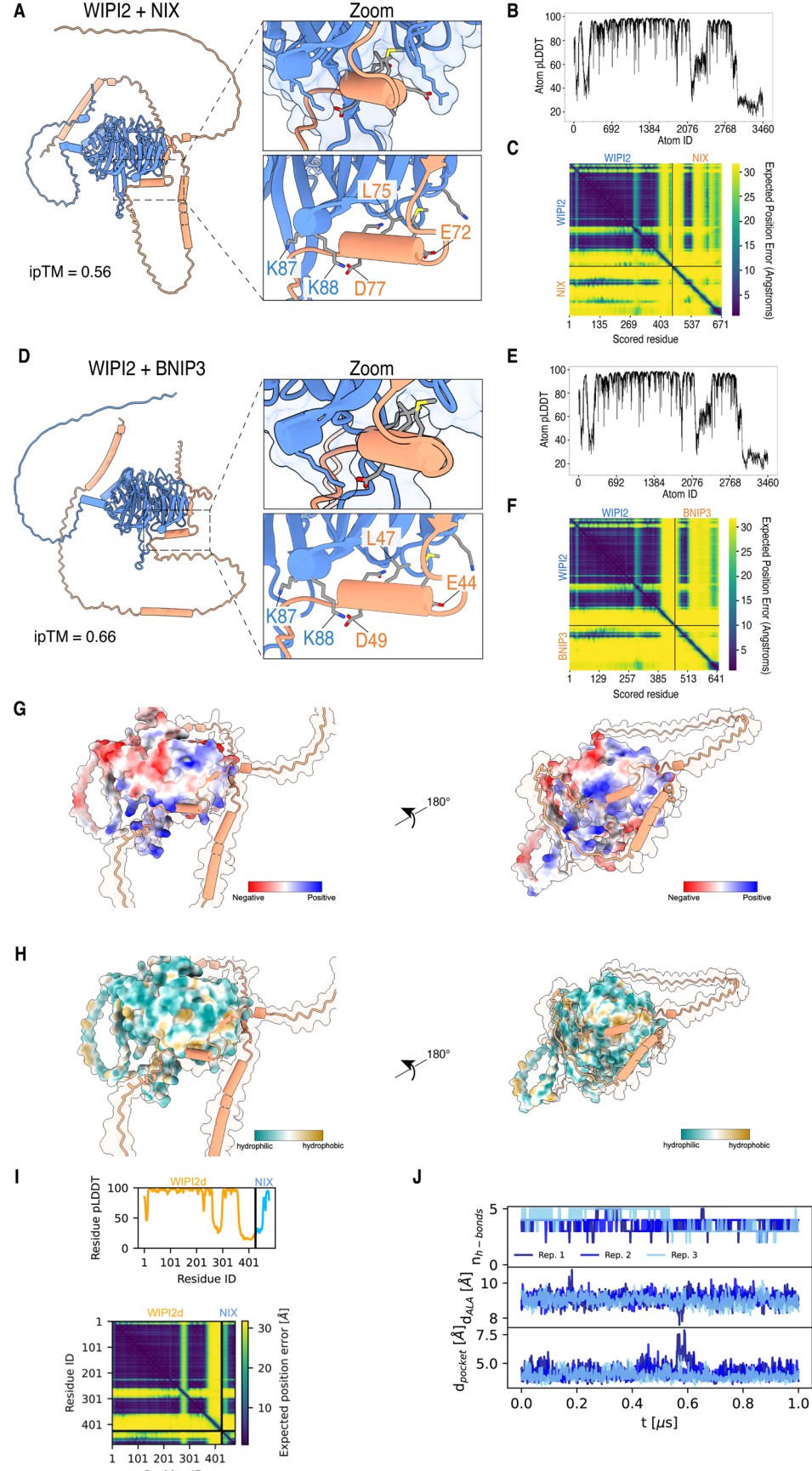

**Extended Data Fig. 4 | See next page for caption.**

**Extended Data Fig. 4 | AlphaFold2 prediction and MD simulations of BNIP3/ NIX-WIPI2 complex.** (**a**) AlphaFold2 predicted structure of NIX (orange) and WIPI2 (blue) with zoom-in on the interaction interface. (**b, c**) pLDDT and PAE plots for NIX-WIPI2 structure. (**d**) AlphaFold2 predicted structure of BNIP3 (orange) and WIPI2 (blue) with zoom-in on the interaction interface. (**e, f**) pLDDT and PAE plots for BNIP3-WIPI2 structure. (**g**) Predicted structure for the NIX-WIPI2 complex with the surface of WIPI2 coloured based on electrostatics. (**h**) Predicted structure for the NIX-WIPI2 complex with the surface of WIPI2 coloured based on hydrophobics. Note that the indicated residue numbers for WIPI2 correspond to their residue number in the WIPI2d sequence (which matches K105 and K106 in WIPI2b). (**i**) Residue pLDDT and PAE scores for the prediction in Fig. 2i. (**j**) The NIX W36A/L39A (ΔLIR) mutant does not bind the cryptic pocket of WIPI2d. Number of backbone h-bonds $n_{h\text{-bonds}}$ between the LIR of NIX and WIPI2d, insertion depth $d_{ALA}$ of NIX ΔLIR A36, and minimum heavy atom distance $d_{pocket}$ between WIPI2d F169 and I133 from three 1 μs MD simulations. Data are presented for three independent experiments.

**A**

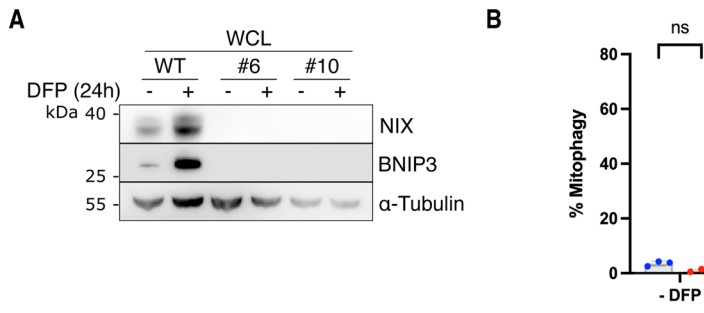

**B**

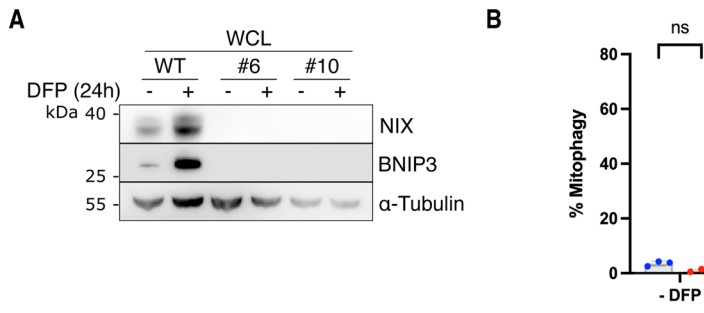

**C**

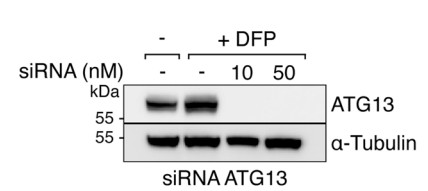

**D**

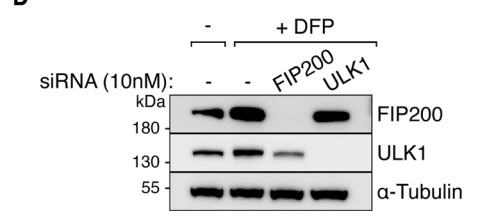

**Extended Data Fig. 5 | Validation of knockout and knockdown cell lines.**
(**a**) Analysis of whole cell lysates (WCL) by SDS-PAGE and western blotting for NIX/BNIP3 double-knockout clones #6 and #10, with and without induction of mitophagy by 24 h of DFP treatment. Results are representative of two biologically independent replicates. (**b**) Mitophagy flux measured by flow cytometry of wild-type (WT) or NIX/BNIP3 double-knockout (DKO) HeLa cells (clone #6), left untreated or treated with DFP for 24 h. Data are presented as mean ± s.d. (n = 3 biologically independent experiments). Two-way ANOVA with Tukey's multiple comparisons test. ****$P < 0.0001$. ns, not significant. (**c**) Analysis of knockdown efficiency for ATG13. HeLa cells were transfected 72 h prior to the

FACS experiment, with the addition of DFP for the last 24 h to induce mitophagy, and analysed by flow cytometry. Cells were collected after the experiment and analysed with SDS-PAGE and western blotting. We selected the concentration of 10 nM for the FACS experiments throughout this manuscript. Results are representative of two biologically independent replicates. (**d**) As in (**c**), but for HeLa cells transfected with siRNAs against FIP200, ULK1 or scrambled as a control (−). Results are representative of two biologically independent replicates. Source numerical data, including exact $P$ values, and unprocessed blots are available in source data.

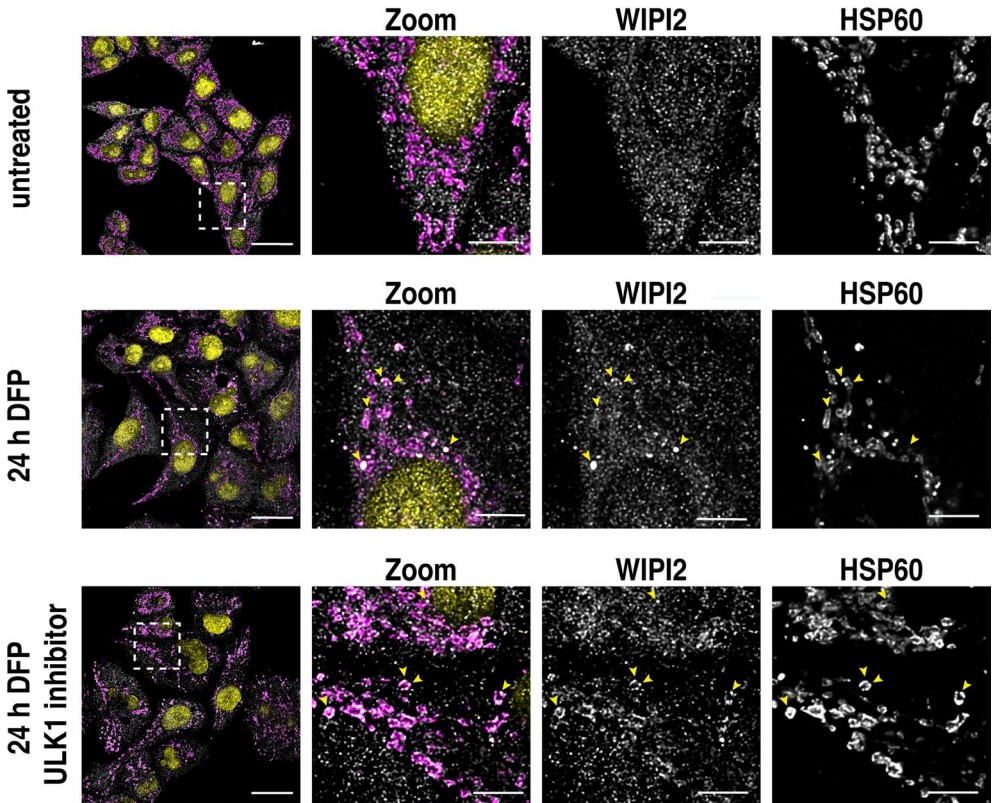

**Extended Data Fig. 6 | ULK1 acts downstream of WIPI2.** Deconvoluted confocal microscopy images of untreated or 24 h DFP-treated HeLa cells in the presence or absence of a ULK1 inhibitor. All cells were depleted for PPTC7, which activates a low level of mitophagy in untreated cells but boosts mitophagy profoundly in DFP-treated cells – aiding in the visualisation of mitophagy. Data are presented for two biologically independent experiments. Scale bars, 20 μm and 5 μm (zoom).

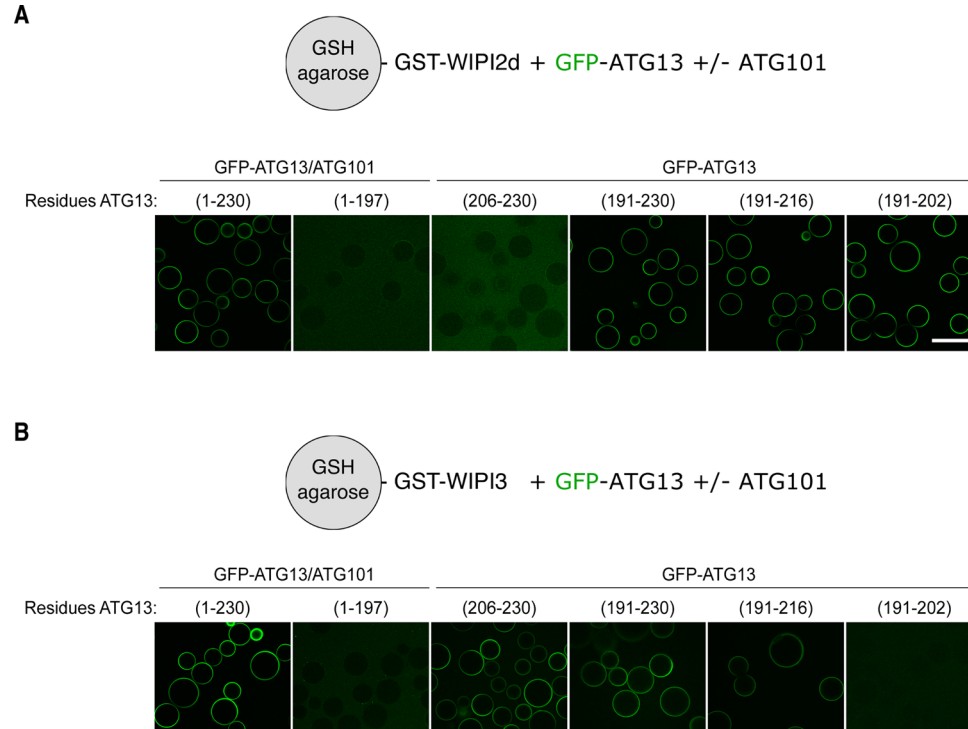

**Extended Data Fig. 7 | Biochemical mapping of binding sites of WIPI-ATG13 interaction.** (**a**) Microscopy-based bead assay of agarose beads coated with GST-tagged WIPI2d or (**b**) WIPI3 and incubated with GFP-tagged ATG13/ATG101 subcomplex or fragments of ATG13 alone. Results are representative of three independent replicates. Scale bars, 100 μm.

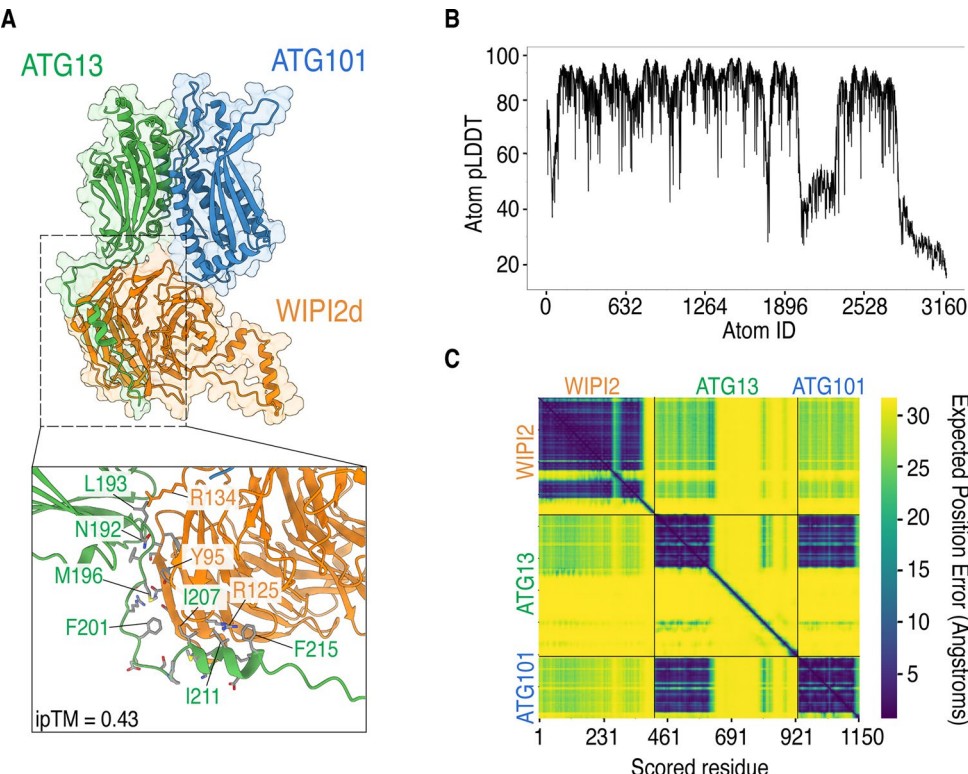

**Extended Data Fig. 8 | AlphaFold2 prediction of WIPI2d and ATG13/101 subcomplex.** (**a**) AlphaFold2 Multimer predicted structure for WIPI2, lacking the most C-terminal ten amino acids, with ATG13/101, with zoom in on the interaction interface, (**b, c**) pLDDT and PAE plots for WIPI2-ATG13/101 structure.

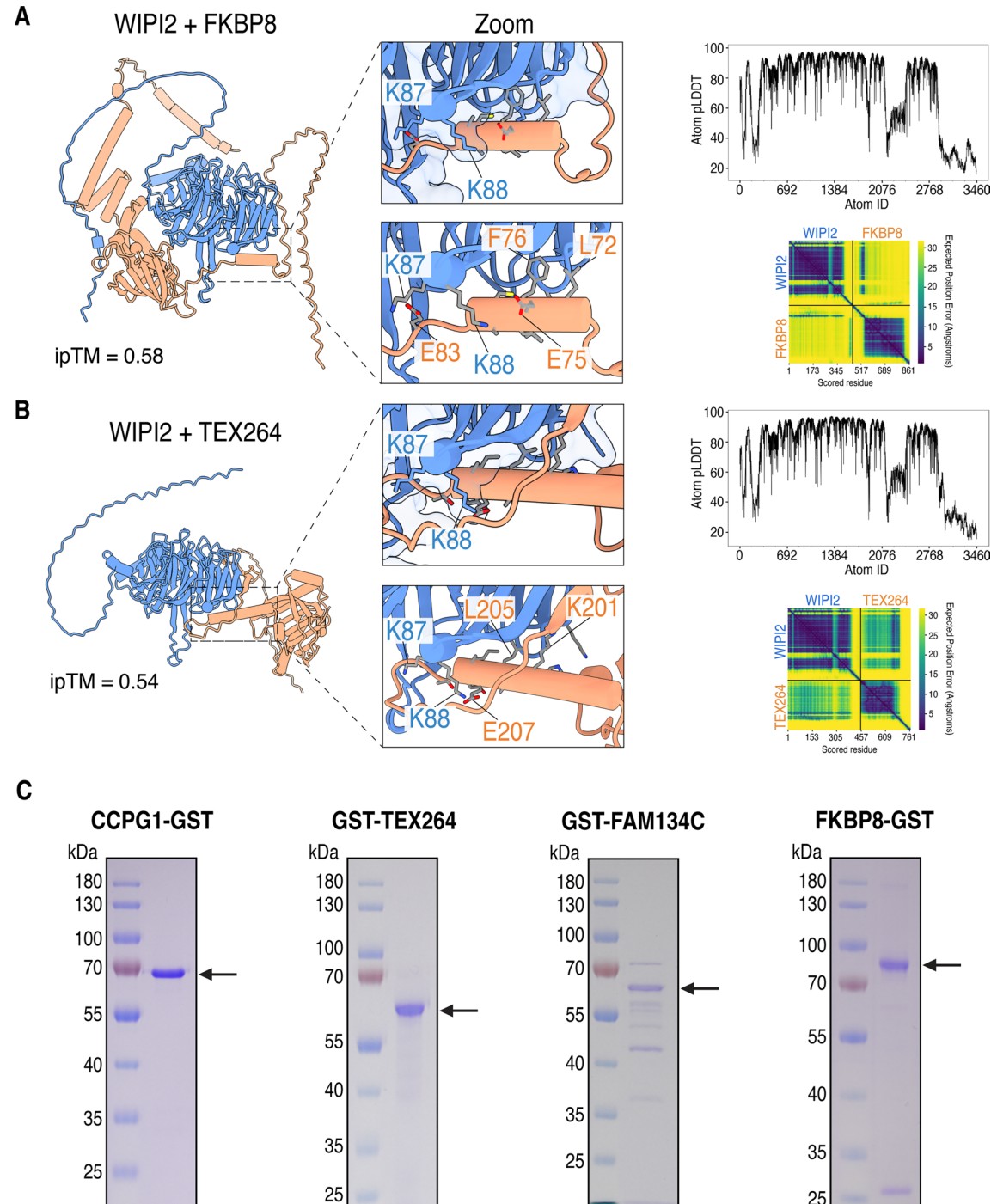

**Extended Data Fig. 9 | AlphaFold2 prediction of WIPI2d and transmembrane cargo receptors.** (**a**, **b**) AlphaFold3 predicted structure for WIPI2 with (**a**) TEX264 or (**b**) FKBP8, with zoom on the interaction interface. Note that the indicated residue numbers for WIPI2 correspond to their residue number in the WIPI2d sequence. pLDDT plots and predicted alignment error (PAE) heatmap are also shown. (**c**) Representative SDS-PAGE gels stained with Coomassie Brilliant Blue of purified CCPG1 (residues 1–212)-GST, GST-TEX264 (residues 28–313), GST-FAM134C (residues 250–466), and FKBP8 (residues 1–391)-GST. Arrows indicate the predicted molecular weight. Unprocessed blots are available in source data.

Sascha Martens

# Reporting Summary

## Statistics

For all statistical analyses, confirm that the following items are present in the figure legend, table legend, main text, or Methods section.

| n/a | Confirmed | |
|---|---|---|
| ☐ | ☒ | The exact sample size (*n*) for each experimental group/condition, given as a discrete number and unit of measurement |
| ☐ | ☒ | A statement on whether measurements were taken from distinct samples or whether the same sample was measured repeatedly |
| ☐ | ☒ | The statistical test(s) used AND whether they are one- or two-sided *Only common tests should be described solely by name; describe more complex techniques in the Methods section.* |
| ☒ | ☐ | A description of all covariates tested |
| ☐ | ☒ | A description of any assumptions or corrections, such as tests of normality and adjustment for multiple comparisons |
| ☐ | ☒ | A full description of the statistical parameters including central tendency (e.g. means) or other basic estimates (e.g. regression coefficient) AND variation (e.g. standard deviation) or associated estimates of uncertainty (e.g. confidence intervals) |
| ☐ | ☒ | For null hypothesis testing, the test statistic (e.g. *F*, *t*, *r*) with confidence intervals, effect sizes, degrees of freedom and *P* value noted *Give P values as exact values whenever suitable.* |
| ☒ | ☐ | For Bayesian analysis, information on the choice of priors and Markov chain Monte Carlo settings |
| ☒ | ☐ | For hierarchical and complex designs, identification of the appropriate level for tests and full reporting of outcomes |
| ☒ | ☐ | Estimates of effect sizes (e.g. Cohen's *d*, Pearson's *r*), indicating how they were calculated |

*Our web collection on statistics for biologists contains articles on many of the points above.*

## Software and code

Policy information about availability of computer code

| Data collection | Confocal microscopy images were collected using ZEN software version 2022 (Carl Zeiss Microscopy, GmbH, Germany) connected to a LSM700 or Zen Blue software (Carl Zeiss Microscopy, GmbH, Germany; RRID:SCR_013672) connected to a LSM900. Alphafold-2 (RRID:SCR_025454), Alphafold Multimer (10.1101/2021.10.04.463034) , and Alphafold-3  (RRID:SCR_025454) were used to predict protein and protein complex structures. MD simulations were performed with Gromacs (versions 2023.3 and 2023.4; RRID:SCR_014565) with amber-disp force field. Code for MD simulations can in part be found here: https://github.com/bio-phys/nix-lir-binding-to-wipi2d |
|---|---|
| Data analysis | 1. FlowJo10 (version 10.9.0) software (Tree Star Inc., Ashland, OR, USA) for FACS-data analysis (RRID:SCR_008520). |

2. PRISM 9 software (version 9.5.1; Graphpad Software, La Jolla, CA, USA) for statistical analysis and generating graphs (RRID:SCR_005375).
3. ImageJ software (Schindelin et al. 2015) for immunofluorescence microscopy image analysis (RRID:SCR_003070).
4. FACSDiva software (BD FACSDiva software) for flow cytometry experiments (RRID:SCR_001456).
5. FreeStyle 1.7 software (Thermo Scientific) for mass spectrometry data.
6. Fragpipe (version 19.1 and 20.0) for mass spectrometry data analysis (RRID:SCR_022864) - https://github.com/Nesvilab/FragPipe
7. MSFragger (version 3.7 and 3.8) for mass spectrometry data analysis - https://github.com/Nesvilab/MSFragger
8. IonQuant (version 1.8.10 and 1.9.8) for mass spectrometry data analysis - https://github.com/Nesvilab/IonQuant
9. Philosopher (version 4.8.0 and 5.0.0) for mass spectrometry data analysis - https://github.com/Nesvilab/philosopher
10. MaxQuant software version 1.6.17.0 for mass spectrometry data analysis.
11. Spectronaut (version 18.3 or 19.5, Biognosys) for mass spectrometry data analysis.
12. MSReport (version 0.019 or 0.0.27) for mass spectrometry data analysis.
13. LIMMA (version 3.52.1) for statistical analysis of mass spectrometry data.
14. Common laboratory contaminants database (developed in house; https://github.com/maxperutzlabs-ms/perutz-ms-contaminants).
15. ChimeraX-1.8 for visualisation of predicted protein structures (RRID:SCR_015872).

16. Python3 (version 3.10.9) for MD analysis (RRID:SCR_008394).
17. Anaconda3 (version 2019.10) for MD analysis (RRID:SCR_025572).
18. iPython (version 8.10.0) for MD analysis (RRID:SCR_001658).
19. Numpy (version 1.26.2) for MD analysis (RRID:SCR_008633).
20. Matplotlib (version 3.7.1) for MD analysis (RRID:SCR_008624).
21. MDAnalysis (version 2.3.0) for MD analysis (RRID:SCR_025610).
22. VMD (version 1.9.3 and 1.9.4) for MD analysis (RRID:SCR_001820).
23. LINCS algorithm to describe hydrogen bonds during MD analysis
24. Gromacs (version 2023.3 and 2023.4) for MD analysis ((RRID:SCR_014565)).
25. Airyscan processing plug-in in Zen Blue software (Zeiss)
26. Huygens Professional 24.04 (Scientific Volume Imaging) was used to deconvolve LSM700 images (RRID:SCR_014237)
27. Synthego ICE CRISPR analysis (version 2) (RRID:SCR_024508) https://www.synthego.com/products/bioinformatics/crispr-analysis

For manuscripts utilizing custom algorithms or software that are central to the research but not yet described in published literature, software must be made available to editors and reviewers. We strongly encourage code deposition in a community repository (e.g. GitHub). See the Nature Portfolio guidelines for submitting code & software for further information.

# Data

Policy information about availability of data

All manuscripts must include a data availability statement. This statement should provide the following information, where applicable:
- Accession codes, unique identifiers, or web links for publicly available datasets
- A description of any restrictions on data availability
- For clinical datasets or third party data, please ensure that the statement adheres to our policy

Raw files associated with this work have been made available on Zenodo (https://doi.org/10.5281/zenodo.14867723). The mass spectrometry proteomics data have been deposited to the ProteomeXchange Consortium via the PRIDE partner repository with the dataset identifiers PXD060351, PXD060356, and PXD060363. Source data are provided with this paper. Plasmids constructed for and used in this manuscript are available at Addgene. The data, protocols, and key lab materials used in this study are listed in a Key Resource Table alongside their persistent identifiers as Supplementary file. All other data supporting the findings of this study are available from the corresponding author on reasonable request. Further information on the research design is available in the Nature Research Reporting Summary linked to this article.

Zenodo: https://doi.org/10.5281/zenodo.14867723
Github MD analysis: https://github.com/bio-phys/nix-lir-binding-to-wipi2d
Pride (MS datasets): PXD060351, PXD060356, and PXD060363

# Research involving human participants, their data, or biological material

Policy information about studies with human participants or human data. See also policy information about sex, gender (identity/presentation), and sexual orientation and race, ethnicity and racism.

| Reporting on sex and gender | N/A |
| --- | --- |
| Reporting on race, ethnicity, or other socially relevant groupings | N/A |
| Population characteristics | N/A |
| Recruitment | N/A |
| Ethics oversight | N/A |

Note that full information on the approval of the study protocol must also be provided in the manuscript.

# Field-specific reporting

Please select the one below that is the best fit for your research. If you are not sure, read the appropriate sections before making your selection.

☒ Life sciences      ☐ Behavioural & social sciences      ☐ Ecological, evolutionary & environmental sciences

For a reference copy of the document with all sections, see nature.com/documents/nr-reporting-summary-flat.pdf

# Life sciences study design

All studies must disclose on these points even when the disclosure is negative.

| Sample size | No statistical methods were applied to pre-evaluate sample size. Experiments were performed at least as three replicates, according to current practices in the field. Statistical analysis was performed on experiments for which the sample size included at least 3 biological replicates. Sample sizes were based on previous experience and current standards in the field. |
| --- | --- |

| Data exclusions | No data were excluded from the analyses. |
| Replication | All experiments were replicated at least three times with similar findings. Samples sizes are provided in the figure legends. |
| Randomization | Samples were allocated into experimental groups by genotype of knockout condition. Covariates were controlled for by maintaining all samples in the same growth and media conditions. |
| Blinding | The investigators were not blinded to treatment or genotype allocations during this study. For cell based and biochemistry experiments, it was not possible to blind the experimenter. |

# Reporting for specific materials, systems and methods

We require information from authors about some types of materials, experimental systems and methods used in many studies. Here, indicate whether each material, system or method listed is relevant to your study. If you are not sure if a list item applies to your research, read the appropriate section before selecting a response.

## Materials & experimental systems

| n/a | Involved in the study |
|---|---|
| ☐ | ☒ Antibodies |
| ☐ | ☒ Eukaryotic cell lines |
| ☒ | ☐ Palaeontology and archaeology |
| ☒ | ☐ Animals and other organisms |
| ☒ | ☐ Clinical data |
| ☒ | ☐ Dual use research of concern |
| ☒ | ☐ Plants |

## Methods

| n/a | Involved in the study |
|---|---|
| ☒ | ☐ ChIP-seq |
| ☐ | ☒ Flow cytometry |
| ☒ | ☐ MRI-based neuroimaging |

## Antibodies

| Antibodies used | anti-α-Tubulin (1:5000, Abcam Cat# ab7291, RRID:AB_2241126)<br>anti-ATG13 (1:1000 for western blot, Cell Signaling Technology Cat# 13468, RRID:AB_2797419)<br>anti-ATG13 (1:200 for immunofluorescence, Cell Signaling Technology, Cat# 13468; RRID:AB_2797419)<br>anti-Beclin1 (1:1000, Cell Signaling Technology Cat# 3738, RRID:AB_490837)<br>anti-phospho-Beclin1 Ser30 (1:1000, Cell Signaling Technology Cat# 54101, RRID:AB_3102019)<br>anti-BNIP3 (1:1000, Cell Signaling Technology Cat# 44060, RRID:AB_2799259)<br>anti-CCPG1 (1:1000, Cell Signaling Technology Cat# 80158, RRID:AB_2935809)<br>anti-COXII (1:1000, Abcam Cat# ab110258, RRID:AB_10887758)<br>anti-COXII (1:1000, Cell Signaling Technology Cat# 31219, RRID:AB_2936222)<br>anti-4EBP1 (1:1000, Proteintech Cat# 60246-1-Ig, RRID:AB_2881368)<br>anti-FIP200 (1:1000, Cell Signaling Technology Cat# 12436, RRID:AB_2797913)<br>anti-GFP (1:1000, Millipore Cat# MABC1689, RRID:AB_3675504)<br>anti-GST (1:1000, Sigma-Aldrich, SAB4200237 , RRID: AB_2858197)<br>anti-penta-His (1:1000, Qiagen Cat# 34660, RRID:AB_2619735)<br>anti-LC3B (1:500, Nanotools Cat# 0260-100/LC3-2G6, RRID:AB_2943418)<br>anti-NIX/BNIP3L (1:1000, Cell Signaling Technology Cat# 12396, RRID:AB_2688036)<br>anti-OPTN (1:500, Sigma Aldrich Cat# HPA003279, RRID:AB_1079527)<br>anti-HSP60 (1:800, Abcam Cat# ab46798, RRID:AB_881444)<br>anti-phospho-OPTN Ser177 (1:1000, Cell Signaling Technology Cat# 57548, RRID:AB_2799529)<br>anti-p62/SQSTM1 (1:1000, Abnova Cat# H00008878-M01, RRID:AB_437085)<br>anti-phospho-p62/SQSTM1 Ser403 (1:1000, Cell Signaling Technology Cat# 39786, RRID:AB_2799162)<br>anti-PPTC7 (1:1000, Abcam, ab122548, RRID: AB_11127117)<br>anti-TEX264 (1:1000, Sigma-Aldrich Cat# HPA017739, RRID:AB_1857910)<br>anti-ULK1 (1:1000, Cell Signaling Technology Cat# 8054, RRID:AB_11178668)<br>anti-V5 (1:1000, Thermo Fisher Scientific Cat# R960-25, RRID:AB_2556564)<br>anti-WIPI1 (1:200, Santa Cruz Biotechnology, Cat# sc-376205, RRID:AB_10989262)<br>anti-WIPI2 (1:1000 for western blot, Bio-Rad Cat# MCA5780GA, RRID:AB_ 10845951)<br>anti-WIPI2 (1:100 for immunofluorescence, Abcam Cat# ab105459, RRID:AB_10860881)<br>anti-WIPI3 (1:200, Santa Cruz Biotechnology, sc-514194, RRID:AB_3101990)<br>anti-WIPI4 (1:1000, Abcam, ab168532; RRID:AB_3101989)<br><br>HRP conjugated polyclonal goat anti-mouse (Jackson ImmunoResearch Labs Cat# 115-035-003, RRID:AB_10015289)<br>HRP conjugated polyclonal goat anti-rabbit (Jackson ImmunoResearch Labs Cat# 111-035-003, RRID:AB_2313567)<br>AlexaFluor-488 goat anti-Mouse IgG (H+L) (1:500, Thermo Fisher Scientific Cat# A-11001, RRID:AB_2534069)<br>AlexaFluor-546 goat anti-rabbit IgG (H+L) (1:500, Thermo Fisher, Cat# A-11035; RRID: AB_2534093). |
| Validation | Antibodies were selected based on their use in other publications and/or validation by the manufacturers for their respective application. Where possible, knockout cell lines were used to validate the specificity of the antibodies further. |

# Eukaryotic cell lines

Policy information about cell lines and Sex and Gender in Research

| | |
|---|---|
| Cell line source(s) | All parental cell lines (HeLa, HEK293T, HEK293F) were acquired from the American Type Culture Collection (ATCC). HeLa knockout cell lines were generated during this study and submitted to Cellosaurus. HAP1 cells (RRID:CVCL_Y019) were acquired from Horizon Discovery. Sf9 insect cells were acquired from Thermo Fisher (12659017, RRID:CVCL_0549). |
| Authentication | Authentication was performed upon first arrival in the lab based on morphology according to ATCC. |
| Mycoplasma contamination | All cell lines were routinely tested for mycoplasma contamination. All cell lines were negative throughout the study. |
| Commonly misidentified lines (See ICLAC register) | The cell lines used in this study are not listed as commonly misidentified cell lines. This was verified in the ICLAC table of commonly misidentified cell lines. |

# Plants

| | |
|---|---|
| Seed stocks | N/A |
| Novel plant genotypes | N/A |
| Authentication | N/A |

# Flow Cytometry

## Plots

Confirm that:

☒ The axis labels state the marker and fluorochrome used (e.g. CD4-FITC).

☒ The axis scales are clearly visible. Include numbers along axes only for bottom left plot of group (a 'group' is an analysis of identical markers).

☒ All plots are contour plots with outliers or pseudocolor plots.

☒ A numerical value for number of cells or percentage (with statistics) is provided.

## Methodology

| | |
|---|---|
| Sample preparation | HeLa cells were transduced with lentiviral or retroviral vectors that would express the fluorophore. Cells were treated according the experimental protocol and then collected by removing the medium, washing the cells with 1x PBS (14190169, Thermo Fisher), trypsinisation (T3924, Sigma), and resuspending in complete DMEM medium (41966052, Thermo Fisher). Filtered through 35 µm cell-strainer caps (352235, Falcon) and analysed by an LSR Fortessa Cell Analyzer (BD Biosciences). Lysosomal mt-mKeima was measured using dual excitation ratiometric pH measurements at 405 (pH 7) and 561 (pH 4) nm lasers with 710/50-nm and 610/20-nm detection filters, respectively. Additional channels used for fluorescence compensation were BFP and GFP. Single fluorescence vector expressing cells were prepared to adjust photomultiplier tube voltages to make sure the signal was within detection limits, and to calculate the compensation matrix in BD FACSDiva Software. Depending on the experiment, we gated for BFP-positive, GFP-positive, and mKeima-positive cells with the appropriate compensation. For each sample, 10,000 mKeima-positive events were collected, and data were analyzed in FlowJo (version 10.9.0). |
| Instrument | LSR Fortessa Cell Analyzer (BD Biosciences) |
| Software | BD FACSDiva software during data collection and FlowJo10 software (Tree Star Inc., Ashland, OR, USA) for data analysis. |
| Cell population abundance | Cells were only included when they were viable, single cells (exclusion doublets), and depending on the experiment whether they were GFP-, and/or mt-mKeima positive. |
| Gating strategy | Gating was optimized, depending on the experiment, for GFP- and/or BFP- and mt-mKeima positive cells after viable singlets were separated from potentially dead cells or doublets based on scatter. |

☒ Tick this box to confirm that a figure exemplifying the gating strategy is provided in the Supplementary Information.

