## [Peer Review File · Nature Cell Biology]

Reconstitution of BNIP3/NIX-mitophagy initiation reveals hierarchical flexibility of the autophagy machinery

Corresponding Author: Professor Sascha Martens

Version 0:

Decision Letter:

*Please delete the link to your author homepage if you wish to forward this email to co-authors.

Dear Professor Martens,

Thank you again for submitting your manuscript, "Reconstitution of BNIP3/NIX-mediated autophagy reveals two pathways and hierarchical flexibility of the initiation machinery", to Nature Cell Biology. It has now been seen by 3 referees, who are experts in selective autophagy, mitophagy (Referee #1); selective autophagy (Referee #2); and selective autophagy, ER-phagy (Referee #3). As you will see from their comments (attached below), they found the work of potential interest but have raised substantial concerns, which in our view would need to be addressed with considerable revisions before we can consider publication in Nature Cell Biology.

Nature Cell Biology editors discuss the referee reports in detail within the editorial team, including the chief editor, to identify key referee points that should be addressed with priority, as opposed to requests that are beyond the scope of the current study. To guide the scope of the revisions, I have listed these points below. Our standard revision period is six months, and we are committed to providing a fair and constructive peer-review process, so please feel free to contact me if you would like to discuss any of the referee comments further or if you anticipate any issues or delays addressing the reviews.

In particular, it would be essential to dedicate efforts in revision to the following points:

1- The reviewers requested in-cell validations of key conclusions regarding the order of the recruitment of the proteins during selective autophagy, and we agree that their requests should be addressed rigorously:

Rev#1 paragraphs starting with "The authors conclusions are primarily.."; "All the experiments investigating.."; "Figure 3B – the authors demonstrate.."

Rev#2 paragraph starting with "Second, the authors describe the hierarchy.."

Rev#3 comments #1, #2, #3 (Line 350-351 - regarding Fig 8B)

2- Please also address the reviewers' requests for controls (including requests from Revs#1 and #2 about the BafA1 analyses and Rev#2's comment about the need for controls for kinase and phosphatase activities). All other referee concerns pertaining to strengthening existing data, methodological details, clarifications and textual changes, including scholarly discussion of published work should be addressed.

3- Finally, please pay close attention to our guidelines on statistical and methodological reporting (listed below) as failure to do so may delay the reconsideration of the revised manuscript. In particular, please provide:

We would be happy to consider a revised manuscript that would satisfactorily address these points, unless a similar paper is published elsewhere, or is accepted for publication in Nature Cell Biology in the meantime.

- ensure that it conforms to our format instructions and publication policies (see below and <https://www.nature.com/nature/for-authors>).

- provide a point-by-point rebuttal to the full referee reports verbatim, as provided at the end of this letter.

- provide the completed Reporting Summary (found here <https://www.nature.com/documents/nr-reporting-summary.pdf>). This is essential for reconsideration of the manuscript will be available to editors and referees in the event of peer review. For more information see <http://www.nature.com/authors/policies/availability.html> or contact me.

Nature Cell Biology is committed to improving transparency in authorship. As part of our efforts in this direction, we are now requesting that all authors identified as 'corresponding author' on published papers create and link their Open Researcher and Contributor Identifier (ORCID) with their account on the Manuscript Tracking System (MTS), prior to acceptance. ORCID helps the scientific community achieve unambiguous attribution of all scholarly contributions. You can create and link your ORCID from the home page of the MTS by clicking on 'Modify my Springer Nature account'. For more information please visit www.springernature.com/orcid.

This journal strongly supports public availability of data. Please place the data used in your paper into a public data repository, or alternatively, present the data as Supplementary Information. If data can only be shared on request, please explain why in your Data Availability Statement, and also in the correspondence with your editor. Please note that for some data types, deposition in a public repository is mandatory - more information on our data deposition policies and available repositories appears below.

Link Redacted

We hope that you will find our referees' comments, and editorial guidance helpful. Please do not hesitate to contact me if there is anything you would like to discuss. Thank you again for considering NCB for your work.

Best wishes,

Melina

Melina Casadio, PhD
Senior Editor, Nature Cell Biology
ORCID ID: <https://orcid.org/0000-0003-2389-2243>

Reviewers' Comments:

Reviewer #1 (Remarks to the Author):

In this study, the authors have investigated the mechanism by which mitophagy receptors that are anchored in the outer mitochondrial membrane can initiate autophagosome biogenesis. Specifically, the authors have focused on delineating the process by which various mitophagy receptors, with a more specific focus on NIX and BNIP3, recruit the autophagy machinery to the mitochondria to activate mitophagy. The authors report that these transmembrane cargo receptors can initiate autophagy through either recruiting the FIP200/ULK1 complex or by recruiting a WIPI-ATG13 complex. Their results demonstrate that BCL2L13 recruit FIP200, while NIX and BNIP3 recruit WIPI proteins. In contrast, FKBP8 can use either process, suggesting diverse regulation among these transmembrane receptors. Although this is an interesting study, the novelty of the findings is somewhat modest since a study by Bunker et al (EMBO J, 2023 PMID: 37621214) recently reported

that NIX recruits WIPI to the outer mitochondrial membrane (OMM) to initiate mitophagy. Similar to the current study, Bunker et al also used AlphaFold-2 Multimer to model the NIX-WIPI2 interaction. The same amino acid region was identified and similar mutational analysis to confirm the interaction and function in mitophagy were performed. Some additional experiments are also needed to confirm the interaction between NIX-WIPI2 in cells upon activation of mitophagy. The findings in this study rely on purified proteins and forced anchoring of WIPI2 to the mitochondrial membrane, hence some concerns whether this occurs in the endogenous setting exists. Detailed of concerns are listed below.

-The authors conclusions are primarily based on interactions using purified proteins as well as cell culture experiments where WIPI proteins are forcibly tethered to OMM using the FKBP and FRB system. Because artificially tethering WIPI proteins to the outer mitochondrial membrane leads to induction of mitophagy (as assessed using the MitoKeima reporter), the authors conclude that NIX and BNIP3 induce autophagosome formation by recruiting WIPI to the mitochondria before the FIP200-ULK1 complex. A major concern with this approach is that tethering WIPI to the OMM alone triggers the cascade, thus induction of mitophagy is an artifact of targeting WIPI to OMM and not physiologically relevant. Therefore, the authors still need to confirm that WIPI proteins are recruited to OMM prior to FIP200-ULK1 in response to DFP in cells in a NIX/BNIP3-dependent manner. A time course imaging experiment in live cells (WT vs NIX/BNIP3 DKO) could address the sequence of recruitment.

-All the experiments investigating the protein-protein interactions use purified proteins. To confirm that these interactions occur in vivo (i.e. in cells) upon induction of mitophagy, the authors need to perform co-ip experiments in cells treated with vehicle of DFP.

-Figure 3B – the authors demonstrate that forced localization of WIPI1, WIPI2, and WIPI3 can induce mitophagy. However, they claim that these proteins induce autophagosome biogenesis, but they have little experimental data confirming this. Only MitoKeima is used as a readout which is not a readout of autophagosome formation. It can be used to monitor whether mitochondria are in a normal or acidic environment. Therefore, the authors should confirm formation of autophagosomes and corresponding reduction in mass by performing immunoblotting for LC3 and select mitochondrial proteins under these same conditions. Also, these experiments should also be performed in BNIP3/NIX DKO HeLa cells to confirm that these proteins are no longer needed if WIPI proteins are forcible tethered to OMM. Presumably this would be the case.

-Fig. 3C – based on the data, the authors state that Bafilomycin A1 treatment completely inhibited mitochondrial turnover and that this confirms that tethering WIPI1, WIPI2, WIPI3 to the mitochondria surface is sufficient to induce mitophagy. This might not necessarily be true. Bafilomycin A1 is not an inhibitor of autophagosome formation but inhibits the v-ATPase on the lysosome which disrupts acidification. This means that autophagosomes can still form in the presence of BafA1. This experiment only confirms that the degradation of mitochondria is dependent of lysosomes.

-Fig. 7C-D – Were experiment using OA to induce mitophagy performed in cells overexpressing Parkin? It is not clear from the legend. Parkin overexpression is only mentioned for Fig. 7E.

Reviewer #2 (Remarks to the Author):

Selective autophagy is crucial for maintaining cellular homeostasis by degrading damaged or superfluous organelles, but how transmembrane cargo receptors initiate this process is not well understood. This study reveals that some transmembrane receptors, like BNIP3/NIX, can trigger autophagy via interaction with a WIPI-ATG13 complex, which appears to be a distinct mechanism when compared with soluble ubiquitin binding autophagy receptors. These findings highlight the flexibility of autophagy initiation and offer insights for potential therapeutic applications. In addition, this paper takes a step towards a long-term goal of the field to fully reconstitute organelle autophagic capture in a test tube.

Overall, the paper is very nice and will make an important contribution to the literature. Most of the experiments are well done and well controlled.

I would have two general comments. First, the title indicating “Reconstitution of BNIP3/NIX-mediated autophagy” seems too strong. The way that is worded, it conjures up the full-scale assembly of autophagosomes around mitochondria in vitro, which has not been performed in the paper. In parts of the text, the phrase is more measured – like “reconstitute initiation”. When I first read the title, I was like – wow - this must be amazing, but then it became clear that full reconstitution of mitophagy was not really achieved in this case. Not to take anything away from the work but expectations were very high based on the title.

Second, the authors describe the hierarchy in the discussion with WIPI recruitment being an initiating event and the other machinery coming subsequently. But I wonder whether this is really the case under normal conditions. It may be that for some reason, the binding reactions for FIP200 or other machinery is missing some modification or factor that is required: i.e. you still need all of the upstream pathways AND the WIPI interactions with the receptor, but the order of WIPI binding first isn't required in vivo. Many of the experiments trying to demonstrate such a model use overexpression of WIPIs and artificial recruitment to mitochondria and it may be that the copy number of molecules under these conditions can drive the assembly essentially in reverse order – i.e. you still need the ATG13 interaction to get productive mitophagy but the order isn't WIPI binding first under endogenous protein concentrations. It seems like it would be worth mentioning this caveat when discussing the model. Although there are clearly many modules and ways to recruit the various factors, as these authors have demonstrated now for several different receptor assemblies, it could be that the majority of receptor-driven autophagy

pathways build a large “solid-state” assembly with many modular interactions that all fit together for efficient activation/recruitment of LC3 etc, but sub-assemblies can enter in various combinations or order rather than a strict hierarchy. I think of this like puzzle pieces (each piece being a different protein) that interlock together; you can build a section of the puzzle coming from all different directions, but the ultimate solution is always the same and the picture looks identical regardless of which puzzle pieces were added first. There might be data in the paper that would argue against this possibility but it was hard for me to tell for sure if one or more experiments exclude such an interpretation. The detailed knowledge of these investigators of the various modules within the system should allow them to comment on whether the order of interaction proposed is truly a requirement under conditions of endogenous protein concentrations.

A few specific comments

Line 94: at face value, the experiments with kinases do not have positive controls that the kinases were active in the experiment. Similarly, for the lambda phosphatase experiment, it is possible that any phosphosites present on the target proteins are not dephosphorylated by the phosphatase or that the proteins were not phosphorylated in the first place. Perhaps the size of the proteins would allow analysis of phospho tags to see if there is any phosphatase collapsible phosphorylation, or any detection of multiple forms of the recombinant proteins.

In the introduction (paragraph 4), it might be worth mentioning the potential role of membrane receptor ubiquitylation (as for FAM134), as one aspect of mechanism that has been proposed recently (PMID: 37225996). Ubiquitin-driven clustering, which may be operative in some cases but not universal

Line 115 – would suggest “and any of the tested upstream autophagy machinery”

Line 182 – The use of BafA as an autophagy inhibitor, especially for long periods of time as in Fig 3C is probably not the best approach. There is evidence of expulsion of endolysosomal vesicles and significant disruption of the pathway with as much as 8 hours of BafA treatment. Later in the paper, the authors demonstrate blockade by for example VPS34 inhibitor, so this is probably a better experiment.

Line 219 – might be worth mentioning that other forms of mitophagy (i.e. Parkin dependent) – require TBK1 in certain contexts (PMID: 26365381, PMID: 37207627, PMID: 26266977).

Line 276 – The AF multimer analysis in this section doesn't have the needed documentation of AF scores, etc, as was provided in some of the other AF sections (including in the supplemental figures). How good are the scores for the ATG13 analysis?

Fig 7 addresses the interaction of other membrane-embedded receptors with WIPI initially using AF, with reasonable AF scores with TEX264 and FAM134C. A previous study has already demonstrated a possible interaction WIPI2 with TEX264 based on APEX of TEX264 (see Ref 29). In fact, it is one of the strongest hits in the proteomics. This result could be references as in cell evidence for the interactions predicted and examined in vitro.

Reviewer #3 (Remarks to the Author):

The BNIP3/NIX-mediated autophagy pathway is involved in the quality control of mitochondria, peroxisome, and in some cases, the endoplasmic reticulum. While the downstream events of BNIP3/NIX-mediated autophagy pathway have been well characterised, the upstream recruitment of general autophagy machineries has been assumed to follow the canonical autophagy activation dogma. In this work, the authors present unexpected data showing that BNIP3 and NIX do not directly recruit ULK1 complex, but instead, first recruits WIPI2/3 that bridges the ULK1 complex with BNIP3/NIX. Overall, this is a high-quality manuscript with elegant biochemical data supported by adequate cell biology observations. The manuscript is also well-written. The observations are also very timely and will be important to the BNIP/NIX-autophagy field. I only have a few questions and comments.

Comment 1 (Fig4F-I): The authors claim that WIPI2 initiates autophagosome formation together with ATG13, but still require ULK1-complex downstream to initiate mitophagy. This is demonstrated in Figure 4F-I, where depletion of ULK1 complex components or chemical inhibition of ULK1 causes inhibition of mitophagy. However, this data does not show whether WIPI2 is still recruited to mitochondria in these conditions. This leaves open the possibility of ULK1 inhibition having impact on WIPI2 recruitment. Therefore it would be important to demonstrate by immunofluorescence that WIPI2/3 recruitment to the mitochondria is not perturbed upon ULK1 inhibition.

Comment 2 (Fig 4A): Along the same line, while the ATG13 recruitment to mitochondria was shown to be dependent on WIPI2 in Fig 4A, that recruitment is induced using the artificial rapalog system. It will be important to show that upon DFP treatment, this recruitment is also dependent on WIPI2/3 using knockdown/knockout of WIPI2 and WIPI3.

Comment 3 (Line 350-351 – regarding Fig 8B): This is a generalisation based on the BNIP3/NIX observation which I think is a very strong statement. The authors would need to demonstrate cell biology that WIPI2/3 depletion has an impact on ER-phagy.

Comment 3 (Fig 8E): While the data does show that both FIP200 and WIPI2d can bind TEX264/FKBP8 in an in vitro

biochemical setting, it does not necessarily support the statement that TEX264 and FKBP8 can bind both FIP200 and WIPI2d at the same time. The authors should discuss the possibility of competitive binding, sequential binding, or FIP200 and WIPI2d being recruited to these receptors preferentially depending on physiological context.

Minor comments:

1. I believe Row 164's reference to Fig 2I should be about Fig 2L
2. Please double check the labelling on Fig 7A, I believe the +/- signs on top of the blots are mislabelled

ABSTRACT AND MAIN TEXT – please follow the guidelines that are specific to the format of your manuscript, as listed in our Guide to Authors (http://www.nature.com/ncb/pdf/nbc_gta.pdf) Briefly, Nature Cell Biology Articles, Resources and Technical Reports have 3500 words, including a 150 word abstract, and the main text is subdivided in Introduction, Results, and Discussion sections. Nature Cell Biology Letters have up to 2500 words, including a 180 word introductory paragraph (abstract), and the text is not subdivided in sections.

REFERENCES – are limited to a total of 70 for Articles, Resources, Technical Reports; and 40 for Letters. This includes references in the main text and Methods combined. References must be numbered sequentially as they appear in the main text, tables and figure legends and Methods and must follow the precise style of Nature Cell Biology references. References only cited in the Methods should be numbered consecutively following the last reference cited in the main text. References only associated with Supplementary Information (e.g. in supplementary legends) do not count toward the total reference limit and do not need to be cited in numerical continuity with references in the main text. Only published papers can be cited, and each publication cited should be included in the numbered reference list, which should include the manuscript titles.

Footnotes are not permitted.

Methods should be written concisely, but should contain all elements necessary to allow interpretation and replication of the results. As a guideline, Methods sections typically do not exceed 3,000 words. The Methods should be divided into subsections listing reagents and techniques. When citing previous methods, accurate references should be provided and any alterations should be noted. Information must be provided about: antibody dilutions, company names, catalogue numbers and clone numbers for monoclonal antibodies; sequences of RNAi and cDNA probes/primers or company names and catalogue numbers if reagents are commercial; cell line names, sources and information on cell line identity and authentication. Animal studies and experiments involving human subjects must be reported in detail, identifying the committees approving the protocols. For studies involving human subjects/samples, a statement must be included confirming that informed consent was obtained. Statistical analyses and information on the reproducibility of experimental results should be provided in a section titled "Statistics and Reproducibility".

All Nature Cell Biology manuscripts submitted on or after March 21 2016 must include a Data availability statement as a separate section after Methods but before references, under the heading "Data Availability". For Springer Nature policies on data availability see <http://www.nature.com/authors/policies/availability.html>; for more information on this particular policy see <http://www.nature.com/authors/policies/data/data-availability-statements-data-citations.pdf>. The Data availability statement should include:

- Accession codes for primary datasets (generated during the study under consideration and designated as "primary accessions") and secondary datasets (published datasets reanalysed during the study under consideration, designated as "referenced accessions"). For primary accessions data should be made public to coincide with publication of the manuscript. A list of data types for which submission to community-endorsed public repositories is mandated (including sequence, structure, microarray, deep sequencing data) can be found here <http://www.nature.com/authors/policies/availability.html#data>.
- Unique identifiers (accession codes, DOIs or other unique persistent identifier) and hyperlinks for datasets deposited in an approved repository, but for which data deposition is not mandated (see here for details <http://www.nature.com/sdata/data-policies/repositories>).
- At a minimum, please include a statement confirming that all relevant data are available from the authors, and/or are included with the manuscript (e.g. as source data or supplementary information), listing which data are included (e.g. by figure panels and data types) and mentioning any restrictions on availability.
- If a dataset has a Digital Object Identifier (DOI) as its unique identifier, we strongly encourage including this in the Reference list and citing the dataset in the Methods.

We recommend that you upload the step-by-step protocols used in this manuscript to [protocols.io](https://www.protocols.io). More details can be found at <https://www.protocols.io/help/publish-articles>.

All imaging data should be accompanied by scale bars, which should be defined in the legend.

Cropped images of gels/blots are acceptable, but need to be accompanied by size markers, and to retain visible background signal within the linear range (i.e. should not be saturated). The boundaries of panels with low background have to be demarcated with black lines. Splicing of panels should only be considered if unavoidable, and must be clearly marked on the figure, and noted in the legend with a statement on whether the samples were obtained and processed simultaneously. Quantitative comparisons between samples on different gels/blots are discouraged; if this is unavoidable, it should only be performed for samples derived from the same experiment with gels/blots were processed in parallel, which needs to be stated in the legend.

The total number of Supplementary Figures (not including the "unprocessed scans" Supplementary Figure) should not exceed the number of main display items (figures and/or tables (see our Guide to Authors and March 2012 editorial <http://www.nature.com/ncb/authors/submit/index.html#suppinfo>; <http://www.nature.com/ncb/journal/v14/n3/index.html#ed>). No restrictions apply to Supplementary Tables or Videos, but we advise authors to be selective in including supplemental data.

GUIDELINES FOR EXPERIMENTAL AND STATISTICAL REPORTING

REPORTING REQUIREMENTS – We are trying to improve the quality of methods and statistics reporting in our papers. To that end, we are now asking authors to complete a reporting summary that collects information on experimental design and reagents. The Reporting Summary can be found here <https://www.nature.com/documents/nr-reporting-summary.pdf> If you would like to reference the guidance text as you complete the template, please access these flattened versions at <http://www.nature.com/authors/policies/availability.html>.

Version 1:

Decision Letter:

Our ref: NCB-A55430A

20th March 2025

Dear Dr. Martens,

Thank you for submitting your revised manuscript "Reconstitution of BNIP3/NIX-mitophagy initiation reveals hierarchical flexibility of the autophagy machinery" (NCB-A55430A). It has now been seen by the original referees and their comments are below. The reviewers find that the paper has improved in revision, and therefore we'll be happy in principle to publish it in Nature Cell Biology, pending minor revisions to satisfy the referees' final requests and to comply with our editorial and formatting guidelines.

If the current version of your manuscript is in a PDF format, please email us a copy of the file in an editable format (Microsoft Word or LaTeX)– we can not proceed with PDFs at this stage.

Thank you again for your interest in Nature Cell Biology Please do not hesitate to contact me if you have any questions.

Sincerely,

Angela R Parrish, PhD
Locum Senior Editor
Nature Cell Biology

Reviewer #1 (Remarks to the Author):

The authors have done a good job addressing my concerns. No further comments.

Reviewer #2 (Remarks to the Author):

The authors have done a great job of addressing the reviewer's criticisms. This is an impactful paper that sets the standard for elucidating mechanisms of modules within the selective autophagy pathway. In my view, this is an exciting paper for its clarity and depth of analysis.

Reviewer #3 (Remarks to the Author):

In this re-submission, the authors have sufficiently addressed the concerns with additional experimental work and text revision. I have no further comments

Version 2:

Decision Letter:

Dear Sascha,

I am very sorry for the delay in officially accepting your manuscript.

I am pleased to inform you that your manuscript, "Reconstitution of BNIP3/NIX-mitophagy initiation reveals hierarchical flexibility of the autophagy machinery", has now been accepted for publication in Nature Cell Biology.

Please note that *Nature Cell Biology* is a Transformative Journal (TJ). Authors may publish their research with us through the traditional subscription access route or make their paper immediately open access through payment of an article-processing charge (APC). Authors will not be required to make a final decision about access to their article until it has been accepted. <https://www.springernature.com/gp/open-research/transformative-journals> Find out more about Transformative Journals

Authors may need to take specific actions to achieve

research/funding/policy-compliance-faqs"> compliance with funder and institutional open access mandates. If your research is supported by a funder that requires immediate open access (e.g. according to Plan S principles) then you should select the gold OA route, and we will direct you to the compliant route where possible. For authors selecting the subscription publication route, the journal's standard licensing terms will need to be accepted, including self-archiving policies. Those licensing terms will supersede any other terms that the author or any third party may assert apply to any version of the manuscript.

If you have not already done so, we strongly recommend that you upload the step-by-step protocols used in this manuscript to protocols.io (<https://protocols.io>), an open online resource that allows researchers to share their detailed experimental know-how. All uploaded protocols are made freely available and are assigned DOIs for ease of citation. Protocols and Nature Portfolio journal papers in which they are used can be linked to one another, and this link is clearly and prominently visible in the online versions of both. Authors who performed the specific experiments can act as primary authors for the Protocol as they will be best placed to share the methodology details, but the Corresponding Author of the present research paper should be included as one of the authors. By uploading your Protocols onto protocols.io, you are enabling researchers to more readily reproduce or adapt the methodology you use, as well as increasing the visibility of your protocols and papers. You can also establish a dedicated workspace to collect your lab Protocols. Further information can be found at <https://www.protocols.io/help/publish-articles>.

Nature Cell Biology encourages authors presenting evidence for cell, biological, molecular, and genetic interactions to consider communicating these findings using Biofactoid (<https://biofactoid.org/>). This tool helps users share a searchable representation of interactions (e.g. binding, gene expression, post-translational modification) between genes, gene products, or chemicals. Information added to Biofactoid, with author attribution, is shared on social media and public databases, such as Pathway Commons, where it can be discovered and analyzed in the context of a large and growing corpus of knowledge.

With kind regards,

Angela R Parrish, PhD
Locum Senior Editor
Nature Cell Biology

** Visit the Springer Nature Editorial and Publishing website at www.springernature.com/editorial-and-publishing-jobs for more information about our career opportunities. If you have any questions please click here.**

Reviewers' Comments:

Reviewer #1 (Remarks to the Author):

In this study, the authors have investigated the mechanism by which mitophagy receptors that are anchored in the outer mitochondrial membrane can initiate autophagosome biogenesis. Specifically, the authors have focused on delineating the process by which various mitophagy receptors, with a more specific focus on NIX and BNIP3, recruit the autophagy machinery to the mitochondria to activate mitophagy. The authors report that these transmembrane cargo receptors can initiate autophagy through either recruiting the FIP200/ULK1 complex or by recruiting a WIPI-ATG13 complex. Their results demonstrate that BCL2L13 recruit FIP200, while NIX and BNIP3 recruit WIPI proteins. In contrast, FKBP8 can use either process, suggesting diverse regulation among these transmembrane receptors. Although this is an interesting study, the novelty of the findings is somewhat modest since a study by Bunker et al (EMBO J, 2023 PMID: 37621214) recently reported that NIX recruits WIPI to the outer mitochondrial membrane (OMM) to initiate mitophagy. Similar to the current study, Bunker et al also used AlphaFold-2 Multimer to model the NIX-WIPI2 interaction. The same amino acid region was identified and similar mutational analysis to confirm the interaction and function in mitophagy were performed. Some additional experiments are also needed to confirm the interaction between NIX-WIPI2 in cells upon activation of mitophagy. The findings in this study rely on purified proteins and forced anchoring of WIPI2 to the mitochondrial membrane, hence some concerns whether this occurs in the endogenous setting exists. Detailed of concerns are listed below.

We would like to thank the reviewer for taking the time to read our manuscript carefully. We are grateful for the detailed comments described below, which allowed us to strengthen our manuscript considerably. However, we would disagree with the reviewer's statement that '*the novelty of the findings is somewhat modest since a study by Bunker et al...*'. While the Youle lab elegantly demonstrated that NIX can bind to WIPI2, our work provides substantial progress:

- (i) Our work demonstrates that full-length NIX and BNIP3 can only drive autophagy when able to bind WIPI2.
- (ii) Perhaps even more importantly, our work demonstrates that this is the driving mechanism for NIX/BNIP3 mitophagy, as they cannot bind FIP200. This aspect was not covered in the Youle study but provides a major step forward, as it provides insight into the importance of recruiting WIPI2. In particular, we show that WIPI2 compensates for the inability of NIX/BNIP3 to recruit FIP200 directly.
- (iii) We then continue by showing that WIPI recruitment can be sufficient to initiate autophagosome biogenesis, also not shown by the Youle lab.
- (iv) We show how the recruitment of WIPIs leads to the recruitment of the ULK1 complex, which is also essential for NIX/BNIP3 mitophagy and also not shown by the Youle study.
- (v) Through these obtained mechanistical insights, our work also provides new tools to discriminate PINK1/Parkin from BNIP3/NIX mitophagy with targeted ATG13 mutants.
- (vi) Finally, we demonstrate that other selective autophagy receptors, such as ER-phagy receptor TEX264 and mitophagy receptor FKBP8 can also employ this WIPI-ATG13 mechanism.

For these reasons, we believe that while our work is in line with the findings in the Youle study, we do go well beyond those findings and provide molecular explanations and mechanisms for the contribution of WIPIs to mitophagy. Altogether, we think our work provides new insights into how the WIPI-ATG13 complex can function as an alternative to the FIP200/ULK1 complex, an unexpected finding that could not have been predicted from the Youle study. Moreover, we provide a detailed mechanistic understanding of this new WIPI-ATG13 axis and for which we show that it might be shared by several selective autophagy receptors (across different organelles).

-The authors conclusions are primarily based on interactions using purified proteins as well as cell culture experiments where WIPI proteins are forcibly tethered to OMM using the FKBP and FRB system. Because artificially tethering WIPI proteins to the outer mitochondrial membrane leads to induction of mitophagy (as assessed using the MitoKeima reporter), the authors conclude that NIX and BNIP3 induce autophagosome formation by recruiting WIPI to the mitochondria before the FIP200-ULK1 complex. A major concern with this approach is that tethering WIPI to the OMM alone triggers the cascade, thus induction of mitophagy is an artifact of targeting WIPI to OMM and not physiologically relevant. Therefore, the authors still need to confirm that WIPI proteins are recruited to OMM prior to FIP200-ULK1 in response to DFP in cells in a NIX/BNIP3-dependent manner. A time course imaging experiment in live cells (WT vs NIX/BNIP3 DKO) could address the sequence of recruitment.

We thank the reviewer for this comment and have addressed this experimentally in the following way: we tested whether WIPI2 could be recruited to the OMM in FIP200 knockout cells and compared this to WT cells. Based on our model, if WIPI2 is recruited upstream of FIP200, then the accumulation of WIPI2 on the OMM should not be affected by the presence or absence of FIP200.

The activity of BNIP3/NIX is dampened by PPTC7, which complicates imaging of BNIP3/NIX-driven mitophagy as a relatively slow and sporadic event. We, therefore, depleted PPTC7 by siRNA and could observe that WIPI2 was recruited to the OMM in wild-type cells upon mitophagy induction with DFP. We could also observe autophagosome-like circular structures engulfing mitochondrial fragments, consistent with the idea that those cells are undergoing mitophagy. Importantly, we also detected WIPI2 recruitment to the OMM in FIP200 knockout cells, indicating that WIPI2 acts upstream of FIP200 and is consistent with the model we propose in this manuscript. In fact, we could even see a clearer accumulation of WIPI2 on the OMM in FIP200 knockout cells compared to WT cells. This is consistent with our other data showing that although downstream of WIPI2, FIP200 remains an essential factor for BNIP3/NIX mitophagy. So, the observation that WIPI2 accumulates on the OMM in FIP200 knockout cells indicates that BNIP3/NIX recruit WIPI2 upstream of FIP200 but that mitophagy cannot be completed due to the lack of FIP200, explaining the phenotype of WIPI2 accumulation. In FIP200 knockout cells, we also no longer observed these circular autophagosome-like structures that engulfed pieces of the mitochondrial network, further indicating that mitophagy cannot be completed in the FIP200 cells.

In addition to this experiment, and as requested by Reviewer 3, we also performed these experiments in wild-type cells treated with the ULK1 inhibitor. According to our model, ULK1 remains essential for NIX/BNIP3 mitophagy and its inhibition would thus be expected to phenocopy the FIP200 knockout cells, leading to the accumulation of WIPI2 (as it acts upstream of FIP200), but no formation of autophagosome-like structures. Indeed, this is exactly what we observe in WT cells that are treated with an ULK1-kinase inhibitor: absence of autophagosome-like structures, but still recruitment and accumulation of WIPI2 on the mitochondrial surface.

We have added these data to Figure 4J-K and the supplementary figure 6.

Figure 4

Extended data Figure 6

-All the experiments investigating the protein-protein interactions use purified proteins. To confirm that these interactions occur in vivo (i.e. in cells) upon induction of mitophagy, the authors need to perform co-ip experiments in cells treated with vehicle of DFP.

We thank the reviewer for this suggestion and have performed the following co-immunoprecipitation experiments:

(i) NIX-WIPI2 interaction:

(ii) TEX264-WIPI2 interaction:

We would also like to point to some very elegant work from Wade Harper's lab. They previously observed that WIPI2 is the top hit in their proximity labeling experiments for TEX264-APEX2 in cells (An et al. 2019 Mol Cell). This is consistent with our findings that TEX264 can directly bind to WIPI2. Moreover, as you can read below in response to Reviewer 3 (comment 3), we also observed that TEX264 is no longer turned over upon induction of ER-phagy in the absence of WIPI2, while other ER-phagy receptors like CCPG1 are turned over normally. This further supports that the protein-protein interaction identified for TEX264-WIPI2 is important in cells.

-Figure 3B – the authors demonstrate that forced localization of WIPI1, WIPI2, and WIPI3 can induce mitophagy. However, they claim that these proteins induce autophagosome biogenesis, but they have little experimental data confirming this. Only MitoKeima is used as a readout which is not a readout of autophagosome formation. It can be used to monitor whether mitochondria are in a normal or acidic environment. Therefore, the authors should confirm formation of autophagosomes and corresponding reduction in mass by performing immunoblotting for LC3 and select mitochondrial proteins under these same conditions. Also, these experiments should also be performed in BNIP3/NIX DKO HeLa cells to confirm that these proteins are no longer needed if WIPI proteins are forcibly tethered to OMM. Presumably this would be the case.

We thank the reviewer for this comment and agree that these experiments would strengthen our conclusion. We have, therefore, performed several experiments that together provide further evidence that tethering of WIPI proteins induces mitophagy.

First, as suggested by the reviewer, we tested whether rapalog-induced tethering of the different FKBP-WIPIs leads to LC3B lipidation and turnover of mitochondrial substrates. Indeed, tethering does not only lead to LC3B lipidation, but it also results in turnover of mitochondrial proteins like COXII. Note that these experiments were performed without bafilomycin to allow for the lysosomal turnover of mitochondrial proteins.

We then repeated the tethering of FKBP-GFP-WIPI2 in NIX/BNIP3 knockout (2KO) cells and confirmed that this also leads to LC3B lipidation and COXII degradation. Consistently, we also repeated the mt-mKeima FACS assay in those cells and observed robust mitophagy induction.

Finally, we also sent triplicate samples for mass spec analysis to obtain a more global picture of protein turnover. This revealed that just as in wild-type HeLa cells treated with DFP, tethering of FKBP-GFP-WIPI2 resulted in the degradation of MitoCarta 3.0 annotated mitochondrial proteins. Moreover, this was the case in both HeLa wild-type cells and NIX/BNIP3 double knockout (2KO) HeLa cells.

Together, these data demonstrate that tethering of WIPI proteins to the mitochondrial surface leads to mitophagy induction, as confirmed by several complementary methods:

- (i) mt-mKeima FACS assays,
- (ii) LC3B lipidation and COXII degradation visualized by western blotting,
- (iii) Global proteome analysis with mass spectrometry.

-Fig. 3C – based on the data, the authors state that Bafilomycin A1 treatment completely inhibited mitochondrial turnover and that this confirms that tethering WIPI1, WIPI2, WIPI3 to the mitochondria surface is sufficient to induce mitophagy. This might not necessarily be true. Bafilomycin A1 is not an inhibitor of autophagosome formation but inhibits the v-ATPase on the lysosome which disrupts acidification. This means that autophagosomes can still form in the presence of BafA1. This experiment only confirms that the degradation of mitochondria is dependent of lysosomes.

The reviewer rightfully points out that the VPS34 inhibitor would have been a better approach to address our question. We are therefore grateful for the reviewer’s comment, and we have now replaced the experiment in Fig. 3C with the same experimental set-up, however this time by using the Vps34-inhibitor. This showed that mitochondrial degradation is inhibited by the VPS34-inhibitor. We have therefore changed the text in the manuscript into:

To confirm that this mitochondrial turnover was mediated by autophagy, we repeated the experiment for WIPI1, WIPI2, and WIPI3 in the presence of a VPS34 kinase-inhibitor, a component of the PI3KC3-C1 complex (composed of VPS34, VPS15, Beclin1, and ATG14), which blocks autophagosome formation (Fig. 3c). VPS34-inhibitor treatment completely inhibited mitochondrial turnover, confirming that tethering WIPI1, WIPI2, WIPI3 to the mitochondrial surface is sufficient to induce mitophagy.

-Fig. 7C-D – Were experiment using OA to induce mitophagy performed in cells overexpressing Parkin? It is not clear from the legend. Parkin overexpression is only mentioned for Fig. 7E.

We thank the reviewer for this question and apologize that this was insufficiently clear from the submitted manuscript. All experiments in Figure 7, where we treated the cells with O/A were performed in HeLa cells stably overexpressing Parkin. We have now specified this better in the figure legend.

Reviewer #2 (Remarks to the Author):

Selective autophagy is crucial for maintaining cellular homeostasis by degrading damaged or superfluous organelles, but how transmembrane cargo receptors initiate this process is not well understood. This study reveals that some transmembrane receptors, like BNIP3/NIX, can trigger autophagy via interaction with a WIPI-ATG13 complex, which appears to be a distinct mechanism when compared with soluble ubiquitin binding autophagy receptors. These findings highlight the flexibility of autophagy initiation and offer insights for potential therapeutic applications. In addition, this paper takes a step towards a long-term goal of the field to fully reconstitute organelle autophagic capture in a test tube.

Overall, the paper is very nice and will make an important contribution to the literature. Most of the experiments are well done and well controlled.

I would have two general comments. First, the title indicating “Reconstitution of BNIP3/NIX-mediated autophagy” seems too strong. The way that is worded, it conjures up the full-scale assembly of autophagosomes around mitochondria *in vitro*, which has not been performed in the paper. In parts of the text, the phrase is more measured – like “reconstitute initiation”. When I first read the title, I was like – wow - this must be amazing, but then it became clear that full reconstitution of mitophagy was not really achieved in this case. Not to take anything away from the work but expectations were very high based on the title.

Second, the authors describe the hierarchy in the discussion with WIPI recruitment being an initiating event and the other machinery coming subsequently. But I wonder whether this is really the case under normal conditions. It may be that for some reason, the binding reactions for FIP200 or other machinery is missing some modification or factor that is required: i.e. you still need all of the upstream pathways AND the WIPI interactions with the receptor, but the order of WIPI binding first isn't required *in vivo*. Many of the experiments trying to demonstrate such a model use overexpression of WIPIs and artificial recruitment to mitochondria and it may be that the copy number of molecules under these conditions can drive the assembly essentially in reverse order – i.e. you still need the ATG13 interaction to get productive mitophagy but the order isn't WIPI binding first under endogenous protein concentrations. It seems like it would be worth mentioning this caveat when discussing the model. Although there are clearly many modules and ways to recruit the various factors, as these authors have demonstrated now for several different receptor assemblies, it could be that the majority of receptor-driven autophagy pathways build a large “solid-state” assembly with many modular interactions that all fit together for efficient activation/recruitment of LC3 etc, but sub-assemblies can enter in various combinations or order rather than a strict hierarchy. I think of this like puzzle pieces (each piece being a different protein) that interlock together; you can build a section of the puzzle coming from all different directions, but the ultimate solution is always the same and the picture looks identical regardless of which puzzle pieces were added first. There might be data in the paper that would argue against this possibility but it was hard for me to tell for sure if one or more experiments exclude such an interpretation. The detailed knowledge of these investigators of the various modules within the system should allow them to comment on whether the order of interaction proposed is truly a requirement under conditions of endogenous protein concentrations.

We thank the reviewer for taking the time to read our manuscript carefully. We appreciate the kind words about our work and are grateful for the comments included below, which have allowed us to strengthen our manuscript considerably.

Regarding the title, we apologize that the submitted title may have suggested that we had reconstituted the entire mitophagy cascade. As the reviewer correctly notes, we did not attempt to make this claim but may have unintendedly insinuated this with the submitted title. We are therefore grateful for the

reviewer pointing this out, and we have adjusted the title as follows: *Reconstitution of BNIP3/NIX-mitophagy initiation reveals hierarchical flexibility of the autophagy machinery*. Note, however, that by adding 'initiation', we were over the character limit and therefore removed 'two pathways' from the title.

Regarding the reviewer's comment on the order of recruitment, we absolutely agree with the reviewer's view on how the autophagy machinery assembles. In the past years, a significant number of protein-protein interactions between different complexes of the autophagy machinery have been identified. Based on these findings, and as the reviewer elegantly describes, a picture has emerged that the machinery forms different puzzle pieces that interlock together, providing a potential explanation for why random encounters of these protein complexes in the cytosol may not trigger the formation of autophagosomes at non-intended spots across the cytosol. In particular, since many of these protein-protein interactions between the different components are individually relatively weak, but due to a mechanism of mutual stabilization, it leads to the productive formation of autophagosome biogenesis, for which each component is required. However, this also implies that the recruitment and clustering of the machinery at the cargo material, where the different pieces of the puzzle come together and lead to the productive formation of an autophagosome, forms a critical step in selective autophagy. As the selective autophagy receptors orchestrate this exact step, it suggests that, in this model, the cargo receptors play an essential role as they indicate where autophagosome biogenesis is to be initiated by placing the first piece of the puzzle. Hence, the recruitment of the different autophagy protein complexes to the surface of the to-be degraded cargo forms a critical step, and our results now reveal that the cargo receptors may not only initiate this cascade of events by first recruiting the FIP200/ULK1 complex, but also by first recruiting the WIPI/ATG13 complex. We, therefore, agree with how the reviewer thinks about assembling the machinery and believe that our work adds another important piece to that model by demonstrating that, also at the level of the cargo receptors, there might be more than one entry point. This could be particularly important for developing therapeutics if specific selective autophagy pathways get stalled; alternative entry points could be exploited to compensate for the defective autophagy pathways.

A few specific comments

Line 94: at face value, the experiments with kinases do not have positive controls that the kinases were active in the experiment. Similarly, for the lambda phosphatase experiment, it is possible that any phosphosites present on the target proteins are not dephosphorylated by the phosphatase or that the proteins were not phosphorylated in the first place. Perhaps the size of the proteins would allow analysis of phostag gels to see if there is any phosphatase collapsable phosphorylation, or any detection of multiple forms of the recombinant proteins.

We thank the reviewer for this comment. While we had performed extensive validation of the kinase/phosphatase activity of our purified components prior to initiating these experiments (on known targets of the kinases/phosphatases), we agree with the reviewer that our manuscript did not contain evidence for the activity of the kinases/phosphatases in the assay on BNIP3/NIX. We have therefore added the control experiments to demonstrate that the kinases/phosphatase were active.

First of all, each of the kinases/phosphatase was purified to high purity (as also displayed in Figure S2A) and displayed the expected profile in the gel filtration runs, indicative of a correctly folded protein. We then continued with more specific assays to validate the activity of each kinase/phosphatase.

For TBK1, we demonstrated that it is very active towards its known substrates OPTN and NAP1, as shown below. Please note that the blot for OPTN was added to this manuscript as Figure S2D, while the blot for NAP1 was already published in Adriaenssens et al. 2024 NSMB (PMID: 38918639).

For ULK1, we demonstrated that the kinase is very active towards known substrates, such as Beclin1. Please note that the blot for Beclin1 was added to this manuscript as Figure S2E.

For CK2, this kinase was newly purified during this study and subjected to the following controls. We first showed that the purified kinase was active towards a peptide with a known CK2 motif, as shown in Figure S2B, and compared it head-to-head with a commercially available version of CK2. This revealed that our purified CK2 was very active towards this peptide substrate, considerably more active than the commercially available control protein. The activity was specific, as we could completely block it by adding the known kinase inhibitor CX4945.

For SRC, this kinase was also newly purified during this study and subjected to the following controls. We first showed that the purified kinase was active towards a peptide with a known SRC motif, as shown in Figure S2C. Consistent with the literature, the introduction of the Y530F mutation converted the kinase into a constitutively active form, as demonstrated by our kinase assay with a known peptide substrate, which demonstrated that the SRC-Y530F was several times more active compared to its WT counterpart.

For Lambda phosphatase, we incubated the recombinant phosphatase with HeLa cell lysate for 30 minutes and immunoblotted for a known phosphorylation substrate 4EBP1. Addition of Lambda phosphatase readily removed the phosphorylation on 4EBP1, as demonstrated by the downward shift on the SDS-PAGE gel (Figure S2F).

30degrees for 30 minutes

While all the above assays are consistent with our kinases/phosphatases being active, we agree with the reviewer that these are different substrates and hence the kinases/phosphatases may respond differently towards our substrates of interest BNIP3/NIX. Hence, we performed additional control experiments, using mass spectrometry as a read out, to assess the activity of our kinases/phosphatase towards NIX.

To this end, we incubated NIX-GST together with TBK1, MBP-ULK1, SRC-Y530F, or CK2, at concentration ratios identical to our bead assay experiments. This revealed that NIX is unlikely to be a bona fide substrate for TBK1, ULK1, and SRC-Y530F as our in vitro kinase assays yielded only

spurious phosphorylation events of the different kinases on NIX (further suggesting the kinases are active), but far too low numbers if the kinase/substrate interaction would be specific.

For CK2, however, we observed substantial phosphorylation events and can conclude that NIX is a substrate of CK2. The phosphorylation sites identified were also conserved between BNIP3 and NIX.

These findings are consistent with our cellular experiments, which revealed that TBK1 does not regulate BNIP3/NIX mitophagy. Moreover, in some of our unpublished work, we also observed a role for CK2 in regulating BNIP3/NIX mitophagy, consistent with our mass spectrometry data from above. However, as we are currently working on a subsequent study where we are dissecting the role of CK2 in BNIP3/NIX mitophagy, we decided to leave these results about the specific phosphorylation residues for our next manuscript and limit this study to the experimental validation of the kinases as displayed in Figure S2.

In the introduction (paragraph 4), it might be worth mentioning the potential role of membrane receptor ubiquitylation (as for FAM134), as one aspect of mechanism that has been proposed recently (PMID: 37225996). Ubiquitin-driven clustering, which may be operative in some cases but not universal.

We thank the reviewer for this excellent suggestion and have incorporated this into our introduction section.

Line 115 – would suggest “and any of the tested upstream autophagy machinery”

We thank the reviewer for this excellent suggestion and have incorporated this into our revised manuscript.

Line 182 – The use of BafA as an autophagy inhibitor, especially for long periods of time as in Fig 3C is probably not the best approach. There is evidence of expulsion of endolysosomal vesicles and significant disruption of the pathway with as much as 8 hours of BafA treatment. Later in the paper, the authors demonstrate blockade by for example VPS34 inhibitor, so this is probably a better experiment.

The reviewer rightfully points out that the VPS34 inhibitor would have been a better approach to address our question. We are therefore grateful for the reviewer’s comment, and we have now replaced the experiment in Fig. 3C with the same experimental set-up, however this time by using the Vps34-inhibitor. This showed that the VPS34-inhibitor inhibits mitochondrial degradation. We have therefore changed the text in the manuscript to:

To confirm that this mitochondrial turnover was mediated by autophagy, we repeated the experiment for WIPI1, WIPI2, and WIPI3 in the presence of a VPS34 kinase-inhibitor, a component of the PI3KC3-C1 complex (composed of VPS34, VPS15, Beclin1, and ATG14), which blocks autophagosome formation (Fig. 3c). VPS34-inhibitor treatment completely inhibited mitochondrial turnover, confirming that tethering WIPI1, WIPI2, WIPI3 to the mitochondrial surface is sufficient to induce mitophagy.

Line 219 – might be worth mentioning that other forms of mitophagy (i.e. Parkin dependent) – require TBK1 in certain contexts (PMID: 26365381, PMID: 37207627, PMID: 26266977).

We thank the reviewer for this suggestion and have incorporated this into the revised manuscript.

Line 276 – The AF multimer analysis in this section doesn't have the needed documentation of AF scores, etc, as was provided in some of the other AF sections (including in the supplemental figures). How good are the scores for the ATG13 analysis?

We thank the reviewer for this suggestion and agree that additional documentation on the AF prediction was missing, which will help readers interpret the data. We have, therefore, added the pLDDT and PAE plots as supplementary figure (Fig. S8) in addition to the ipTM score, which was already indicated in Fig. 6D.

Fig 7 addresses the interaction of other membrane-embedded receptors with WIPI initially using AF, with reasonable AF scores with TEX264 and FAM134C. A previous study has already demonstrated a possible interaction WIPI2 with TEX264 based on APEX of TEX264 (see Ref 29). In fact, it is one of the strongest hits in the proteomics. This result could be references as in cell evidence for the interactions predicted and examined in vitro.

We thank the reviewer for this suggestion and agree that our data nicely corroborate with these high-quality proximity labeling experiments from the Harper lab, identifying WIPI2 as the top hit for TEX264. We have therefore added the following sentence to the manuscript: *These results corroborate with the previous identification of WIPI2 as the strongest hit for TEX264 in proximity labeling experiments conducted in cells undergoing ER-phagy*²⁹.

Reviewer #3 (Remarks to the Author):

The BNIP3/NIX-mediated autophagy pathway is involved in the quality control of mitochondria, peroxisome, and in some cases, the endoplasmic reticulum. While the downstream events of BNIP3/NIX-mediated autophagy pathway have been well characterised, the upstream recruitment of general autophagy machineries has been assumed to follow the canonical autophagy activation dogma. In this work, the authors present unexpected data showing that BNIP3 and NIX do not directly recruit ULK1 complex, but instead, first recruits WIPI2/3 that bridges the ULK1 complex with BNIP3/NIX. Overall, this is a high-quality manuscript with elegant biochemical data supported by adequate cell biology observations. The manuscript is also well-written. The observations are also very timely and will be important to the BNIP/NIX-autophagy field. I only have a few questions and comments.

We would like to thank the reviewer for taking the time to carefully read our manuscript and are grateful for the kind words about our work. We have addressed the comments below in detail, which has allowed us to strengthen our manuscript considerably.

Comment 1 (Fig4F-I): The authors claim that WIPI2 initiates autophagosome formation together with ATG13, but still require ULK1-complex downstream to initiate mitophagy. This is demonstrated in Figure 4F-I, where depletion of ULK1 complex components or chemical inhibition of ULK1 causes inhibition of mitophagy. However, this data does not show whether WIPI2 is still recruited to mitochondria in these conditions. This leaves open the possibility of ULK1 inhibition having impact on WIPI2 recruitment. Therefore it would be important to demonstrate by immunofluorescence that WIPI2/3 recruitment to the mitochondria is not perturbed upon ULK1 inhibition.

We thank the reviewer for this comment and have performed the suggested experiment.

Based on our model, if WIPI2 is recruited upstream of FIP200/ULK1, then the accumulation of WIPI2 on the OMM should not be affected by the presence or absence of FIP200 or the ULK1 kinase activity. However, as correctly mentioned by the reviewer, we could not rule out that the ULK1 kinase activity would contribute to the WIPI2 recruitment. To strengthen this part of the manuscript further, we have therefore performed the following two experiments:

- (i) We performed IF experiments and analysed the WIPI2 recruitment in wild-type cells or wild-type cells treated with the ULK1 inhibitor. Please note that as the activity of BNIP3/NIX is dampened by PPTC7, we depleted PPTC7 to boost the number of cells undergoing mitophagy, making the approach more suitable for IF analysis. Doing so, we saw in wild-type cells that WIPI2 was recruited to the mitochondrial surface upon 24 h DFP treatment and could observe sporadic events of autophagosome-like structures that would engulf mitochondrial fragments. However, the addition of the ULK1 inhibitor to wild-type cells resulted in the disappearance of the autophagosome-like structures engulfing mitochondrial fragments but did not abrogate the recruitment of WIPI2 to the mitochondrial surface. This, therefore, indicates that the ULK1 kinase activity is required for mitophagosome formation but not for the recruitment of WIPI2 to the mitochondrial surface, which is consistent with our model.

Extended data Figure 6

- (ii) Supporting our model further, and as requested by Reviewer 1, we also analyzed WIPI2 recruitment in FIP200 knockout cells. If WIPI2 acts upstream of FIP200, as proposed by our model for NIX/BNIP3 mitophagy, then the absence of FIP200 should not prevent WIPI2 from being recruited to the mitochondrial surface. Indeed, we could also see WIPI2 recruitment to the OMM in FIP200 knockout cells, indicating that WIPI2 acts upstream of FIP200 and is consistent with the model we propose in this manuscript. In fact, we could even see a clear accumulation of WIPI2 on the OMM in FIP200 knockout cells. This is consistent with our other data showing that although downstream of WIPI2, FIP200 remains an essential factor for BNIP3/NIX mitophagy.

Figure 4

Together, these data support a model where WIPI2 is recruited directly by the cargo receptors NIX/BNIP3, upstream of FIP200 and ULK1. Moreover, while FIP200 and ULK1 are both essential for NIX/BNIP3 mitophagy, neither factor appears to be required for WIPI2 recruitment to the OMM.

We have added these data to Figure 4J and the Extended data figure 6.

Comment 2 (Fig 4A): Along the same line, while the ATG13 recruitment to mitochondria was shown to be dependent on WIPI2 in Fig 4A, that recruitment is induced using the artificial rapalog system. It will be important to show that upon DFP treatment, this recruitment is also dependent on WIPI2/3 using knockdown/knockout of WIPI2 and WIPI3.

We thank the reviewer for this comment and agree that such experiment would further strengthen our manuscript. However, despite our efforts to generate CRISPR KO lines for a combination of WIPIs, we have not succeeded in generating such cell lines.

We then tested whether we could combine WIPI2 KO cells with siRNAs against WIPI1 and WIPI3 to obtain complete WIPI1/2/3 depletion. Unfortunately, also via this strategy, we never succeeded in fully depleting all the WIPIs (potentially hinting at the essentiality of WIPI1-2-3 for HeLa cells – although the Mizushima lab succeeded in generating a full WIPI knockout line in HEK293T cells). Nevertheless, we decided to test whether these cells with considerable WIPI depletion would display mitophagy defects. Unfortunately, we did not observe mitophagy defects, suggesting that the low WIPI levels are sufficient to sustain NIX/BNIP3-driven mitophagy.

However, as we will show below in response to Comment 3, we did succeed in blocking a WIPI-dependent selective autophagy receptor while studying the ER-phagy receptor TEX264, whose turnover was blocked entirely in the absence of WIPI2 (see data in response to comment 3). The difference being that NIX/BNIP3 have the diversity to interact with WIPI1, WIPI2, and WIPI3 based on our in vitro experiments, while TEX264 would bind only to WIPI2, making it more susceptible and thus better suited for a WIPI-depletion experiment. See the next comment for more details.

Comment 3 (Line 350-351 – regarding Fig 8B): This is a generalisation based on the BNIP3/NIX observation which I think is a very strong statement. The authors would need to demonstrate cell biology that WIPI2/3 depletion has an impact on ER-phagy.

We thank the reviewer for this comment and implemented the following changes:

- (i) We toned down our sentence in the revised manuscript by introducing ‘might’: *Together, these findings suggest that WIPI-mediated autophagy initiation might represent a conserved mechanism across multiple organelles.*
- (ii) We added a reference to some beautiful work from the Harper lab, who performed mass spectrometry analysis and identified WIPI2 as the top hit in their cellular assays for TEX264 under ER-phagy conditions, which we believe nicely corroborates our findings. Therefore, we added the following sentence to the manuscript: *These results align with the previous identification of WIPI2 as the strongest hit for TEX264 in proximity labelling experiments conducted in cells undergoing ER-phagy²⁹.*
- (iii) We performed additional experiments to generate more support of this statement. As suggested by the reviewer, we used WIPI2 knockout cells and treated them with starvation medium to induce ER-phagy. We then blotted for TEX264 and CCPG1, which revealed that CCPG1 was degraded similarly in WIPI2 knockouts as in wild-type cells. However, TEX264 was no longer degraded in WIPI2 knockouts, indicating that the interaction between TEX264 and WIPI2 is critical during starvation-induced ER-phagy (consistent with WIPI2 appearing as a top hit in the proximity labeling experiments of the Harper lab). We also used a FIP200 knockout line as a control and analyzed it side-by-side with the wild-type and WIPI2 knockout cells. As expected, the degradation of both CCPG1 and TEX264 was blocked in FIP200 knockout cells.

Altogether, this supports a model where the WIPI2-binding by TEX264 is critical under certain ER-phagy conditions such as starvation.

Comment 4 (Fig 8E): While the data does show that both FIP200 and WIPI2d can bind TEX264/FKBP8 in an in vitro biochemical setting, it does not necessarily support the statement that TEX264 and FKBP8 can bind both FIP200 and WIPI2d at the same time. The authors should discuss the possibility of competitive binding, sequential binding, or FIP200 and WIPI2d being recruited to these receptors preferentially depending on physiological context.

We thank the reviewer for this comment. While our biochemical assays, in which both FIP200 and WIPI2 were present, we did not observe any signs of competitive binding; we agree that our reconstituted assays do not inform on physiological contexts where preferential binding might still occur. Therefore, we have amended the manuscript to tone down our suggestions (by stating ‘under the conditions tested’) and added that ‘we cannot rule out’. The revised manuscript, therefore, now reads: *This revealed that both TEX264 and FKBP8 can bind and recruit FIP200 and WIPI2d at the same time (Fig. 8e), with no indication of competitive binding or overlapping binding sites under the tested conditions. This suggests the potential formation of a mega-initiation complex. However, we cannot rule out the possibility of competitive or sequential binding to these receptors in cells, depending on the physiological context.*

Minor comments:

1. I believe Row 164’s reference to Fig 2I should be about Fig 2L

We thank the reviewer for this comment and apologize that this was insufficiently clear. In our submitted draft, the letter indicating the panel was written as non-capitalized and hence the l (L) may have looked like the i (I).

2. Please double check the labelling on Fig 7A, I believe the +/- signs on top of the blots are mislabelled

The reviewer correctly notes that the labelling of the blot is incorrect. We therefore thank the reviewer for pointing this out and apologize for the confusion it may have caused. We have now corrected the labelling in the revised manuscript.